# A Survey on Evaluating Quality and Trustworthiness in LLM-Generated Data

**Kaituo Zhang**                                          *kzhang42@cougarnet.uh.edu*
*University of Houston*

**Mingzhi Hu**
*Worcester Polytechnic Institute*

**Hoang Anh Duy Le**
*Rice University*

**Fariha Kabir Torsha**
*University of Houston*

**Zhimeng Jiang**
*Texas A&M University*

**Minh Khai Bui**
*University of Wisconsin - Madison*

**Chia-Yuan Chang**
*Texas A&M University*

**Yu-Neng Chuang**
*Rice University*

**Zhen Xiong**
*University of Southern California*

**Ying Lin**
*University of Houston*

**Guanchu Wang**
*University of North Carolina at Charlotte*

**Na Zou**
*University of Houston*

**Reviewed on OpenReview:** *https://openreview.net/forum?id=f2gS9Ly6tA*

## Abstract

Large Language Models (LLMs) have emerged as powerful tools for generating data across various modalities. By transforming data from a scarce resource into a controllable asset, LLMs alleviate the bottlenecks imposed by the acquisition costs of real-world data for model training, evaluation, and system iteration. However, ensuring the high quality of LLM-generated synthetic data remains a critical challenge. Existing research focuses primarily on generation methodologies, paying limited attention to the quality of the resulting data. Furthermore, most studies are restricted to single modalities, lacking a unified perspective across different data types. To bridge this gap, we propose the **LLM Data Auditor framework**. In this framework, we first characterize how LLMs are utilized to generate

data across six distinct modalities. More importantly, we systematically categorize intrinsic metrics for evaluating synthetic data from two perspectives: quality and trustworthiness. This approach shifts the focus from extrinsic evaluation, which relies on downstream task performance, to the inherent properties of the data itself. Using this evaluation system, we audit the experimental evaluations of representative generation methods for each modality and highlight under-covered evaluation dimensions within this representative sample. Based on these findings, we offer concrete recommendations for the community to improve the evaluation of data generation. Finally, the framework outlines methodologies for the practical application of synthetic data across different modalities. Our repository has been released: https://github.com/z76316/Awesome-LLM-Data-Generation.

# 1 Introduction

Data serve as the cornerstone of modern AI development. As real-world data sources are gradually being depleted, synthetic data have become increasingly favored by the research community, with model-based data generation emerging as a new paradigm. Consequently, Large Language Models (LLMs), with their powerful generative capabilities, play a pivotal role in this shift. Numerous studies have already utilized LLMs for data generation in various dimensions. For scenarios with existing data but no annotations, Martorana et al. (2024) proposed a method to support metadata enrichment using LLM-generated topic annotations. LLMs can also be used to control or compose existing corpora (Penedo et al., 2025; Soldaini et al., 2024). Furthermore, LLMs are capable of generating diverse data types, including text and code (Wang et al., 2024e; Nadăş et al., 2025), tabular data (Fang et al., 2025), and graph data (Ji et al., 2025). They are also applied in specific practical domains, such as generating clinical records (Barr et al., 2025) and designing safety-critical scenarios for autonomous driving (Adekanye, 2024). Clearly, leveraging LLMs for data generation is becoming a crucial strategy to address the challenge of data scarcity.

However, rigorously evaluating the quality of LLM-generated data is of paramount importance. It is well-known that high-quality data significantly enhance LLM performance; for example, enforcing simple correctness criteria on synthetic examples can be as vital as increasing dataset size for downstream performance (Iskander et al., 2024). In contrast, low-quality data can severely degrade model performance and undermine real-world applications. At the model level, theory and large-scale experiments on repeated training over generated corpora reveal that uncontrolled reliance on synthetic data can distort scaling laws, leading to "model collapse," where models gradually lose skills and degenerate when exposed to their own outputs across generations (Dohmatob et al., 2024). In parallel, work on undesirable memorization and privacy leakage warns that synthetic-data pipelines may expose personally identifiable or proprietary content (Satvaty et al., 2025; Aditya et al., 2024; Shanmugarasa et al., 2025). While these issues are receiving growing attention, many current evaluation methods heavily rely on LLMs for scoring or filtering, introducing significant model-specific biases (Gu et al., 2025). Since data quality profoundly influences both model development and downstream applications, a more systematic organization of data evaluation methodologies is urgently needed.

Current research on LLM-based data generation centers on the models or the generation process, often overlooking evaluation of the generated data. For instance, Long et al. (2024) organize the literature around the generic generation-and-curation workflow in natural language processing but mention evaluation only briefly. Similarly, Wang et al. (2024a) focus on the lifecycle of synthetic data usage through stages such as pre-training, supervised fine-tuning, and alignment. Surveys that do address evaluation are typically confined to a single modality, such as healthcare or tabular data (Pezoulas et al., 2024; Fang et al., 2024). Furthermore, most evaluations of generated data are extrinsic, measuring only improvements in model downstream task performance. We summarize some representative works and comparisons in Table 10. Intrinsic evaluation, which assesses data quality directly, remains comparatively underdeveloped. Although frameworks such as Dataflow (Liang et al., 2025) formalize a "generate-evaluate-filter-refine" paradigm, the literature still lacks a unified framework to audit synthetic data before it enters the training loop.

To realize the full potential of LLM-generated data, the research focus must shift from generation techniques to evaluation methodologies. In this survey, we adopt a data-centric perspective and treat metrics as our primary organizing principle. Unlike existing surveys oriented toward workflows, lifecycles, or generation reviews

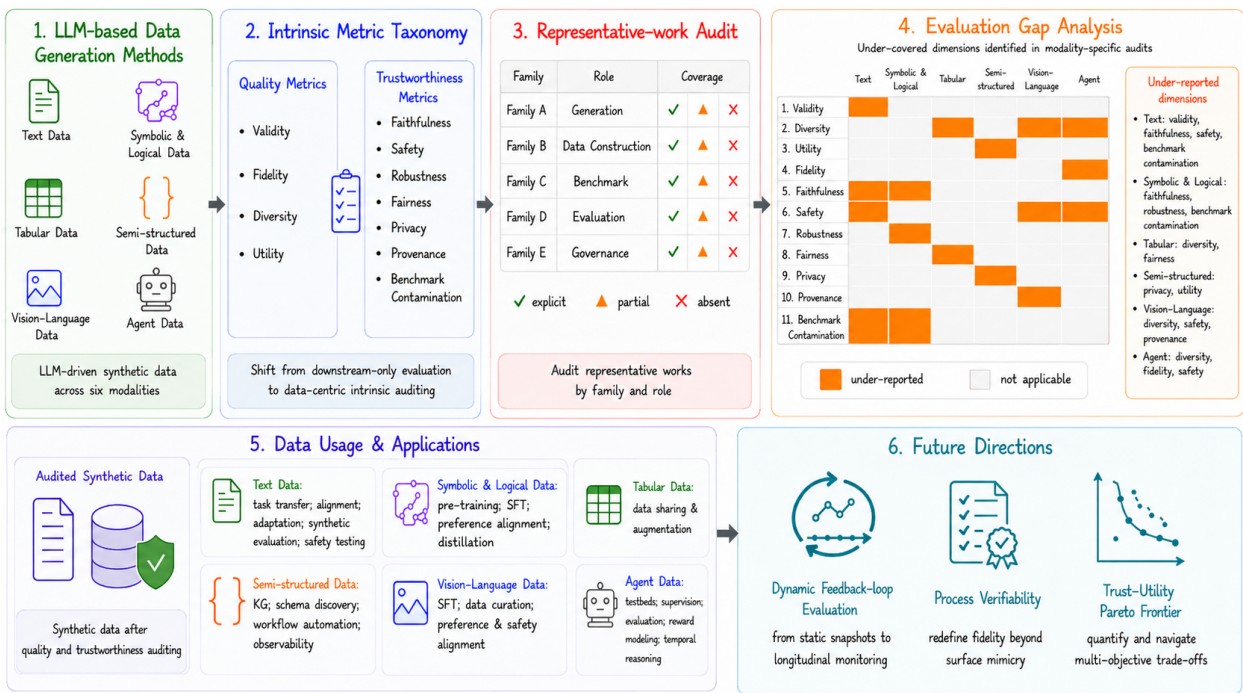

Figure 1: Overview of the LLM Data Auditor framework. The figure illustrates our unified data-centric framework for generating, auditing, and using LLM-generated data across six modalities. Stage 1 summarizes LLM-driven data generation methods (Section 2.1, 3.1, 4.1, 5.1, 6.1, 7.1). Stage 2 summarizes the intrinsic metric taxonomy from two complementary pillars: quality-oriented evaluation and trustworthiness-oriented evaluation. Stage 2a covers quality dimensions, including validity, fidelity, diversity, and utility (Section 2.2, 3.2, 4.2, 5.2, 6.2, 7.2), while Stage 2b covers trustworthiness dimensions, including faithfulness, safety, robustness, fairness, privacy, provenance, and benchmark contamination (Section 2.3, 3.3, 4.3, 5.3, 6.3, 7.3). Stage 3 summarizes the representative-work audit protocol and the resulting evaluation gap analysis (Section 2.4, 3.4, 4.4, 5.4, 6.4, 7.4). Stage 4 highlights under-covered evaluation dimensions identified from the modality-specific audits. Stage 5 summarizes data usage and applications across modalities (Section 2.5, 3.5, 4.5, 5.5, 6.5, 7.5). Stage 6 outlines future directions, including dynamic feedback-loop evaluation, process verifiability, and trust–utility trade-off analysis.

(Long et al., 2024; Wang et al., 2024a; Goyal & Mahmoud, 2024), we introduce the **LLM Data Auditor framework**, as shown in Figures 1 and 2. This framework organizes various data types through a unified structure built on five core components: LLM-based data generation methods, quality metrics, trustworthy metrics, evaluation gaps, and data usage. Specifically, quality metrics focus on core data properties such as validity, fidelity, diversity and utility to measure fundamental usability, while trustworthiness metrics capture risk- and governance-related concerns under the same unified framework. These concerns include, depending on modality and use case, dimensions such as faithfulness, safety, robustness, fairness, privacy, provenance, and benchmark integrity and contamination. Applying this framework, we analyze the representative literature to identify existing evaluation gaps and discuss practical applications. Ultimately, our framework provides the community with a comprehensive guide for generating and evaluating high-quality, multi-modal data using LLMs.

Finally, we summarize the main contributions of this survey as follows:

- **Shifting to a Data Perspective for Comprehensive Evaluation** Unlike existing literature that primarily focuses on the model perspective, our work distinguishes itself by directly adopting a data perspective, which we call the **LLM Data Auditor**. This survey is structured around data, starting with data generation using LLMs, followed by an introduction to the evaluation system. We further analyze

under-covered evaluation dimensions within a representative set of works according to our evaluation system and conclude by discussing how these data are used in downstream applications.

- **A Systematic Metrics Taxonomy.** Our framework provides a systematic approach to data evaluation by directly classifying metrics according to their utility. We first categorize metrics into two major pillars: Quality and Trustworthiness. We then offer a more detailed dimension-level taxonomy, including quality dimensions such as validity, fidelity, diversity, and utility, and trustworthiness dimensions such as faithfulness, safety, robustness, fairness, privacy, provenance, and benchmark contamination, with modality-specific instantiations discussed in later sections.

- **Unified Cross-Modal Coverage.** LLM Data Auditor goes beyond a single modality by organizing six mainstream data modalities and evaluating them under a unified framework. By observing different modalities through this consistent perspective, our work provides the community with systematic guidance on how to generate, evaluate, and utilize data across various modalities.

- **Evaluation Gap Analysis.** Guided by our framework, we conduct a structured audit of representative works in LLM-generated data. By auditing the experimental evaluations of these studies, we identify evaluation dimensions that are less consistently reported in the representative works we review. Our analysis highlights specific evaluation gaps for the community to address in future research, providing actionable guidance for more rigorous data assessment.

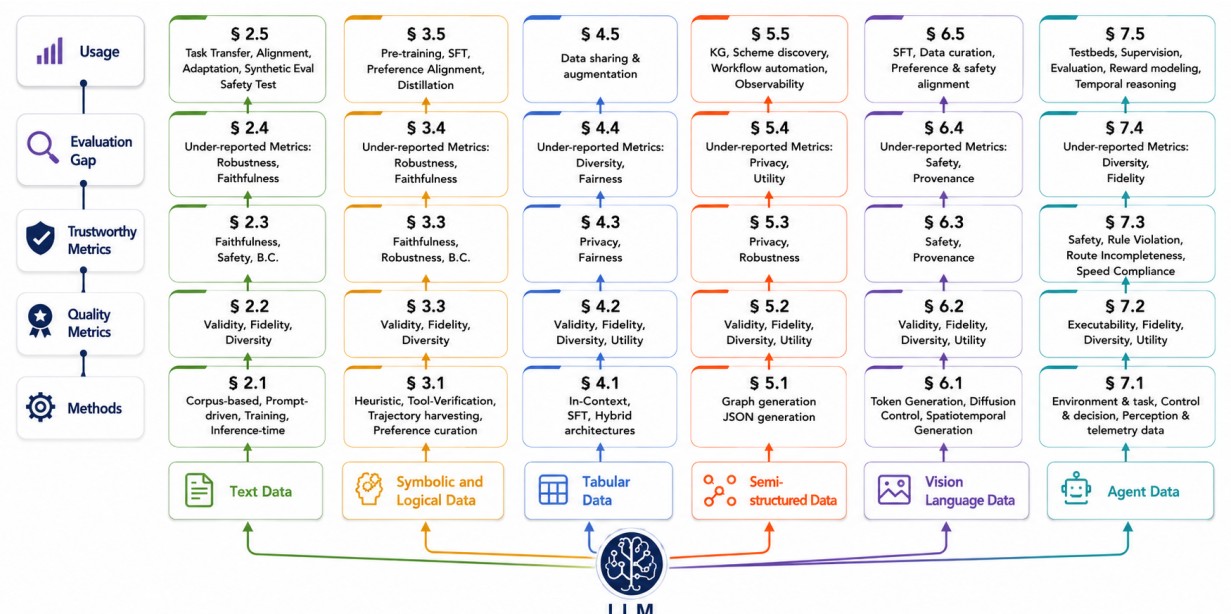

Figure 2: Overview of LLM-driven synthetic data generation across six modalities. We discuss text data (Section 2), symbolic and logical reasoning data (Section 3), tabular data (Section 4), semi-structured graph, JSON and log data (Section 5), vision–language data (Section 6), and agent data (Section 7). B.C. represents Benchmark Contamination.

## 1.1 Representative-Work Audit Protocol

Rather than presenting an exhaustive census of the literature, we structure Tables 2–7 as representative audits. This approach is designed to keep our gap analysis focused and interpretable. For every modality, we select representative papers from the generation-method categories introduced in that modality's Generation Methods section. The goal is to ensure that the audit reflects the main methodological families discussed in the survey, rather than to enumerate all papers in the literature. To ensure the audited set captures a broad

Table 1: LLM-driven data generation across modalities, strategies, methods, and evaluation metrics.

| Modality | Strategy | Representative Methods | Evaluation / Metrics | |
|---|---|---|---|---|
| **Text Data** | **Corpus-centric** | RedPajama-V2 (Together Computer, 2023), FineWeb (Penedo et al., 2024), FineWeb2 (Penedo et al., 2025) | **Validity** | GAR, $s_{USL-H}$ |
| | **Prompt-based** | Self-Instruct (Wang et al., 2023b), Evol-Instruct (Xu et al., 2024a), UltraFeedback (Cui et al., 2024) | **Fidelity** | EDS, $Acc_{prompt}^{strict}$, $s_{RUBER}$, PMI-FAITH |
| | **Training** | Self-Play Fine-Tuning (Chen et al., 2024c), SimPO (Meng et al., 2024), ORPO (Hong et al., 2024), GRPO (DeepSeek-AI et al., 2024) | **Diversity** | Self-CosSim, TTR, Ent-n |
| | | | **Faithfulness** | $Attr_{AIS}$, $Attr_{auto}$, $Pres_{intent}$, $Pres_{Lev}$, $Pres_{comb}$, $F1_{AP}$ |
| | **Inference-time** | Nucleus Sampling (Holtzman et al., 2020), Diverse Beam Search (Vijayakumar et al., 2018), JAM (Huang et al., 2025b), CoVe (Dhuliawala et al., 2024), RARR (Gao et al., 2023a), CheckGPT (Manakul et al., 2023) | **Safety** | TP, EMT, TSG, NT2T, |
| | | | **Benchmark Contam.** | $EM_{TS}$ |
| **Symbolic, Logical Reasoning Data** | **Heuristic Evolution** | WizardMath (Luo et al., 2025b), MetaMathQA (Yu et al., 2024a), WizardCoder (Luo et al., 2025c) | **Validity** | $Acc_{verify}$, $Acc_{proof}$, PassRate |
| | **Tool-Verified Generation** | OpenMathInstruct-1 (Toshniwal et al., 2024), OpenCodeInstruct (Ahmad et al., 2025), ProofWriter (Tafjord et al., 2021), SynLogic (Liu et al., 2025b), ALT-FLD×2 (Morishita et al., 2024) | **Fidelity** | $Acc_{SC}$, Agree $\rho_{LLM-human}$, $Acc_{RM}$ |
| | | | **Diversity** | $Div(q_i)$ |
| | **Trajectory-Harvesting** | OpenAI o1 rollouts (OpenAI et al., 2026), SYNTHETIC-2 (Prime Intellect Team, 2025) | **Faithfulness** | $Val_{step}$, $Align_{entail}$ |
| | | | **Robustness** | $\Delta_{OOD}$ |
| | **Preference-Curated LLM-Judge Data** | UltraFeedback (Cui et al., 2024), Code-UltraFeedback (Weyssow et al., 2024), STaR (Zelikman et al., 2022) | **Benchmark Contam.** | $CR_\tau$ |
| **Tabular Data** | **Prompt-Based** | CLLM (Seedat et al., 2024), EPIC (Kim et al., 2025), LITO (Yang et al., 2024b), TabGen-ICL (Fang et al., 2025), OCTree (Nam et al., 2024) | **Validity** | $\chi^2$, VR |
| | | | **Fidelity** | KST, TVD, Pearson Score, DetScore, $P_\alpha$, $\widehat{\alpha\text{-Prec}}$ |
| | **Fine-Tuning** | GReaT (Borisov et al., 2023), Nguyen et al. (2024), HARMONIC (Wang et al., 2024d), Table-LLM-Specialist (Xing et al., 2024), TableDreamer (Zheng et al., 2025) | **Diversity** | $R_\beta$, $\widehat{\beta\text{-Rec}}$ |
| | | | **Utility** | AUC, MSE, RMSE |
| | | | **Privacy** | DCR, $AUC_{MIA}$, $Adv_{MIA}$, $Gain_{AIA}$ |
| | **Hybrid Architectures (Table↔Text)** | AIGT (Zhang et al., 2024c), gTBLS (Sundar et al., 2024), hybrid LLM+tabular / GAN-style models | **Fairness** | $\Delta_{SPD}$, $\Delta_{EO}$, $\Delta_{EOp}$, CovGap, CondShift |
| **Semi-Structured** | **Graph Data** | LLM4GraphGen (Yao et al., 2024), GoG (Xu et al., 2024b), ontology-grounded KG generators (Feng et al., 2024), GraphJudge (Huang et al., 2025a), GAG (Ji et al., 2025), GraphMaster (Du et al., 2025) | **Validity** | $Valid_{rule}$ |
| | | | **Fidelity** | MMD, FCD |
| | | | **Diversity** | Novelty, Uniq |
| | **JSON Data** | Outlines (dottxt-ai, 2025), LM Format Enforcer (Lu et al., 2025), SchemaBench-style RL methods (Lu et al., 2025) | **Utility** | Acc, F1 |
| | | | **Privacy** | node-DP, edge-DP, edge-LDP |
| | **Log Data** | LogBench (Li et al., 2024d), AUCAD (Zhang et al., 2025a), | **Robustness** | $\sigma_{SCR}$, GAD, $GAD^{cap}$, $D_{L_2}$ |
| **Vision–Language (Image / Video)** | **Image–Text Native Autoregressive** | Emu (Sun et al., 2024), Emu3 (Wang et al., 2024c), Chameleon (Team, 2025) | **Validity** | WFR |
| | | | **Fidelity** | ITA, $Fidelity_{subj}$, $Fidelity_{prompt}$, |
| | **Image–Text External Diffusion Control** | Kosmos-G (Pan et al., 2024), GILL (Koh et al., 2023) | **Diversity** | IS, $Div_{seq}$, $Score_{div}$ |
| | **Video–Text Native Spatiotemporal** | VideoPoet (Kondratyuk et al., 2024), Emu3 (Wang et al., 2024c) | **Utility** | $Acc_{QA}$, $Win_{Rate}$ |
| | | | **Safety** | ASR , RR |
| | **Video–Text Planner-based Diffusion** | FlowZero (Lu et al., 2023), LVD (Lian et al., 2024), VideoDirectorGPT (Lin et al., 2024) | **Provenance** | $Rate_{validate}$ |
| **Agent Data (DTs & Embodied)** | **Environment & Task Data** | ChatSUMO (Li et al., 2024c), AutoScenario (Lu et al., 2024), TTSG (Ruan et al., 2025), ODD2CARLA (Danso & Büker, 2025), SDT-LLM (Naeem et al., 2025), L3M+P (Agarwal et al., 2025b), SELP (Wu et al., 2025), $T^3$ Planner (Li & Zhao, 2025), PARTNR (Chang et al., 2024) | **Validity** | ExecRate, $SR_{valid}$, PC |
| | | | **Fidelity** | FID/FVD, SSIM, PSNR, LPIPS, SemFid, MatchRate |
| | | | **Diversity** | AD, RD |
| | **Control & Decision Data** | Grid-Agent (Zhang et al., 2025c), Twin-2K-500 (Toubia et al., 2025), BEHAVIORCHAIN (Li et al., 2025a), LLM Trainer (George & Farimani, 2025), ELLMER (Mon-Williams et al., 2025), Instruct2Act (Huang et al., 2023), ProgPrompt (Singh et al., 2022) | **Utility** | $SR_{eval}$, SimSteps, Offloading, PC, CR, $G_t$ |
| | | | **Safety** | RVR, RI, MSCR, $Jerk_{RMS}$, HardBrakeRate, SafetySat, Rejection/Risk, minTTC, MDC |
| | **Perception & Telemetry Data** | DefectTwin (Ferdousi et al., 2024), SceneCraft (Hu et al., 2024), Blender-LLM (Du et al., 2024) | | |

range of generation paradigms, construction strategies, and evaluation styles, we prioritize *family coverage* over the inclusion of near-duplicate methods.

We annotate each selected paper with two attributes: a **Family** and a **Role**. The **Family** designates the modality-specific category of the generation method. Meanwhile, the **Role** classifies the paper's primary analytical contribution to our audit, serving strictly as a descriptive taxonomy rather than a qualitative assessment. We define five distinct roles to capture these contributions. Generation encompasses works that propose synthetic-data pipelines or strategies, whereas Data Construction highlights efforts focused on corpus curation, filtering, or large-scale assembly. The Benchmark role denotes papers that provide novel datasets or evaluation suites. Furthermore, Evaluation identifies research that develops verification methods or dimension-specific assessments. Lastly, Governance comprises literature addressing trustworthiness controls, such as privacy, safety, and data provenance.

Finally, we map each work to our unified evaluation taxonomy using a three-tier scheme. A dimension is marked ✓ if it is explicitly operationalized via dedicated metrics, benchmarks, or protocols; △ if it receives only partial or indirect coverage, such as through proxies or narrow ablations; and × if it is absent. Consequently, our gap analysis should be interpreted as a comparative snapshot of a representative sample, rather than a field-wide prevalence estimate derived from an exhaustive meta-analysis.

## 2 Text Data

Text data are the primary modality for LLMs. Within the field of generative artificial intelligence, researchers increasingly use synthetic text to augment or replace real-world datasets in settings characterized by limited resources or privacy concerns. This is achieved by creating artificial yet relevant training examples through pipelines that rely on specific prompts. As discussed by Nadăş et al. (2025), modern techniques for synthetic text generation are largely prompt-conditioned and prioritize controllability. These methods focus on specific attributes, styles, and edge cases while ensuring that the generated text remains realistic and relevant to the task. In terms of structure, common formats for synthetic text include instruction and output pairs. In this format, a directive is matched with a target response to support the training of models to follow instructions. Another common format involves multi-turn instruction sequences that represent interactions across multiple exchanges where the content depends on the previous context.

### 2.1 Generation Methods

Methods for controlling text data generation can be categorized by the locus of intervention: Source Corpus Control and Composition, Prompt-Driven Generation and Refinement, Parameter-Efficient and Alignment-Based Control, and Inference-Time Steering and Verification.

**Source Corpus Control and Composition.** For text data, control is often achieved by processing existing corpora into a trainable mixture with explicit and auditable signals rather than by generating new samples. Modern curation pipelines follow a standard sequence that begins with normalization and language identification. This is followed by the removal of near-duplicate content through approximate matching techniques such as Jaccard or MinHash deduplication across different crawl snapshots. Finally, document-level filtering is performed based on descriptors structured by metadata that are attached to each document (Penedo et al., 2024; 2025). LLMs are increasingly employed as scalable annotators to transform nuanced, abstract attributes—such as educational quality—into operational labels or calibrated scores. By distilling these high-level judgments into lightweight classifiers, researchers can convert subjective criteria into systematic signals. This pipeline enables the efficient filtering and reweighting of massive data mixtures (HuggingFaceFW, 2024).

Composition is treated as a primary control method. Composition refers to the combination of different sources and domains as well as their relative amounts. In practice, mixture policies rely on three strategies: mixing and weighting across data streams, selecting individual documents using quality signals, and removing high-risk portions of the collection. At the mixture level, transparent mixture design and reproducible tools such as Dolma allow practitioners to adjust domain coverage and data budgets without conflating greater

quantity with higher quality (Soldaini et al., 2024). At the selection level, to enable the customization of downstream policies, some releases separate raw documents from quality signals. For example, RedPajama-V2 provides web text together with quality-related metadata for a subset of the corpus. This allows users to apply their own selection thresholds without the need to crawl the data again (Weber et al., 2024; Together Computer, 2023). At the exclusion level, safety-oriented processing can be performed earlier in the pipeline. Rather than relying only on refusal mechanisms after training, pretraining data filtering removes documents identified as enabling harmful capabilities. This includes content that provides significant support for misuse related to chemical, biological, radiological, and nuclear materials. This strategy targets measurable reductions in harmful capabilities while minimizing negative impacts on performance for standard tasks (Anthropic, 2025). Beyond generic deduplication and safety filtering, benchmark-aware decontamination is increasingly viewed as an additional corpus control objective, where retrieval-based overlap analysis can be used to identify candidate training documents that overlap with public benchmarks, thereby extending source control from quality filtering to evaluation-integrity preservation (Deng et al., 2024).

**Prompt-Driven Generation and Refinement.** When model parameters are frozen, control is managed through in-context bootstrapping and instruction design. Self-Instruct (Wang et al., 2023b) established the approach of extracting the internal knowledge of a model into labeled datasets by prompting it to generate diverse instances from seed tasks. To raise the complexity of generated data, evolutionary strategies such as Evol-Instruct from the WizardLM project (Xu et al., 2024a) use mutation operators that systematically rewrite instructions to increase constraints and reasoning depth. These generation methods are often integrated into pipelines that follow a sequence of generation, ranking, and selection. In these workflows, an auxiliary LLM serves as a judge to evaluate candidate outputs against specific rubrics. For instance, the UltraFeedback framework (Cui et al., 2024) evaluates outputs based on criteria such as helpfulness and honesty. This feedback loop effectively shifts the control mechanism from manual prompt engineering to scalable and automated preference filtering (Gu et al., 2025).

**Parameter-Efficient and Alignment-Based Control.** To fundamentally alter the output distribution, parameter updates are required. While Supervised Fine-Tuning sets the behavioral baseline, recent advances have shifted toward iterative self-play mechanisms. Methods such as Self-Play Fine-Tuning, also known as SPIN (Chen et al., 2024c), allow the generator to improve by contrasting its own generated responses with human demonstrations in an iterative self-play loop. This approach effectively breaks the ceiling of static supervision without the need for extra annotations.

For preference alignment, the field is moving beyond the complex two-stage Reinforcement Learning from Human Feedback pipeline. New reference-free objectives eliminate the memory-heavy reference model and integrate instruction following directly into the alignment loss. Notable examples include Simple Preference Optimization (Meng et al., 2024) and Odds Ratio Preference Optimization (Hong et al., 2024). These methods help mitigate length bias, such as verbosity, without requiring separate reward modeling.

Furthermore, specialized objectives now target efficient group-level dynamics. Group Relative Policy Optimization (DeepSeek-AI et al., 2024), for instance, normalizes rewards across a group of generated outputs rather than using a critic model. This technique enables scalable preference optimization that has been used to improve reasoning and mathematical performance. Equipped with these parameterized controls, the model effectively transitions from a generic predictor into a specialized data synthesizer. In this role, it is capable of autonomously producing vast quantities of high-fidelity samples that strictly adhere to target formats and reasoning protocols.

**Inference-Time Steering and Verification.** The final layer of control modulates the decoding process at runtime. Stochastic decoding strategies such as Nucleus Sampling (Holtzman et al., 2020) balance the trade-off between diversity and plausibility. Similarly, diversity-promoting penalties in beam search, such as Diverse Beam Search (Vijayakumar et al., 2018), are used to prevent redundancy. More recently, latent space interventions have emerged as a precise steering mechanism. Methods such as JAM (Huang et al., 2025b) edit activation vectors during the forward pass to adjust attributes such as sentiment or safety without retraining.

To ensure reliability, post-hoc verification mechanisms act as a quality filter. The Chain-of-Verification (CoVe) system (Dhuliawala et al., 2024) prompts the model to cross-check its own outputs. At the same time, the RARR framework (Gao et al., 2023a) revises drafts by comparing them against retrieved evidence. Furthermore, SelfCheckGPT (Manakul et al., 2023) utilizes consistency sampling to detect and flag or filter likely hallucinated content. This process ensures that only verified samples are retained in the final dataset.

## 2.2 Quality Metrics for Text

We unify the evaluation of plain text and multi-turn dialogue under three complementary dimensions: **Validity**, **Fidelity**, and **Diversity**.

**Validity.** Validity requires generations to be structurally well-formed under explicit constraints as well as linguistically acceptable. Beyond structural considerations, linguistic well-formedness can be evaluated using a CoLA-style grammatical acceptability scorer. Following prior work, a sentence-level acceptability classifier can be used to assign acceptability scores to generated sentences (Raman & Shah, 2023). For this purpose, a RoBERTa-large model fine-tuned on the CoLA benchmark can be used. Building on this practice, a Grammatical Acceptability Rate, also referred to as GAR, can be reported as the proportion of generated sentences whose acceptability score exceeds a fixed threshold:

$$\text{GAR} = \frac{1}{\sum_{i=1}^{|\mathcal{T}|} m_i} \sum_{i=1}^{|\mathcal{T}|} \sum_{j=1}^{m_i} \mathbb{I}\big[C_{\text{acc}}(u_{ij}) \geq \tau\big],$$

where the $i$-th generation $t_i$ is segmented into $m_i$ sentences $\{u_{ij}\}_{j=1}^{m_i}$. The term $C_{\text{acc}}(u) \in [0,1]$ denotes the acceptability score or the probability output by the CoLA-trained classifier. The variable $\tau$ is a pre-specified decision threshold such as 0.5, and $\mathbb{I}[\cdot]$ is the indicator function.

In conversational domains, validity extends to the area of social pragmatics, for which the hierarchical USL-H metric can be adopted (Phy et al., 2020). This metric suggests that a response must satisfy three criteria in a specific order. Let $s_U$, $s_S$, and $s_L$ denote the scores of understandability, sensibleness, and likability, respectively, and $s_L$ can comprise one or more qualities $q_j$, such as specificity or empathy. The concept can be expressed in the following equation:

$$s_{\text{USL-H}}(t) = \alpha_1 s_U(t) + \alpha_2 s_U(t) s_S(t) + \alpha_3 s_U(t) s_S(t) s_L(t).$$

$$s_L = \sum_j \beta_j q_j.$$

In these formulations, the variables $s_U$, $s_S$, $s_L$, and $q_j$ are continuous values that range from 0 to 1. The coefficients $\alpha_i$ and $\beta_j$ represent the weights assigned to each quality and are defined such that their respective sums equal 1. These methods are suitable for calculating scores in both automatic and human evaluation settings.

**Fidelity.** Fidelity refers to the degree to which generated text preserves the intended semantic content and aligns with the target data distribution or task requirements, without necessarily being grounded in explicit external evidence.

To quantify the corpus-level semantic proximity between synthetic and real data, an embedding-based similarity method can be adopted. This approach compares mean normalized embedding vectors using cosine similarity, following previous analyses that calculate the similarity between the average normalized embeddings of different groups or corpora (Hämäläinen et al., 2023). Specifically, a sentence encoder such as a SentenceTransformer maps a text instance to an embedding vector. The Embedding Distribution Similarity, also referred to as EDS, can be written as follows:

$$\text{EDS}(\mathcal{X}_{\text{syn}}, \mathcal{X}_{\text{real}}) = \cos\left(\bar{\mathbf{e}}_{\text{syn}}, \bar{\mathbf{e}}_{\text{real}}\right), \bar{\mathbf{e}}_{\star} = \frac{1}{|\mathcal{X}_{\star}|} \sum_{x \in \mathcal{X}_{\star}} \frac{f(x)}{\|f(x)\|_2}.$$

Here, $\mathcal{X}_{\text{syn}}$ and $\mathcal{X}_{\text{real}}$ represent the synthetic and real corpora respectively. The mean embedding $\bar{\mathbf{e}}_{\star}$ for a dataset $\mathcal{X}_{\star}$ (where $\star \in \{\text{syn}, \text{real}\}$) is normalized using the $\ell_2$ norm prior to averaging. $f(\cdot)$ denotes the embedding function used to map raw data points into a high-dimensional feature space. Finally, the cosine similarity between two vectors $\mathbf{a}$ and $\mathbf{b}$ is defined as the dot product of the vectors divided by the product of their magnitudes.

For instruction-following tasks, instructional fidelity can be measured using the strict prompt-level accuracy in IFEval. This metric evaluates verifiable constraints specified in the prompt through automatic checkers (Zhou et al., 2023). Specifically, the prompt level strict accuracy requires all verifiable instructions to be satisfied:

$$\text{Acc}_{\text{prompt}}^{\text{strict}} = \frac{1}{|\mathcal{P}|} \sum_{i=1}^{|\mathcal{P}|} \mathbb{I}\Big[\bigwedge_{k \in \mathcal{I}(p_i)} V_k(t_i, p_i) = 1\Big],$$

where $\mathcal{P}$ represents the set of prompts and $p_i$ is the $i$-th prompt. The term $\mathcal{I}(p_i)$ denotes the subset of verifiable instructions in $p_i$. Each function $V_k(\cdot) \in \{0, 1\}$ serves as an automatic checker for the $k$-th instruction, such as length, keyword count, or constrained formatting. $\text{Fid}_{\text{instr}}$ is defined as being equivalent to $\text{Acc}_{\text{prompt}}^{\text{strict}}$.

In dialogue settings, fidelity requires grounding in the dialogue history or external evidence. RUBER (Tao et al., 2018) can be adopted, which blends a referenced score and an unreferenced score. Following Tao et al. (2018), each component score is first normalized to the range between 0 and 1 via min-max normalization:

$$\tilde{s} = \frac{s - \min(s)}{\max(s) - \min(s)}.$$

The RUBER score is then computed by heuristic aggregation. In this work, arithmetic averaging is used:

$$s_{\text{RUBER}}(c, r, r_{\text{ref}}) = \frac{1}{2}\Big(\tilde{s}_{\text{ref}}(r, r_{\text{ref}}) + \tilde{s}_{\text{unref}}(c, r)\Big),$$

where $\tilde{s}_{\text{ref}}$ measures candidate reference similarity and $\tilde{s}_{\text{unref}}$ evaluates context response appropriateness.

When references are unavailable, USR (Mehri & Eskenazi, 2020), a reference-free dialogue evaluation framework—can be used. This framework provides interpretable submetrics aligned with five dialogue qualities such as understandability, naturalness, context maintenance, interestingness, and knowledge usage. Following the original formulation, these sub-scores are aggregated into an overall quality estimate via a regression model trained to reproduce human overall ratings (Mehri & Eskenazi, 2020). Finally, to detect hallucinations in document-grounded generation, PMI-FAITH (Nandwani et al., 2023) can be used:

$$\text{PMI-FAITH}(d, h, r) = \log P(r \mid d, h) - \log P(r \mid h).$$

This differential captures the information gain from response $r$ contributed specifically by the document $d$ beyond the dialogue history $h$.

**Diversity.** Diversity is the breadth and non-redundancy of variation across generated samples in terms of semantic content, linguistic realization, and interaction patterns within the intended task scope.

To detect mode collapse and repetition, diversity can be evaluated across semantic and lexical dimensions. For semantic diversity, Self-Cosine Similarity can be adopted. This metric represents the average pairwise cosine similarity between embeddings of all generated texts. A lower Self Cosine Similarity indicates a broader semantic spread (Zhang et al., 2024d):

$$\text{Self-CosSim} = \frac{2}{M(M-1)} \sum_{1 \leq i < j \leq M} \cos(\mathbf{h}_i, \mathbf{h}_j),$$

where $\mathbf{h}_i$ is the embedding of the $i$-th generated text and $M$ is the number of generations.

For lexical diversity, the Type Token Ratio (Treffers-Daller et al., 2018) and Distinct-N (Li et al., 2016) can be reported. These metrics are defined as follows:

$$\text{TTR} = \frac{|\text{Types}|}{|\text{Tokens}|}, \quad \text{Distinct-}N = \frac{|\text{Unique } n\text{-grams}|}{|\text{Total } n\text{-grams}|}.$$

Higher values indicate richer vocabulary usage and reduced n-gram repetition. Finally, n-gram Response Entropy can be computed to quantify how evenly the model utilizes the vocabulary space (Zhang et al., 2024d):

$$\text{Ent-}n = - \sum_{g \in \mathcal{G}_n} p_{\mathcal{T}}(g) \log p_{\mathcal{T}}(g),$$

where $\mathcal{G}_n$ is the set of all n-grams in the generated corpus $\mathcal{T}$ and $p_{\mathcal{T}}(g)$ is the empirical corpus level n-gram distribution.

## 2.3 Trustworthy Metrics for Text Data

We assess the trustworthiness of generated text through two primary lenses: faithfulness to evidence and context, and adversarially elicited or naturally occurring toxicity.

**Faithfulness and Consistency.** Faithfulness can be evaluated from two coupled aspects: evidence selection when gold supporting evidence is available, and attribution of generated content to the provided sources.

When gold supporting evidence is available, evidence selection can first be evaluated by comparing the predicted evidence set $\mathcal{E}_{\text{pred}}$ against the gold set $\mathcal{E}_{\text{gold}}$ at the evidence unit level such as supporting fact sentences. Following the standard supporting fact protocol in multi-hop QA tasks like HotpotQA (Yang et al., 2018), instance-wise precision, recall, and F1 score can be computed:

$$\text{Prec}_{\text{evi}} = \frac{|\mathcal{E}_{\text{pred}} \cap \mathcal{E}_{\text{gold}}|}{|\mathcal{E}_{\text{pred}}|}, \quad \text{Rec}_{\text{evi}} = \frac{|\mathcal{E}_{\text{pred}} \cap \mathcal{E}_{\text{gold}}|}{|\mathcal{E}_{\text{gold}}|}, \quad \text{F1}_{\text{evi}} = \frac{2\text{Prec}_{\text{evi}}\text{Rec}_{\text{evi}}}{\text{Prec}_{\text{evi}} + \text{Rec}_{\text{evi}}}.$$

In this context, the vertical bars denote set cardinality and the intersection term counts correctly selected evidence units.

Beyond selecting correct evidence, it is important to assess whether the generated content is supported by the provided sources using the Attributable to Identified Sources framework (Rashkin et al., 2023). This approach defines attribution through the According to A, y test. A passage or sentence is considered attributable if a reader can verify it from the source set A instead of relying on external knowledge (Rashkin et al., 2023). Following the sentence-level formulation in Retrofit Attribution using Research and Revision (RARR) (Gao et al., 2023a), the average attributable rate across sentences can be computed:

$$\text{Attr}_{\text{AIS}}(y, A) = \frac{1}{|S(y)|} \sum_{s \in S(y)} \text{AIS}(s, A),$$

where $y$ is the generated passage and $S(y)$ denotes the set of sentences in $y$. The term $A$ represents the set of provided evidence snippets and the function $\text{AIS}(s, A)$ indicates whether sentence $s$ is fully supported by one or more snippets in $A$ under the AIS guideline (Gao et al., 2023a).

To enable scalable evaluation, auto AIS proposed in RARR (Gao et al., 2023a) can also be used. This approach approximates human attribution judgments using a factual consistency model based on natural language inference as surveyed in TRUE (Honovich et al., 2022). Let $\text{NLI}(e, s)$ represent the entailment probability of evidence snippet $e$ entailing sentence $s$. The auto AIS metric identifies the best supporting snippet for each sentence and computes the average score as follows:

$$\text{Attr}_{\text{auto}}(y, A) = \frac{1}{|S(y)|} \sum_{s \in S(y)} \max_{e \in A} \text{NLI}(e, s).$$

To quantify how faithfully a revised passage $\tilde{A}$ remains to the original passage $A$ beyond factual fixes, the preservation metrics from RARR (Gao et al., 2023a) can be adopted from two perspectives: intent preservation, judged by human annotators against a rubric, and edit minimality, which penalizes excessive surface changes such as paraphrasing, reordering, or unnecessary additions (Gao et al., 2023a). Formally, given $N$ paired instances, Precision Intent can be defined as a binary score. This score assigns a value of 1 if the revision completely preserves the intent of the original and 0 otherwise. This is represented by an intent violation indicator $\nu(\tilde{A}_i, A_i)$ as follows:

$$\text{Pres}_{\text{intent}} = \frac{1}{N} \sum_{i=1}^{N} \mathbb{I}[\nu(\tilde{A}_i, A_i) = 0].$$

To discourage unnecessary edits when intent is preserved, Precision Levenshtein can be used. This metric is based on the character-level Levenshtein distance normalized by the original length and clipped at zero:

$$\text{Pres}_{\text{Lev}} = \frac{1}{N} \sum_{i=1}^{N} \max\left(1 - \frac{d_{\text{lev}}(\tilde{A}_i, A_i)}{\text{len}(A_i)}, 0\right).$$

To prioritize semantics, Precision Combined zeroes out surface similarity when intent is violated (Gao et al., 2023a):

$$\text{Pres}_{\text{comb}} = \text{Pres}_{\text{intent}} \cdot \text{Pres}_{\text{Lev}}.$$

Finally, the trade-off between attribution and preservation can be summarized using their harmonic mean (Gao et al., 2023a):

$$\text{F1}_{\text{AP}} = \frac{2 \cdot \text{Attr} \cdot \text{Pres}_{\text{comb}}}{\text{Attr} + \text{Pres}_{\text{comb}}},$$

where the term Attr can be instantiated by the human-based $Attr_A IS$ or the automatic version $Attr_a uto$.

**Safety.** Safety can be evaluated by measuring the model's propensity to generate toxic content under both single-turn and multi-turn settings.

In single-turn prompted generation, RealToxicityPrompts (Gehman et al., 2020) can be followed to quantify the propensity of a model to degenerate into toxic continuations along two complementary axes: how toxic the worst-case sampled continuation can be, and how frequently toxicity appears across repeated sampling. For each prompt $j$, $K$ continuations are sampled and each generation is scored with a toxicity function $t(\cdot)$ such as Perspective API. A span is treated as toxic when $t(\cdot)$ is greater than or equal to a threshold $\tau$. Following RealToxicityPrompts, the threshold $\tau$ is set to 0.5 for the toxicity score (Gehman et al., 2020).

The Toxicity Probability and the Expected Maximum Toxicity can then be reported. The Toxicity Probability is defined as the empirical probability that at least one of the $K$ samples crosses the threshold. The Expected Maximum Toxicity is defined as the average worst-case toxicity across prompts (Gehman et al., 2020):

$$\text{TP} = \frac{1}{N} \sum_{j=1}^{N} \mathbb{I}\left[\max_{1 \le \ell \le K} t(g_{j,\ell}) \ge \tau\right], \qquad \text{EMT} = \frac{1}{N} \sum_{j=1}^{N} \max_{1 \le \ell \le K} t(g_{j,\ell}).$$

In multi-turn interactions, dialogue-level safety measurements from ToxicChat (Chen et al., 2023a) can be adopted. These measurements define risk over an entire conversation rather than an isolated response. Let $\mathcal{D}_{\text{dlg}}$ be a set of dialogues. For each dialogue $d$ in the set $\mathcal{D}_{\text{dlg}}$, let $R_d$ denote the set of all chatbot responses and let $q_{d,i}$ and $r_{d,i}$ represent the query-response pair at turn $i$. The Toxic Sentence Generation rate, referred to as TSG, can be computed as the percentage of conversations in which the chatbot generates at least one toxic response during the interaction (Chen et al., 2023a):

$$\text{TSG} = \frac{1}{|\mathcal{D}_{\text{dlg}}|} \sum_{d \in \mathcal{D}_{\text{dlg}}} \mathbb{I}\left[\max_{r \in R_d} t(r) \ge \tau\right].$$

To isolate cases where toxicity is elicited even with non-toxic user inputs, the non-toxic to Toxic rate, referred to as NT2T, can be reported. This metric represents the percentage of conversations in which the chatbot produces at least one toxic response at any point during the interaction while the corresponding user query is non-toxic (Chen et al., 2023a):

$$\text{NT2T} = \frac{1}{|\mathcal{D}_{\text{dlg}}|} \sum_{d \in \mathcal{D}_{\text{dlg}}} \mathbb{I}[\exists i : \ t(q_{d,i}) < \tau \ \wedge \ t(r_{d,i}) \ge \tau].$$

This dialogue-level NT2T definition is consistent with prior turn-level toxicity categorization of query-response pairs into NT2T, NT2NT, T2T, or T2NT. Within this classification, NT2T specifically denotes a non-toxic query that elicits a toxic response (Si et al., 2022).

Finally, to ensure safety interventions do not degrade linguistic quality, Perplexity and Distinct N can be reported (Liang et al., 2024; Jozefowicz et al., 2016; Li et al., 2016).

**Benchmark Integrity and Contamination.** Benchmark contamination is the risk that generated or training data overlap with evaluation benchmarks, inflating downstream scores and compromising evaluation integrity (Dong et al., 2024; Deng et al., 2024). This issue is particularly relevant to synthetic text pipelines, where benchmark instances may be copied, paraphrased, or stylistically imitated (Deng et al., 2024).

One practical metric is Test-set Slot Guessing (TS-Guessing) (Deng et al., 2024), which masks a crucial word or an answer option in a benchmark instance and asks the model to recover the missing content. Let $\mathcal{B}_{\text{mask}} = \{(x_i, z_i)\}_{i=1}^{N}$ denote a set of masked benchmark instances, where $x_i$ is the masked input and $z_i$ is the original hidden span or option. If $\hat{z}_i$ is the model prediction, this audit can be quantified by an exact match rate:

$$\text{EM}_{\text{TS}} = \frac{1}{N} \sum_{i=1}^{N} \mathbb{I}[\hat{z}_i = z_i].$$

A high $\text{EM}_{\text{TS}}$ suggests that the model can recover benchmark-specific content unusually well, which may indicate contamination or memorization rather than genuine generalization (Deng et al., 2024).

When access to a pretraining corpus is available, contamination can also be audited through retrieval-based overlap search (Deng et al., 2024). A simple survey-level abstraction is to report the fraction of benchmark items whose maximum retrieved similarity exceeds a threshold $\tau$:

$$\text{OR}_{\tau} = \frac{1}{N} \sum_{i=1}^{N} \mathbb{I}\left[\max_{d \in \mathcal{D}_{\text{pre}}} \text{sim}(b_i, d) \ge \tau\right].$$

For black-box settings where the training corpus is unavailable, contamination can be estimated via perplexity-based memorization analysis by comparing benchmark perplexity against memorized and clean reference

Table 2: Coverage of evaluation dimensions by representative text control, data-construction, and alignment methods, together with their method family and functional role. ✓: explicitly evaluated; △: partially or indirectly covered; ×: not reported or not applicable.

| Representative Work | Family | Role | Validity | Fidelity | Diversity | Faithfulness | Safety | Benchmark Contam. |
|---|---|---|---|---|---|---|---|---|
| FineWeb2 (Penedo et al., 2025) | Source Corpus Control and Composition | Data construction | × | △ | △ | × | × | × |
| Dolma (Soldaini et al., 2024) | Source Corpus Control and Composition | Data construction | × | △ | △ | × | ✓ | △ |
| RedPajama (Weber et al., 2024) | Source Corpus Control and Composition | Data construction | △ | △ | △ | × | △ | × |
| Self-Instruct (Wang et al., 2023b) | Prompt-Driven Generation and Refinement | Generation | △ | △ | △ | × | × | × |
| RARR (Gao et al., 2023a) | Inference-Time Steering and Verification | Evaluation | × | × | × | ✓ | × | × |
| PMI-DECODE (Nandwani et al., 2023) | Inference-Time Steering and Verification | Evaluation | × | ✓ | × | ✓ | × | × |
| ORPO (Hong et al., 2024) | Parameter-Efficient and Alignment-Based Control | Generation | × | △ | × | × | × | × |
| Deng et al. (Deng et al., 2024) | Source Corpus Control and Composition | Evaluation | × | × | × | × | × | ✓ |

baselines (Li, 2023). More generally, contamination can also be audited from model output distributions: Dong et al. (Dong et al., 2024) proposed contamination detection via output distribution and a corresponding contamination-mitigated evaluation strategy based on peakedness and duplication patterns in sampled outputs.

## 2.4 Evaluation Practice Gap

We analyze representative methods from Section 2.1 and categorize their reported evaluation protocols in Table 2.

**Validity.** Within the representative audit of eight works in Table 2, none explicitly reports validity using dedicated reproducible metrics, and two provide only partial coverage. Most studies overlook specific scores for grammatical acceptability or dialogue quality. Instead, they rely on downstream benchmarks or human judgments as general proxies for quality. However, these external signals often fail to identify specific validity errors. Even with improved models, large-scale automated pipelines can still suffer from basic structural breakdowns, including encoding noise, formatting violations, and truncation. Explicit validity monitoring therefore remains a practical necessity.

**Faithfulness.** Within Table 2, faithfulness is explicitly evaluated in only two of the eight audited methods, both evaluation-oriented works rather than general generation pipelines. This matters because faithfulness is central to controlling hallucination: as models improve in finding and using exact evidence, they become less likely to produce fabricated information. We therefore provide several metrics that can be used to measure faithfulness in these works.

**Safety.** Within our representative audit, safety is explicitly reported in one of the eight reviewed methods and partially covered in one additional work. Few studies adopt standardized protocols, such as Perspective API or Toxic List, to measure toxicity score or banned words. Consequently, future work on synthetic text should incorporate safety evaluation as a standard reporting component. By analyzing safety alongside quality indicators, researchers can better manage the trade-off between safety and utility. This approach is essential to prevent safety regressions even as performance improves.

**Benchmark Contamination.** Based on Table 2, benchmark contamination is explicitly evaluated in only one of the eight audited methods, with one additional work providing partial coverage. In our representative sample, such evaluation appears mainly in benchmark-integrity or corpus-control oriented studies rather than in general-purpose text generation pipelines. This gap is important because benchmark gains can otherwise become ambiguous: they may indicate better synthetic data, but they may also reflect overlap with public evaluation sets. For this reason, benchmark-aware contamination auditing should be more routinely reported when synthetic text is built from large public corpora or benchmark-adjacent sources.

## 2.5 Usage

We now shift from text and dialogue generation methods to the practical applications of the resulting synthetic corpora. We focus on offline uses, which refer to settings where model outputs are treated as static artifacts for training and evaluation. This approach is distinct from online prompting or control during the decoding process. Synthetic corpora are most effective when they are carefully curated and aligned with target deployment distributions. These corpora should also be explicitly stress tested according to the quality and trustworthiness dimensions of our taxonomy. We first discuss training-time uses such as data augmentation, alignment, and adaptation, and then we describe evaluation uses.

**Supervised Data Augmentation and Task Transfer.** A primary use case is to augment supervised training data in low-resource or zero-resource regimes. For machine reading comprehension, synthetic question answer pairs generated from unlabeled passages can substantially improve downstream accuracy when filtered rather than used wholesale. Early work showed that LLMs can produce synthetic QA corpora that, on their own, support competitive models on benchmarks such as SQuAD (Puri et al., 2020). Additionally, end-to-end synthetic QA generation can further enable domain adaptation when paired with filtering and validation (Shakeri et al., 2020).

More recent approaches integrate LLM-based rewards or selectors to identify high-value examples. For instance, Jin & Wang (2024) treats a generative LLM as a reward model to score synthetic QA pairs and retain only those predicted to be most useful for training. Survey evidence on data selection for instruction tuning suggests diminishing returns from scaling instruction corpora alone. This motivates data-efficient selection based on utility signals such as quality, diversity, difficulty, and complexity. Empirically, highly curated small sets such as LIMA with 1000 instances and large-scale human-AI collaboration corpora such as OpenAssistant conversations can be competitive. Furthermore, automated selectors improve efficiency through model-based filtering and strong LLM scoring using models such as GPT-4 to remove low-quality or incorrect examples and retain diverse and high-utility instances (Albalak et al., 2024; Köpf et al., 2023).

**Instruction Tuning and Alignment Corpora.** Beyond task labels, synthetic text and dialogue are widely used to construct instruction following and preference datasets for alignment. Large-scale instruction tuning corpora routinely mix human-written and LLM-written instructions, demonstrations, and task variants. The synthetic components expand coverage over skills, formats, and domains, while human labor focuses on seeding and spot checking. Preference and critique datasets such as UltraFeedback use a strong LLM like GPT-4 to provide multi-dimensional scores and natural language feedback on candidate responses. This yields large synthetic preference corpora for reward modeling and preference-based fine-tuning (Cui et al., 2024). In practice, synthetic instructions, responses, and AI-generated feedback are often combined into pipelines where models both propose and evaluate data. This process blurs the line between augmentation and alignment.

**Domain Adaptation and Retrieval Grounded Systems.** In domain-specific applications, synthetic corpora support both task adaptation and retrieval augmented generation. For question answering in specialized domains such as biomedical, practitioners often synthesize in-domain questions and answers from domain passages or corpora. They then filter low-quality generations to improve robustness under domain shift (Shakeri et al., 2020). These synthetic examples are then used to fine-tune general-purpose language models. This process simplifies earlier multi-stage synthetic question answering pipelines for domain adaptation (Shakeri et al., 2020). In retrieval grounded setups, model-generated query document or query identifier pairs can be used to train domain-specific retrievers. This approach improves ranking through hard

negative mining and preference learning on distributions that better match deployment scenarios (Wen et al., 2025). Because synthetic queries can be controlled through granularity and domain-specific constraints, they provide controllable testbeds for measuring retrieval faithfulness and downstream task gains.

**Synthetic Evaluation Sets and Judge Labeled Benchmarks.** Once models are trained or adapted, synthetic text and dialogue are increasingly treated as evaluation artifacts. Instead of relying solely on static human-written benchmarks, practitioners construct synthetic test suites that probe specific failure modes under controlled distributions. These modes include compositional reasoning, long context consistency, and safety constraints.

The LLM-as-a-judge pipeline can score large pools of model outputs using fine-grained criteria such as helpfulness, safety, and faithfulness. This approach supports the rapid construction of synthetic evaluation sets as well as public leaderboards and meta evaluation benchmarks such as MT-Bench and Chatbot Arena (Gu et al., 2025; Zheng et al., 2023). Self-verification methods such as Chain-of-Verification explicitly generate verification questions and independent checks to reduce hallucination (Dhuliawala et al., 2024). At the same time, sampling-based approaches like SelfCheckGPT produce alternative generations whose consistency can be logged as a black-box signal for hallucination and robustness (Manakul et al., 2023).

**Safety Testing and Privacy Preserving Surrogates.** Finally, synthetic corpora play a growing role in risk management. Privacy studies exploit LLM-generated variants of sensitive text as surrogates that retain statistical utility while reducing direct exposure of personally identifiable information. Memorization and extraction attacks demonstrate that models can regurgitate rare training examples, including personally identifiable information. This motivates careful deduplication and privacy-aware data handling (Carlini et al., 2021). SRD (Zhang et al., 2026) framework utilizes LLMs to generate toxic and good text and uses this contrast dataset to detoxify the model. For provenance and attribution, organizations increasingly embed watermarks or content credentials into synthetic corpora to support downstream detection and policy enforcement. Keyed statistical watermarks allow reliable statistical detection of model-generated text and are commonly evaluated under robustness and security considerations (Kirchenbauer et al., 2023; Coalition for Content Provenance and Authenticity, 2025). By framing fingerprint injection as a knowledge-editing problem over fingerprint text pairs, RFEdit enables efficient and robust injection with minimal impact on unrelated knowledge (Li et al., 2026). Synthetic safety and red teaming corpora are likewise used to stress test models and safety filters. These corpora consist of adversarial prompts, toxic continuations, or jailbreak dialogues generated by or with LLMs.

Overall, LLM-generated text and dialogue serve as flexible building blocks for training, adapting, evaluating, and governing language technologies. Their benefits are most pronounced when synthetic corpora are aligned with target tasks and distributions. Furthermore, these corpora should be coupled with strong selection and verification mechanisms and instrumented with provenance and privacy safeguards that make their quality and trust properties measurable.

## 3 Symbolic and Logical Data

Symbolic and logical data generation has gained significant attention with the rise of large reasoning models. These models emphasize multi-step inference and structured intermediate reasoning, often produced through chain-of-thought prompting (Wei et al., 2023; Kojima et al., 2023; Wang et al., 2023a; Liu et al., 2025b; Morishita et al., 2024; Toshniwal et al., 2024). Unlike general data augmentation, this field focuses on creating instances that are intrinsically checkable. Each example typically consists of a well-specified problem, a structured reasoning trace or intermediate artifact, and an explicit verification signal. These signals include exact keywords (Li et al., 2025d) or answer matching in mathematics word problems, such as tasks in the GSM8K dataset (Cobbe et al., 2021). They also include compilation and unit tests in code generation or logical validity under symbolic rules (Tafjord et al., 2021; Morishita et al., 2024; Liu et al., 2025b).

In practice, modern LLM-based pipelines do more than generate diverse problem–solution pairs. They also propose intermediate steps, critique and revise candidate solutions, and filter outputs using self-consistency (Wang et al., 2023a) or external tools (Ahmad et al., 2025). These pipelines help scale reasoning supervision

beyond fully human-written datasets. They produce large quantities of verified or partially verified trajectories for distillation and post-training. Recent examples of this approach include reasoning models trained with reinforcement learning or augmented by tools and verifiers (OpenAI et al., 2026; DeepSeek-AI, 2025).

## 3.1 Generation Methods

In this section, we introduce how to generate symbolic and logical data using LLMs.

**Heuristic Evolution Methods for Expanding Reasoning Tasks.** A primary category of reasoning data generation driven by LLMs relies on the iterative evolution of seed tasks. In this approach, strong models repeatedly rewrite problems to increase structural diversity and difficulty while attempting to preserve the validity of the solutions. This pattern originates from the rewriting techniques used in Evol-Instruct (Xu et al., 2024a) and has been adapted to domains such as mathematics and programming. For example, WizardMath applies a mathematics-specific evolution pipeline combined with reinforcement learning from evolution feedback to strengthen step-by-step reasoning (Luo et al., 2025b). Similarly, MetaMath scales up mathematics corpora by generating and diversifying data in the form of questions, answers, and rationales (Yu et al., 2024a).

In the field of programming, WizardCoder adapts instruction evolution to coding tasks by generating progressively harder instructions and responses from simpler seeds (Luo et al., 2025c). Across these studies, the core idea remains the same. The process starts from existing benchmarks or seed instructions and then applies controlled mutations such as paraphrasing, adding constraints, or making the problems more compositional. Finally, the process uses lightweight heuristics to maintain semantic accuracy. These heuristics often involve checking the agreement of answers, verifying basic formats, or screening for consistency. DyCodeEval extends this line of work to contamination-aware benchmarking by dynamically generating semantically equivalent variants from seed programming problems, enabling evaluation on reasoning tasks that are less vulnerable to benchmark leakage (Chen et al., 2025b).

**Tool-Verified Generation Methods with Solvers, Executors, and Provers.** A second approach combines synthesis from LLMs with programmatic verification. This method uses executors, compilers, unit tests, or symbolic checkers as reliable oracles. In mathematical reasoning, OpenMathInstruct-1 synthesizes large-scale problem and solution pairs by generating solutions in the style of a code interpreter. It retains instances where the computations are executable and consistent (Toshniwal et al., 2024).

In the field of programming, OpenCodeInstruct assembles large instruction tuning corpora with execution feedback and unit test signals. This strengthens acceptance criteria beyond surface plausibility (Ahmad et al., 2025). For formal logical reasoning, datasets and generators such as ProofWriter and ALT-FLD×2 construct instances where the validity is grounded in formal rules. This enables the deterministic checking of entailment or proof steps under symbolic constraints (Tafjord et al., 2021; Morishita et al., 2024). SynLogic further advances scalable and verifiable logic synthesis by generating diverse logical tasks whose correctness can be verified by simple rule-based checkers (Liu et al., 2025b). In all these cases, the models propose candidates while external tools act as gatekeepers that filter for correctness and logical structure. Benchmark-aware overlap checks can further strengthen this gatekeeping process by filtering out candidate solutions that are too similar to benchmark gold solutions or training corpora (Riddell et al., 2024).

**Trajectory Harvesting via Rewarded Rollouts and Distillation.** A third category of methods generates reasoning data by collecting trajectories from strong teacher models or reasoners trained through reinforcement learning. These models interact with tasks that allow for verification. In this context, trajectories serve as more than just explanations. They are the results of a policy exploring solution spaces guided by reward and verification signals. These trajectories are then distilled into student models.

Recent reasoning model pipelines follow this approach by collecting large volumes of partially verified rollouts. These rollouts are filtered using automated verifiers and learned judges and are then used as signals for downstream post-training. Similarly, public releases such as SYNTHETIC-2 provide large-scale datasets that are the results of reasoning agents working with verifiers across various task families (Prime Intellect Team, 2025). This approach produces verified reasoning traces that are suitable for distillation. The key

difference in this method lies in the nature of the data. The data are policy traces generated during search and optimization processes instead of static and one-shot synthetic question and answer pairs.

**Preference-Curated Generation via LLM as a Judge.** Finally, in scenarios where rigid programmatic checks are incomplete or unavailable, many pipelines rely on preference curation and supervision using LLMs-as-judge mechanisms. UltraFeedback constructs large-scale preference signals by sampling multiple candidate responses and scoring them using a superior model. These scores are based on dimensions such as helpfulness, truthfulness, and instruction adherence (Cui et al., 2024).

CodeUltraFeedback applies a similar concept to the programming domain. In this context, judges assess candidate solutions based on coding preferences such as readability and adherence to instructions (Weyssow et al., 2024). Related self-training approaches, such as STaR, create rationales through generate and filter loops where correctness serves as a general acceptance signal (Zelikman et al., 2022). Collectively, these methods emphasize model-driven evaluation. Reasoning data are generated at scale and then organized by judge models into ranked or preference-annotated corpora. This process supports alignment, preference optimization, and self-improvement.

### 3.2 Quality Metrics for Symbolic and Reasoning Data

The fundamental criterion for evaluating the quality of reasoning data generated by LLMs is objective correctness. This requires that both intermediate reasoning steps and final conclusions serve as logically valid and empirically verifiable results. Given that reasoning data encompass mathematical, symbolic, programmatic, and open-domain contexts, the specific verification protocols vary by domain. However, these protocols share a unified objective. They validate that generated reasoning traces adhere strictly to ground truth logic or executable truth conditions.

**Validity.** For mathematical and code based reasoning, correctness can often be validated algorithmically through domain verifiers. Let $\mathcal{R} = \{(q_i, c_i, y_i)\}_{i=1}^{N}$ denote reasoning examples, where $q_i$ is the problem, $c_i$ is the generated rationale such as a chain-of-thought or a code interpreter trace, and $y_i$ is the final answer. When a reference answer $y_i^{\star}$ or an executable specification is available, we define a domain-specific checker $f_{\text{check}}$ as follows:

$$f_{\text{check}}(q_i, c_i, y_i) = \begin{cases} 1, & \text{if the candidate is verified as correct for } q_i; \\ 0, & \text{otherwise.} \end{cases}$$

The overall verification accuracy is then reported as

$$\text{Acc}_{\text{verify}} = \frac{1}{N} \sum_{i=1}^{N} f_{\text{check}}(q_i, c_i, y_i).$$

In MetaMath, the answer augmentation stage uses rejection sampling. In this process, diverse reasoning paths are generated, and only those yielding correct answers are retained. This method ensures validity through answer-level verification (Yu et al., 2024a).

In OpenMathInstruct-1, solutions are represented in a code interpreter style format. Correctness is enforced by keeping only solutions that lead to the ground truth answer. The paper also uses training set coverage measured as pass at k. This metric identifies whether any of k sampled solutions reaches the ground truth (Toshniwal et al., 2024).

For program synthesis and code reasoning, validity is commonly measured by unit test execution. Let $\mathcal{T}_i$ be the test suite for example $i$. The unit test pass rate for a candidate solution is defined as follows:

$$\text{PassRate}_i = \frac{1}{|\mathcal{T}_i|} \sum_{t \in \mathcal{T}_i} \mathbb{I}\big(\text{exec}(y_i, t) = \texttt{pass}\big),$$

The dataset-level pass rate is calculated as $\text{PassRate} = \frac{1}{N}\sum_{i=1}^{N}\text{PassRate}_i$. This approach aligns with OpenCodeInstruct, which executes solutions on generated unit tests and records pass rates as metadata (Ahmad et al., 2025).

For logical and deductive reasoning with explicit proofs such as proof graphs, validity requires correct entailment prediction and a correct proof. Following ProofWriter, proof correctness is evaluated using a strict metric called Full Accuracy. In this metric, the predicted proof graph must exactly match a gold proof. If the graphs do not match, the proof scores zero (Tafjord et al., 2021).

Accordingly, let $y_i^\star$ be the gold entailment or answer label and let $c_i^\star$ be a gold proof representation. The strict proof accuracy can be summarized as

$$\text{Acc}_{\text{proof}} = \frac{1}{N}\sum_{i=1}^{N}\mathbb{I}(y_i = y_i^\star)\cdot\mathbb{I}(c_i = c_i^\star).$$

Finally, for robustness style validity checks, FAIRR (Sanyal et al., 2022) defines consistency over an equivalence set of perturbed inputs. For a theory and statement pair $(T, s)$ and its equivalence set $E(T, s) = \{(T_k', s_k')\}_{k=1}^{K}$, consistency is defined by the following equation:

$$C(T, s) = \frac{1}{K}\sum_{k=1}^{K}\mathbb{I}\big(f(T, s) = f(T_k', s_k')\big),$$

This value is averaged over the dataset. FAIRR reports entailment accuracy, strict proof accuracy, and consistency (Sanyal et al., 2022).

**Fidelity.** Fidelity measures the alignment between proxy evaluation signals and actual quality standards. Unlike validity which assesses the objective correctness of a solution against a ground truth, fidelity evaluates the reliability of the tools or models used to judge the data. This metric ensures that automated evaluations reflect true human reasoning or logical consistency when direct verification is not possible.

Following the self-consistency decoding paradigm, multiple reasoning paths are sampled, and the most consistent answer is selected (Wang et al., 2023a). For each question $q_i$, $K$ chains $\{c_{i,k}\}_{k=1}^{K}$ are sampled, and their final answers $\{y_{i,k}\}_{k=1}^{K}$ are extracted. The majority-vote answer is defined as follows:

$$\hat{y}_i = \arg\max_{y}\sum_{k=1}^{K}\mathbb{I}\big(y_{i,k} = y\big).$$

To measure the alignment between the consensus proxy and the reference label, the self-consistency accuracy is computed as:

$$\text{Acc}_{\text{SC}} = \frac{1}{N}\sum_{i=1}^{N}\mathbb{I}\big(\hat{y}_i = y_i\big).$$

To further quantify the internal consistency of sampled answers for the same question, the average pairwise answer agreement is used:

$$\text{Agree}(q_i) = \frac{1}{K(K-1)}\sum_{k\neq k'}\mathbb{I}\big(y_{i,k} = y_{i,k'}\big).$$

For open-ended natural language reasoning where executable checking is often not possible, judges based on LLMs provide automated scoring signals. For example, GPT-4 is frequently used to provide these evaluations.

To evaluate how faithfully these judge signals reflect human judgments, let $u_i^{(\text{LLM})}$ and $u_i^{(\text{human})}$ denote scalar scores from a model judge and a human annotator for the $i$-th example.

The rank association between these scores is measured using Spearman $\rho$ or Pearson correlation. This follows established evaluation methods for alignment in model judges (Lai et al., 2025):

$$\rho_{\text{LLM-human}} = \text{corr}_{\text{Spearman}}\big(u^{(\text{LLM})}, u^{(\text{human})}\big).$$

When the judge outputs $M$ rubric dimensions such as coherence, faithfulness, and factuality, we can aggregate them into an overall proxy quality score:

$$Q_{\text{reason}} = \frac{1}{M} \sum_{m=1}^{M} \text{score}_{i,m}.$$

The reliability of model judge signals is commonly validated on benchmarks such as JudgeLM and MT Bench or the Chatbot Arena. These benchmarks report the agreement between model judges and human preferences (Zhu et al., 2025a; Zheng et al., 2023).

Preference datasets based on AI feedback such as UltraFeedback (Cui et al., 2024) and pipelines using reinforcement learning from AI feedback provide training signals for preference modeling and downstream alignment. These pipelines replace expensive human labels with preferences generated by LLMs (Lee et al., 2024). On pairs labeled with preferences, a reward model $R_\phi$ can be evaluated by whether it ranks the preferred output higher.

We consider a pair $(c_{i,a}, c_{i,b})$ with a binary preference label $z_i$ that is either zero or one.

For example, $z_i$ equals one if $c_{i,a}$ is preferred. The label implied by the model can be written as:

$$\hat{z}_i = \mathbb{I}\big(R_\phi(c_{i,a}) > R_\phi(c_{i,b})\big).$$

Accordingly, the pairwise preference accuracy is given by:

$$\text{Acc}_{\text{RM}} = \frac{1}{N} \sum_{i=1}^{N} \mathbb{I}\big(\hat{z}_i = z_i\big).$$

This metric evaluates reward models on pairwise preference data used in training and benchmarking for reinforcement learning from human feedback (Bai et al., 2022; Frick et al., 2024).

**Diversity.** Diversity measures whether a reasoning dataset captures a broad set of distinct valid solution strategies rather than repeatedly sampling near-identical reasoning traces. This property is particularly important for symbolic and multi-step reasoning tasks, where a single problem may admit multiple correct derivations, decompositions, or proof structures. Prior work on chain-of-thought reasoning demonstrates that sampling multiple reasoning paths for the same problem improves both accuracy and robustness, highlighting the importance of diverse reasoning trajectories (Wang et al., 2023a).

Let $\mathcal{R}_i = \{c_{i,1}, \ldots, c_{i,K}\}$ denote $K$ generated reasoning trajectories for problem $q_i$. A simple diversity signal arises from the distribution over final answers. Following self-consistency, empirical answer distribution $p_i(y)$ is defined and its entropy is computed as:

$$H(q_i) = -\sum_y p_i(y) \log p_i(y),$$

where $p_i(y) = \frac{1}{K} \sum_{k=1}^{K} \mathbb{I}(y_{i,k} = y)$. Higher entropy indicates broader exploration of the solution space. However, answer-level diversity is insufficient for reasoning tasks, as multiple trajectories may share the same final answer while differing substantially in intermediate steps.

To capture reasoning-level diversity, pairwise dissimilarity between trajectories is:

$$\text{Div}(q_i) = \frac{1}{K(K-1)} \sum_{a \neq b} \big(1 - \kappa(c_{i,a}, c_{i,b})\big),$$

where $\kappa$ is a similarity function between reasoning traces. This formulation aligns with recent work showing that diversity among reasoning paths, rather than answer diversity alone, is critical for improving reasoning performance and generalization (Fu et al., 2023; Wang et al., 2023a).

The choice of $\kappa$ depends on the structure of the reasoning data. For natural language chains of thought, semantic similarity can be computed via embedding-based representations or step-level alignment. For symbolic reasoning such as programs or proofs, structural representations are more appropriate. Recent work on program and mathematical reasoning emphasizes the importance of capturing structural variation beyond surface-level differences, as distinct symbolic decompositions correspond to different reasoning strategies (Toshniwal et al., 2024; Ahmad et al., 2025).

Beyond pairwise metrics, recent studies on synthetic data generation highlight that diversity directly impacts generalization. For example, large-scale synthetic reasoning pipelines show that increasing the diversity of generated solutions—through multiple sampling, filtering, and augmentation—improves performance on both in-distribution and out-of-distribution tasks (Yu et al., 2024a; Toshniwal et al., 2024). Similarly, scaling reasoning data via diverse chain-of-thought generation has been shown to be a key factor in improving reasoning capabilities (Fu et al., 2023).

### 3.3 Trustworthy Metrics for Symbolic and Reasoning Data

Even when a generated reasoning trace yields a correct final answer, the logical coherence and faithfulness of intermediate steps are critical. A model that reaches the right answer without valid reasoning introduces spurious patterns and unreliable supervision.

**Faithfulness.** Let $c_i = [c_{i,1}, \ldots, c_{i,T_i}]$ be the chain-of-thought for example $i$. A step verifier $f_{\text{step}}$ checks each step. This verifier could be a symbolic executor in formal domains or an inference based entailment checker for natural language. It evaluates each step as follows:

$$f_{\text{exec}}(c_{i,t}) = \begin{cases} 1, & \text{if } c_{i,t} \text{ is a valid transformation,} \\ 0, & \text{otherwise.} \end{cases}$$

The step validity rate is defined by the following equation:

$$\text{Val}_{\text{step}} = \frac{1}{\sum_i T_i} \sum_i \sum_{t=1}^{T_i} f_{\text{exec}}(c_{i,t}).$$

This metric measures how often intermediate steps obey mathematical, logical, or semantic rules. Correctness metrics at the step level have been explored in reasoning evaluation frameworks such as ReCEval. This framework scores the quality of steps and chains through textual entailment and informativeness (Prasad et al., 2023). Furthermore, causal faithfulness analyses such as FRODO test whether the content of a chain-of-thought has a causal influence on the final answer rather than simply correlating with it (Paul et al., 2024).

Besides executable step verification, entailment models can also serve as a proxy for step support in natural language. For each pair of adjacent steps $c_{i,t-1}$ and $c_{i,t}$, an entailment scorer $g_{\text{entail}}$ outputs:

$$p_{i,t} = g_{\text{entail}}(c_{i,t-1} \Rightarrow c_{i,t}),$$

The average entailment alignment is defined to represent the support within the chain:

$$\text{Align}_{\text{entail}} = \frac{1}{\sum_i (T_i - 1)} \sum_i \sum_{t=2}^{T_i} p_{i,t}.$$

Higher values indicate that the reasoning progresses through steps supported by semantics and logic instead of abrupt or contradictory jumps. Evaluation methods using entailment have been used to assess the correctness and informativeness of reasoning chains (Prasad et al., 2023). These methods have also been applied to assign partial credit based on textual entailment (Yao & Barbosa, 2024).

**Benchmark Integrity and Contamination.** For symbolic and reasoning data, benchmark contamination arises when benchmark problems or reference solutions overlap with training corpora, allowing models to exploit memorized solution patterns rather than genuine reasoning (Riddell et al., 2024; Matton et al., 2024; Jain et al., 2024). This issue is especially salient when benchmarks are publicly available and can be matched at both lexical and structural levels (Riddell et al., 2024).

Following overlap-based contamination audits in code benchmarks (Riddell et al., 2024), let $S_i^{\text{agg}}$ denote the aggregated overlap score between benchmark item $i$ and retrieved training instances. A simple survey-level summary is the contamination rate:

$$\text{CR}_\tau = \frac{1}{N} \sum_{i=1}^{N} \mathbb{I}[S_i^{\text{agg}} \geq \tau].$$

A higher $\text{CR}_\tau$ indicates that a larger fraction of benchmark items may be contaminated by overlapping training evidence.

For dynamic benchmarks, contamination can also be assessed through time-segmented evaluation (Jain et al., 2024). Let $\mathcal{B}_{\text{post}}(t_c)$ denote benchmark problems released after a model's cutoff date $t_c$. A corresponding cutoff gap can be written as

$$\Delta_{\text{cutoff}} = \text{Perf}(\mathcal{B}_{\text{all}}) - \text{Perf}(\mathcal{B}_{\text{post}}(t_c)).$$

A larger $\Delta_{\text{cutoff}}$ suggests that benchmark performance is substantially inflated by pre-cutoff contamination.

**Robustness.** To study generalization under a shift in distribution, we compare the pairwise accuracy of reward models on in-domain evaluations and out-of-domain evaluations. This difference is defined as follows:

$$\Delta_{\text{OOD}} = \text{Acc}_{\text{RM}}^{\text{in}} - \text{Acc}_{\text{RM}}^{\text{out}}.$$

Recent evaluations report these comparisons by measuring in-domain accuracy on preference data such as UltraFeedback. They also measure out-of-domain accuracy on benchmarks such as RewardBench (Mahan et al., 2024; Lambert et al., 2024). A smaller difference indicates more stable preference ranking across different domains.

### 3.4 Evaluation Practice Gap

Table 3 suggests a clear imbalance in our representative audit of symbolic and logical data generation. Among the seven audited works, answer-level validity is explicitly reported in five, whereas faithfulness and benchmark contamination are explicit in only one, and are partial in one, respectively. Furthermore, robustness is only partially covered in two works. We interpret this as a pattern within the audited sample rather than a field-wide prevalence estimate.

**Faithfulness.** Within the representative audit, explicit faithfulness evaluation is concentrated in benchmark-style works that contain structured proof annotations. In these benchmarks, strict proof correctness can be directly measured. For example, ProofWriter reports full proof accuracy based on exact proof graph matching (Tafjord et al., 2021). Similarly, FaiRR includes strict proof accuracy as a core part of its protocol

Table 3: Whether representative symbolic and logical data generation methods explicitly evaluate each dimension in their experimental sections, together with their method family and functional role. ✓: explicitly evaluated; △: partially or indirectly covered; ×: not reported or not applicable.

| Representative Work | Family | Role | Validity | Fidelity | Diversity | Robustness | Faithfulness | Benchmark Contam. |
|---|---|---|---|---|---|---|---|---|
| MetaMath (Yu et al., 2024a) | Heuristic Evolution | Generation | ✓ | × | △ | × | × | × |
| OpenMathInstruct-1 (Toshniwal et al., 2024) | Tool-Verified Generation | Generation | ✓ | ✓ | △ | × | × | × |
| OpenCodeInstruct (Ahmad et al., 2025) | Tool-Verified Generation | Generation | ✓ | △ | × | × | × | × |
| UltraFeedback (Cui et al., 2024) | Preference-Curated LLM-Judge Data | Benchmark | × | ✓ | × | × | × | △ |
| ProofWriter (Tafjord et al., 2021) | Tool-Verified Generation | Benchmark | ✓ | × | × | △ | ✓ | × |
| FaiRR (Sanyal et al., 2022) | Tool-Verified Generation | Evaluation | ✓ | × | × | △ | △ | × |
| DyCodeEval (Chen et al., 2025b) | Heuristic Evolution | Benchmark | × | × | × | × | × | ✓ |

(Sanyal et al., 2022). In contrast, many generation pipelines that rely on answer verification or filtering based on execution prioritize the correctness of the final answer. Notable examples include MetaMath (Yu et al., 2024a), OpenMathInstruct-1 (Toshniwal et al., 2024), and OpenCodeInstruct (Ahmad et al., 2025). As a result, the validity of intermediate reasoning steps and the question of whether they truly support the conclusion often remain under-evaluated.

**Robustness.** Robustness is reported in fewer studies within Table 3. When researchers address this property, it appears in different forms that are not directly comparable. For instance, ProofWriter evaluates out of domain transfer across rule sets (Tafjord et al., 2021). In another case, FaiRR reports consistency based on perturbations alongside entailment and accuracy (Sanyal et al., 2022). Beyond these specific examples, most research related to generation focuses mainly on performance within the same domain. These studies rarely treat changes in data distribution or stability under perturbations as their main evaluation goals. This lack of standardization shows that there is a need to define specific robustness metrics that are suitable for symbolic and logical generation. Researchers should avoid relying only on general proxies for robustness.

**Benchmark Contamination.** Within the representative audit, benchmark contamination is explicitly evaluated in only one work and partially covered in one additional work. Existing coverage is mostly limited to studies that explicitly consider benchmark leakage, while standard reasoning data generation pipelines rarely report contamination audits as a routine evaluation step. As a result, many studies continue to emphasize correctness, execution success, or preference quality, while leaving overlap with public benchmarks or reference solutions under-examined. This gap matters because benchmark gains in reasoning tasks may otherwise reflect memorized solution patterns rather than genuine reasoning ability, suggesting that contamination audits should be more routinely included in standard evaluation practice.

## 3.5 Usage

**Pretraining and Continual Pretraining with Synthetic Corpora.** A primary application of synthetic reasoning data involves integrating reasoning trajectories directly into the foundation model stage. In this paradigm, synthetic corpora are integrated during pretraining or continual pretraining to provide models with structured and multiple-step reasoning capabilities prior to instruction tuning (Yang et al., 2024a; Ying et al., 2024; Luo et al., 2025a; Xie et al., 2025). This approach views reasoning as a fundamental skill. In this context, symbolic, arithmetic, and procedural inference patterns are acquired alongside natural language. This process helps the internal representations of the model move toward compositional reasoning.

Recent advancements in large-scale reasoning models such as DeepSeek R1 and OpenAI o1 confirm the effectiveness of optimization after pretraining. These frameworks build upon pretrained base models and use large-scale reinforcement learning to produce long-horizon and step-wise problem-solving behaviors. This process creates models specialized in reasoning instead of relying on superficial post-hoc fine-tuning (DeepSeek-AI, 2025; OpenAI et al., 2026). Specifically, DeepSeek R1 uses rule-based and verifiable outcome rewards to provide precise feedback for domains such as mathematics and coding. This enables scalable improvements through automatic correctness signals (DeepSeek-AI, 2025). Similarly, OpenAI o1 emphasizes large-scale reinforcement learning to enhance careful reasoning and the effective use of chain-of-thought methods. This achieves strong performance across benchmarks in mathematics, coding, and science (OpenAI et al., 2026).

In parallel, families specialized in mathematics, such as Qwen Math and InternLM Math adopt a strategy of continued pretraining derived from strong base models. These methods incorporate math-focused corpora followed by downstream post-training, e.g., supervised fine-tuning with chain-of-thought data and tool-augmented or verifier-guided supervision, as well as reward-model-based reranking or filtering of multi-step solutions. Together, this continued-pretraining post-training pipeline improves mathematical reasoning through specialization (Yang et al., 2024a; Ying et al., 2024). Furthermore, related data pipelines synthesize new problems and solutions at scale while using screening based on correctness to retain only solutions where the final answer matches a ground truth. This methodology is shown by MetaMathQA and OpenMathInstruct-1. These systems use strong LLMs to generate candidate solutions while using verification based on execution to validate reliability (Yu et al., 2024a; Toshniwal et al., 2024). Early pipelines often used closed source models while later work uses increasingly strong open models. Collectively, these efforts treat synthetic reasoning corpora as primary components of the training distribution. By exposing the underlying language model to step-wise mathematical and logical derivations, these methods improve data efficiency during later specialization stages.

**Supervised Fine-Tuning on Verified Reasoning Traces.** A distinct application of reasoning data generated by LLMs involves explicitly providing chain-of-thought capabilities through supervised fine-tuning. This process uses verified reasoning triplets consisting of an input, a reasoning trace, and a final answer. Unlike pretraining which primarily shapes inductive biases, supervised fine-tuning aligns model outputs with concrete multiple step examples. This encourages the model to internalize intermediate problem solving procedures.

In the domain of mathematical reasoning, synthetic datasets fine-tune open models using explicit verification signals although the specific mechanisms vary. MetaMathQA scales data production through bootstrapped question generation and uses rejection sampling to filter reasoning traces based on the correctness of the final answer (Yu et al., 2024a). OpenMathInstruct-1 emphasizes verification based on execution by allowing solutions to combine natural language reasoning with executable code. This method uses execution by an interpreter as a validation signal (Toshniwal et al., 2024). Similarly, releases such as the synthetic GSM8K dataset by Gretel include structured reflection style instances paired with automated validation. Other variants incorporate rigorous automated checks including model judge evaluations and symbolic verification through tools such as SymPy to screen and refine candidates (AI, 2024). WizardMath extends beyond conventional supervised fine-tuning by synthesizing complex instructions and applying reinforcement learning from evolved instructions feedback to strengthen reasoning behaviors beyond simple imitation (Luo et al., 2025b).

In the programming domain, corpora such as CodeAlpaca and Magicoder provide instruction and solution pairs for code generation. Magicoder further anchors synthesis in open source code snippets to better reflect realistic development scenarios (Chaudhary, 2023; Wei et al., 2024). OpenCodeInstruct improves this paradigm with explicit test suites and execution outcomes, along with quality assessments based on models. This provides verifiable signals that support data curation during training with synthetic programs (Ahmad et al., 2025).

Regarding logical and symbolic reasoning, frameworks such as ALT construct principled synthetic logic corpora. These corpora consist of multiple-step deductive instances generated by programs and grounded in formal logic. This data facilitates supplementary training to improve entailment and inference patterns. In

parallel, SynLogic synthesizes diverse logic tasks paired with rule-based verifiers to enable scalable training with verifiable feedback (Morishita et al., 2024; Liu et al., 2025b).

In summary, supervised fine-tuning on verified reasoning traces offers a controllable method for using synthetic data. Generation pipelines propose candidate problems and explanations while external verifiers provide correctness signals. The resulting validated instances are then used to train models that are capable of explaining and justifying their reasoning step by step.

**Optimization via Preference Modeling and Reinforcement Learning.** A significant application of synthetic reasoning data lies in providing evaluative feedback rather than direct supervision. This process encourages robust reasoning behaviors through preference modeling and reinforcement learning. In this context, reasoning traces generated by LLMs or evaluated by verifiers function as inputs to reward models or policy optimization objectives. These objectives explicitly prioritize logical consistency, correctness, and clarity within multiple step inference.

For instance, UltraFeedback constructs multiple domain preference datasets by sampling candidate responses from diverse models and employing a superior LLM judge to rank them according to specific criteria. These criteria include instruction adherence and truthfulness. This process yields large-scale AI feedback signals suitable for reward model training and optimization styles similar to reinforcement learning from human feedback (Cui et al., 2024). Approaches such as reinforcement learning from AI feedback extend this methodology by substituting human preference labels with comparisons judged by LLMs. This substantially enhances the scalability of preference data collection across tasks and domains (Lee et al., 2024).

Within the reasoning domain, contemporary systems increasingly integrate reinforcement learning with programmatically verifiable feedback for mathematical, coding, and logical problems. Specifically, DeepSeek R1 uses rule-based rewards and executable feedback to score reasoning trajectories during training. This feedback includes actions such as compiling and executing code to enable policy updates that reinforce verifiable correctness (DeepSeek-AI, 2025). OpenAI o1 incorporates large-scale reinforcement learning to refine multiple step reasoning behaviors, although the specific implementation details of the reward and verifier mechanisms remain undisclosed (OpenAI et al., 2026). Furthermore, the SYNTHETIC-2 corpus provides millions of reasoning traces including reinforcement learning rollouts accompanied by reward signals. These signals facilitate downstream distillation and offline reinforcement learning research (Prime Intellect Team, 2025). Consequently, reasoning data evolve from static training examples into dynamic evaluative signals. Synthetic traces combined with judge or verifier outputs define a reward landscape that guides iterative model refinement.

**Knowledge Distillation and Implicit Reasoning Transfer.** A complementary application of reasoning data generated by LLMs involves the compression of explicit reasoning into implicit internal representations. This process distills detailed chain-of-thought traces from complex teacher models into compact student models. These student models are capable of performing robust reasoning without generating extensive explanations during inference. In this paradigm, reasoning traces function as latent supervision that guides representation learning rather than serving only as textual targets.

Methods for implicit chain-of-thought distillation, such as those proposed by Deng et al. (2023), train a teacher model using explicit supervision. They subsequently distill its reasoning competence into a student model through hidden state learning signals. This enables the creation of efficient models that reason implicitly without producing lengthy traces. In symbolic and deductive contexts, ProofWriter provides structured supervision over natural language proofs. This approach offers process-level signals that support the training and potential distillation of lightweight models capable of following proof-like reasoning patterns with minimal explicit rationales (Tafjord et al., 2021).

More broadly, large-scale reasoners trained with reinforcement learning and trajectory corpora such as DeepSeek R1 and SYNTHETIC-2 provide rich multiple-step rollouts that serve as distillation targets. These resources enable compact models to inherit high-level reasoning behaviors from computationally intensive teacher models (DeepSeek-AI, 2025; Prime Intellect Team, 2025). Therefore, synthetic reasoning data facilitates not only explicit chain-of-thought generation but also implicit reasoning transfer. The goal is to

embed process-level competence into models that yield correct answers without necessarily explaining every intermediate step.

**Evaluation, Benchmarking, and Generalization Testing.** Finally, synthetic and programmatically structured reasoning corpora increasingly serve as evaluation instruments. These corpora offer diverse and automatically verifiable tasks to test generalization capabilities beyond static benchmarks curated by humans. The primary objective is the construction of dynamic benchmarks that evolve alongside model capabilities while maintaining rigorous and verifiable scoring protocols.

In the domain of mathematics, GSM-HARD extends the GSM8K dataset by systematically substituting numerical values with larger or more complex alternatives. This process creates a stress-test suite generated by programs to evaluate arithmetic robustness. When integrated with programmatic execution solvers, model outputs can be verified automatically at scale (Gao et al., 2023b). Although not generated by LLMs, SciBench aggregates scientific problems at the college level in physics, chemistry, and mathematics. This enables systematic evaluation and detailed error analysis of scientific problem-solving skills (Wang et al., 2024b).

In the fields of causal and logical reasoning, CREPE introduces a benchmark for assessing causal reasoning regarding event plausibility and entity states. This benchmark uses human judgments to identify discrepancies between model behavior and human reasoning (Zhang et al., 2023). Additionally, SynLogic provides held-out validation splits for synthetic logical tasks. It reports performance using variance-reducing metrics such as the average at eight (Liu et al., 2025b). Collectively, these evaluation frameworks demonstrate the dual utility of synthetic and programmatically structured reasoning datasets. They not only provide training and reinforcement learning signals but also support benchmarks for monitoring reasoning robustness, distribution shifts, and cross-domain generalization as models advance.

## 4 Tabular Data

Tabular data consist of structured records organized according to a fixed schema of heterogeneous features that include rows and columns. As described in the survey by Shi et al. (2025b), the generative modeling of this data requires the joint representation of mixed type attributes including both numerical and categorical domains as well as complex dependencies between columns. At the same time, the model must preserve statistical fidelity to the original source distribution. Validity in this context is defined by a data point representing a row vector that follows schema level constraints and intrinsic functional dependencies. Failure to enforce these constraints results in synthetic tables that might appear statistically plausible but actually violate domain logic or structural rules (Shi et al., 2025b; Xu et al., 2025). In applied settings such as healthcare, Barr et al. (2025) demonstrated the effectiveness of zero-shot prompting with LLMs to produce clinically plausible perioperative datasets. They validated the fidelity of this data through statistical comparison with real world reference data. More broadly, the field treats synthetic tabular data generation as an optimization problem that balances data utility and fidelity against privacy preservation. Consequently, evaluation protocols are organized around these two dimensions of data quality and privacy protection (Shi et al., 2025b).

### 4.1 Generation Methods

**Prompt-Based and In-Context Tabular Synthesis** A prominent category of methods leverages prompt engineering or in context learning to synthesize tabular records without requiring full model fine-tuning. These methods typically prioritize schema validity and coverage of long tail distributions. Within this paradigm, CLLM addresses low data regimes by utilizing LLM priors and introducing a curation pipeline based on learning dynamics to filter synthetic rows using confidence and uncertainty metrics (Seedat et al., 2024). A distinct subset of approaches explicitly steers generation toward minority or under-represented data segments. For instance, EPIC and LITO employ class conditioned or group conditioned prompting strategies including self authentication mechanisms to bias sampling toward rare classes, enhancing both schema compliance and the representativeness of the long tail (Kim et al., 2025; Yang et al., 2024b). Beyond class imbalance and subgroup coverage, prompt-based synthesis has also begun to incorporate fairness-aware control. FairCauseSyn extends this paradigm by combining schema-guided prompting, in context learning, and self-consistency to generate synthetic tabular data under causal fairness constraints. Rather than treating

fairness only as a post hoc diagnostic, it integrates fairness considerations directly into the generation pipeline and evaluates the resulting data through causal effect based criteria (Nagesh et al., 2025). In a complementary approach, TabGen-ICL improves fidelity through the adaptive selection of in context exemplars. It iteratively retrieves real samples that represent the residual between generated and authentic distributions to refine the exemplar pool for a fixed base model (Fang et al., 2025). Beyond direct row synthesis, Nam et al. (2024) incorporate decision tree feedback to guide LLMs in generating new features. This extends prompting methodologies into the domain of feature engineering. Despite these advancements, empirical analyses indicate that unconstrained row wise prompting and arbitrary feature ordering often induce violations of functional dependencies. Furthermore, standard univariate or correlation-based metrics frequently fail to detect these constraint failures. This necessitates the development of strictly schema-aware and constraint-aware prompting and evaluation strategies (Xu et al., 2025).

**Fine-Tuning and Specialized Tabular Generation** A distinct category of methodologies involves fine-tuning LLMs specifically for table synthesis. These approaches aim to model structural constraints and distributional dependencies more accurately to enhance both validity and fidelity. For example, GReaT fine-tunes an autoregressive language model on serialized tabular rows to enable fully conditional sampling across arbitrary feature subsets. This mechanism improves sample realism and reduces the occurrence of implausible cross feature combinations (Borisov et al., 2023). Extending this architecture, Nguyen et al. (2024) introduce permutation based training combined with feature conditional sampling to preserve feature label correlations and further enhance sample realism. REaLTabFormer targets relational tabular synthesis by generating a parent table autoregressively and conditioning child table generation on the sampled keys (Solatorio & Dupriez, 2023). To address memorization risks and capture inter row structural nuances, HARMONIC incorporates $k$-nearest neighbor-based instructional signals that emphasize neighborhood relations across records (Wang et al., 2024d). Other frameworks integrate explicit self-correction mechanisms. Table-LLM-Specialist proposes an iterative generator validator self-training paradigm where candidate supervision is generated by models and validated via table-specific consistency signals such as permutation and execution invariance. This approach facilitates specialist fine-tuning without reliance on manual annotations (Xing et al., 2024). Furthermore, adaptive methods such as TableDreamer progressively synthesize instances that expose model weaknesses by targeting observed failure modes to improve data efficiency and downstream utility (Zheng et al., 2025). Beyond improving fidelity and utility, recent fine-tuning based methods have also started to explicitly regulate fairness sensitive distributions. Towards Universal Debiasing for Language Models-based Tabular Data Generation introduces debiasing objectives such as UDF-DPO and UDF-MIX to adapt LLM-based tabular generators while preserving downstream utility. This extends specialized tabular generation from structural and distributional modeling toward training time fairness control (Li et al., 2025b). Collectively, these approaches address challenges identified in recent empirical analyses by incorporating structural awareness, adaptive correction, and increasingly fairness-aware control during both training and sampling phases (Xu et al., 2025).

**Hybrid Architectures and Structured Synthesis** Hybrid architectures integrate LLMs with external structure or distribution aware components. Alternatively, they reformulate table synthesis as structured question answering to jointly address gaps in validity, fidelity, and utility. In these designs, LLMs typically function as engines for schema reasoning, constraint handling, and consistency checking, while auxiliary modules ensure structural control and statistical alignment. For instance, AIGT leverages table metadata and long token partitioning to facilitate the generation of wide tables while maintaining quality at scale (Zhang et al., 2024c). Regarding text to table transformation, gTBLS formulates cell filling as a conditional question answering task. This approach explicitly targets syntactic validity to reduce the reliance on extensive corrections after the generation process (Sundar et al., 2024). Motivated by findings regarding constraint violations and distributional mismatches (Xu et al., 2025), these systems explicitly combine reasoning based on LLMs with external components. This combination optimizes end-to-end table realism and downstream applicability (Zhang et al., 2024c; Sundar et al., 2024).

## 4.2 Quality Metrics for Tabular Data

**Validity.** Validity is the degree to which synthetic tabular data satisfies schema-level rules, integrity constraints, and functional dependencies, so that each generated row is structurally and logically valid.

For categorical marginals, we additionally report the column wise Chi-squared test as a sanity check criterion. This metric is used in TabSyn as a quality indicator and typically requires passing at a high threshold such as $p \geq 0.95$ (Zhang et al., 2024a). The calculation is defined as follows:

$$\chi^2 = \sum_{c \in \Omega} \frac{(O_c - E_c)^2}{E_c}, \qquad p = \Pr\left(\chi^2_{\mathrm{df}} \geq \chi^2\right),$$

where $O_c$ and $E_c$ represent the observed synthetic counts and the expected real counts for category $c$ within the set $\Omega$. A larger $p$ value indicates closer marginal agreement, which signifies that the test fails to reject the hypothesis that the distributions are similar.

To measure structural validity, we report the violation rate, which we denote as VR. This rate is defined as the fraction of integrity checks that fail:

$$\mathrm{VR} = \frac{\#\mathrm{violations}}{\#\mathrm{checks}}.$$

The checks include functional dependencies, range limits, uniqueness, geographic consistency, and other schema constraints. A lower violation rate indicates higher validity.

**Fidelity.** Fidelity measures how closely the synthetic data distribution matches the real data distribution. Following the low order statistics protocol used in TabSyn, we evaluate fidelity at the marginal, pairwise, and global or sample levels (Zhang et al., 2024a).

Marginal fidelity is evaluated with the Kolmogorov-Smirnov Test statistic for numerical columns and the Total Variation Distance for categorical columns (Zhang et al., 2024a). The formulas are defined as follows:

$$\mathrm{KST} = \sup_x \left| F_{\mathrm{real}}(x) - F_{\mathrm{syn}}(x) \right|,$$
$$\mathrm{TVD} = \tfrac{1}{2} \sum_{\omega \in \Omega} \left| R(\omega) - S(\omega) \right|,$$

where $R$ and $S$ denote the real and synthetic empirical frequencies of a category $\omega$ within the set $\Omega$.

Pairwise fidelity is captured by the Pearson Score for numerical pairs and the Contingency Score, also known as ContSim, for categorical pairs. These metrics follow the methodology described in the TabSyn Appendix E.3 (Zhang et al., 2024a). For a numerical pair consisting of variables $x$ and $y$, the Pearson correlation is calculated as follows:

$$\rho(x, y) = \frac{\mathrm{Cov}(x, y)}{\sigma_x \sigma_y},$$

The Pearson Score aggregates correlation gaps using a normalization factor of one half because the correlation coefficient $\rho$ ranges from negative one to one (Zhang et al., 2024a):

$$\mathrm{PearsonScore} = \tfrac{1}{2} \cdot \frac{1}{K} \sum_{(i,j)} \left| \rho^{(i,j)}_{\mathrm{real}} - \rho^{(i,j)}_{\mathrm{syn}} \right|.$$

For categorical pairs consisting of variables $A$ and $B$, the contingency table discrepancy is defined by the following equation:

$$\mathrm{ContingencyScore} = \tfrac{1}{2} \sum_{\alpha \in A} \sum_{\beta \in B} \left| R_{\alpha, \beta} - S_{\alpha, \beta} \right|.$$

For mixed type pairs involving both numerical and categorical data, TabSyn buckets numerical values into categorical bins before computing the corresponding contingency score (Zhang et al., 2024a).

Global realism and detectability can be assessed by applying a classifier two sample test in the same spirit as TabSyn (Lopez-Paz & Oquab, 2018). This approach uses the SDMetrics detection score based on logistic regression (Zhang et al., 2024a; DataCebo, 2024). In this context, the area under the curve is defined as the mean receiver operating characteristic area under the curve of the discriminator when distinguishing between real and synthetic data over cross validation splits. SDMetrics defines the detection score as follows (DataCebo, 2024):

$$\text{DetScore} = 1 - \big( \max(\text{AUC}, 0.5) \times 2 - 1 \big).$$

This formula maps an area under the curve of zero point five, which indicates that the data is indistinguishable, to a detection score of one. Similarly, an area under the curve of one, which indicates that the data is perfectly distinguishable, is mapped to a detection score of zero. Therefore, a higher detection score suggests higher fidelity from the perspective of global detectability (DataCebo, 2024; Zhang et al., 2024a).

Sample-level fidelity and $\alpha$-precision are reported using the support-based definition of Alaa et al. (2022), which is also adopted by the TabSyn framework (Zhang et al., 2024a). In this evaluation, let $P_r$ and $P_g$ denote the real and synthetic distributions where $S_r$ is the support of the real distribution and $S_g$ is the support of the synthetic distribution. Following the work of Alaa et al. (2022), the $\alpha$-support of the real distribution is defined as the minimum volume subset of $S_r$ that contains a probability mass equal to $\alpha$:

$$S_r^\alpha \triangleq \arg \min_{S \subseteq S_r} \text{Vol}(S) \quad \text{s.t.} \quad P_r(S) = \alpha,$$

In this context, the volume function represents the Lebesgue volume measure. The corresponding $\alpha$-precision is defined as the probability that a synthetic sample lies within the real $\alpha$-support (Alaa et al., 2022):

$$P_\alpha \triangleq \Pr_{x \sim P_g} (x \in S_r^\alpha).$$

In practice, we estimate the alpha support from finite samples and compute the empirical $\alpha$-precision by averaging binary membership indicators over the set of synthetic samples (Alaa et al., 2022):

$$\widehat{\alpha\text{-Prec}} = \frac{1}{|\hat{X}|} \sum_{\hat{x} \in \hat{X}} \mathbf{1}\left[\hat{x} \in \widehat{S}_r^\alpha\right],$$

where $\hat{X}$ represents the set of synthetic samples and $\widehat{S}_r^\alpha$ is an estimated $\alpha$-support of the real distribution.

**Diversity.** The diversity of the generated data is evaluated to measure the coverage of the real distribution.

$\beta$-Recall has been proposed by Alaa et al. (2022), and is also adopted in the TabSyn framework (Zhang et al., 2024a). The $\beta$-support for the synthetic distribution is defined in a similar way as follows:

$$S_{\text{syn}}^\beta = \arg \min_S |S| \quad \text{subject to} \quad \Pr(X_{\text{syn}} \in S) = \beta.$$

Based on this definition, $\beta$-Recall is expressed by the following equation (Alaa et al., 2022):

$$R_\beta = \Pr_{X_{\text{real}} \sim P_{\text{real}}} \big(X_{\text{real}} \in S_{\text{syn}}^\beta\big).$$

The sample-level estimator for this metric can be written as:

$$\widehat{\beta\text{-Rec}} = \frac{1}{|X|} \sum_{x \in X} \mathbf{1}\left[x \in \widehat{S}_{\text{syn}}^\beta\right].$$

A high $\beta$-Recall value indicates that the synthetic data provides broad coverage of the original distribution and serves as a complement to the Alpha-Precision metric (Alaa et al., 2022; Zhang et al., 2024a).

**Utility.** Downstream utility can be evaluated using the Train-on-Synthetic Test-on-Real protocol, which is also referred to as Machine Learning Efficiency in the TabSyn framework (Zhang et al., 2024a).

Following the evaluation methodology of TabSyn, the Area Under the Curve for classification tasks and the Root Mean Square Error for regression tasks are reported (Zhang et al., 2024a):

$$\text{AUC} = \int_0^1 \text{TPR}(\text{FPR}) \, d\text{FPR}, \quad \text{MSE} = \frac{1}{n} \sum_{i=1}^n (\hat{y}_i - y_i)^2, \qquad \text{RMSE} = \sqrt{\text{MSE}}.$$

### 4.3 Trustworthy Metrics for Tabular Data

**Privacy.** Privacy risks can be quantified from three complementary perspectives including geometric memorization through nearest neighbor proximity, empirical inference leakage through membership or attribute inference, and formal privacy guarantees such as differential privacy.

To assess geometric memorization and potential record replication, we analyze the distance to the closest record. Following prior work on tabular synthesis based on LLMs, for each synthetic record $s \in \hat{D}$ we compute its distance to the nearest training record (Borisov et al., 2023; Fang et al., 2024):

$$\mathrm{DCR}_{\mathrm{train}\to\mathrm{gen}}(s) = \min_{x \in D_{\mathrm{train}}} d(s, x),$$

where systematically low values indicate a warning signal of potential copying. Because absolute values for the distance to the closest record depend on the choice of distance $d$ and feature scaling, we recommend comparing the distribution of these values against a real baseline. This baseline is defined as $\mathrm{DCR}_{\mathrm{train}\to\mathrm{test}}(x) = \min_{x' \in D_{\mathrm{train}}} d(x, x')$ for hold out real records, which is a common practice in tabular synthesis evaluations (Borisov et al., 2023). Conversely, to assess whether the generator covers the support of the real distribution instead of collapsing to a few modes, we compute the real to synthetic proximity on hold out data:

$$\mathrm{DCR}_{\mathrm{real}\to\mathrm{syn}}(x) = \min_{s \in \hat{D}} d(x, s),$$

where a distribution comparable to real baselines suggests realistic coverage rather than a degenerate generator (Borisov et al., 2023).

Beyond geometric metrics, empirical inference leakage can be evaluated through adversarial attacks. Membership inference risk can be quantified in the standard setting where an attacker is given a candidate record $x$ and some access to the released model or synthesizer. The attacker aims to decide whether $x$ was part of the training data (Shokri et al., 2017). Let $g(x)$ denote an attack score where larger values indicate higher confidence that $x$ is a member of the training set. This risk can be summarized using the Area Under the Receiver Operating Characteristic Curve and the membership advantage:

$$\mathrm{AUC}_{\mathrm{MIA}} = \Pr[g(x_{\mathrm{train}}) > g(x_{\mathrm{holdout}})], \quad \mathrm{Adv}_{\mathrm{MIA}} = \max_{\tau} \left(\mathrm{TPR}(\tau) - \mathrm{FPR}(\tau)\right).$$

Values for the Area Under the Curve significantly above 0.5 or advantage values significantly above 0 indicate non-trivial leakage. The advantage form matches the standard notion based on the difference between the True Positive Rate and the False Positive Rate used to quantify inference leakage (Yeom et al., 2018).

Attribute inference for a sensitive attribute $A$ can similarly be evaluated. In this setting, an attacker predicts the value of $A$ from the remaining attributes and any access granted by the released model (Yeom et al., 2018). In addition to reporting standard predictive metrics such as accuracy or the Area Under the Curve, a simple gain over majority summary can optionally be reported as an implementation level diagnostic:

$$\mathrm{Gain}_{\mathrm{AIA}} = \Pr[\hat{a} = A] - \max_{a} \Pr[A = a],$$

where larger values indicate stronger recoverability of the attribute beyond a trivial majority baseline.

The differential privacy parameters $\varepsilon$ and $\delta$, which represent the privacy budget, are reported together with downstream utility. This approach allows the trade-off between privacy and utility to be contextualized under the standard differential privacy framework (Dwork & Roth, 2014).

**Fairness.** Fairness can be evaluated in a train on synthetic and test on real setting. This is necessary because downstream models trained on synthetic data may inherit or even amplify disparities when they are deployed on real populations. This setting is a commonly used evaluation protocol for tabular generation. Fairness considerations are concerns that exist across the entire pipeline starting from data through the model and ending at deployment (Barocas et al., 2023).

For outcome fairness including demographic parity and disparate impact, let us consider a protected attribute $A \in \{a, b\}$ and a binary decision $\hat{Y}$. The Statistical Parity Difference and the Disparate Impact Ratio are

given by:

$$\Delta_{\text{SPD}} = \left| \Pr(\hat{Y} = 1 \mid A = a) - \Pr(\hat{Y} = 1 \mid A = b) \right|, \quad \text{DIR} = \frac{\Pr(\hat{Y} = 1 \mid A = a)}{\Pr(\hat{Y} = 1 \mid A = b)}.$$

Ideally, the Statistical Parity Difference should approach 0 and the Disparate Impact Ratio should approach 1. The Disparate Impact Ratio is widely used in the literature concerning disparate impact as a rate ratio criterion such as the eighty percent rule (Feldman et al., 2015; Barocas et al., 2023).

To account for the trade-offs regarding accuracy, we evaluate error rate fairness through equalized odds and equal opportunity disparities. These metrics measure the gaps between groups in the True Positive Rate and the False Positive Rate by conditioning on the true label (Hardt et al., 2016):

$$\Delta_{\text{EO}} = \frac{1}{2} \left( |\Delta\text{TPR}| + |\Delta\text{FPR}| \right), \quad \Delta_{\text{EOp}} = |\text{TPR}_a - \text{TPR}_b|.$$

Because parity degradations in the train on synthetic and test on real setting can come from representational mismatches between synthetic and real data, we also report two simple diagnostics. The subgroup coverage gap measures shifts in group proportions, and the label conditional shift captures label distortion within specific groups:

$$\text{CovGap} = \frac{1}{2} \sum_{g \in \mathcal{A}} |p_{\text{syn}}(A{=}g) - p_{\text{real}}(A{=}g)|, \quad \text{CondShift} = \frac{1}{|\mathcal{A}|} \sum_{g \in \mathcal{A}} \text{TV}(p_{\text{syn}}(Y \mid A{=}g), \, p_{\text{real}}(Y \mid A{=}g)).$$

Large values in these representational diagnostics often help explain why parity metrics degrade after the evaluation. This observation is consistent with broader discussions of fairness as a property of the entire pipeline starting from data through the model and ending at deployment (Barocas et al., 2023).

## 4.4 Evaluation Practice Gap

We analyze representative methods from Section 4.1 and categorize their reported evaluation protocols in Table 4.

Table 4: Whether representative tabular data generation methods explicitly evaluate each dimension in their experimental sections, together with their method family and functional role. ✓: explicitly evaluated; △: partially or indirectly covered; ×: not reported or not applicable.

| Representative Work | Family | Role | Validity | Fidelity | Diversity | Utility | Privacy | Fairness |
|---|---|---|---|---|---|---|---|---|
| GReaT (Borisov et al., 2023) | Fine-Tuning | Generation | ✓ | ✓ | × | ✓ | ✓ | × |
| REaLTabFormer (Solatorio & Dupriez, 2023) | Fine-Tuning | Generation | △ | ✓ | × | ✓ | ✓ | × |
| EPIC (Kim et al., 2025) | Prompt-Based | Generation | △ | ✓ | △ | ✓ | × | × |
| HARMONIC (Wang et al., 2024d) | Fine-Tuning | Generation | × | △ | × | ✓ | △ | × |
| UDF-MIX (Li et al., 2025b) | Fine-Tuning | Generation | × | △ | × | ✓ | × | ✓ |
| FairCauseSyn (Nagesh et al., 2025) | Prompt-Based | Generation | × | △ | × | △ | × | ✓ |
| AIGT (Zhang et al., 2024c) | Hybrid Architectures | Generation | △ | ✓ | × | ✓ | ✓ | × |

**Diversity.** In the representative audit summarized in Table 4, diversity is not explicitly evaluated by any of the seven audited tabular synthesis methods and is only partially covered in one work. This happens because specific measures focused on coverage such as $\beta$-Recall are rarely reported in experimental settings. This gap is very important for generators driven by LLMs. In these models, standard training goals based on likelihood and decoding procedures often show behavior that focuses on the most common patterns. Thus, synthetic distributions tend to overrepresent frequent patterns while failing to capture long-tail values and

rare combinations of features. Although synthetic datasets may seem realistic within dominant patterns, they often lack enough support for areas with low frequency. We therefore recommend reporting coverage metrics such as $\beta$-Recall and diagnostics at the subgroup level alongside standard measures of fidelity and utility.

**Fairness.** Within the representative audit in Table 4, fairness-oriented evaluation appears in only two of the seven audited tabular generation works. We interpret this as a comparative signal within the audited sample rather than a field-wide prevalence estimate. This comparatively sparse reporting matters for the Train-on-Synthetic-and-Test-on-Real (TSTR) protocol. When synthetic data are used to train downstream models for deployment on real populations, representational mismatches can lead directly to disparities in outcome fairness and error rate fairness. These mismatches include shifts in subgroup proportions or label conditional distributions. For example, outcome fairness can be measured by statistical parity, and error rate fairness can be measured by equalized odds. Moreover, improved diversity does not necessarily imply improved fairness. Expanding the coverage of the data may accidentally increase disparities if minority subgroups are created with lower fidelity or if the conditional distributions are distorted. Accordingly, we recommend adding group fairness metrics such as the statistical parity difference and equalized odds to the results of the train on synthetic and test on real protocol. We also suggest using representational diagnostics to make the trade-offs among diversity, fairness, and downstream performance explicit.

## 4.5 Usage

**Data Sharing.** Synthetic tabular data based on LLMs is increasingly used to facilitate safe data sharing and privacy-aware anonymization. By producing records that maintain statistical fidelity to the source distribution while differing from original data points, organizations can release or exchange datasets without compromising sensitive or personally identifiable information. This capability is especially important in regulated domains such as healthcare and finance. In these fields, synthetic data serves as a compliant mechanism for collaboration between different institutions (Miletic & Sariyar, 2024; Barr et al., 2025; Long et al., 2025).

In this context, Miletic & Sariyar (2024) benchmark LLMs based on the Transformer architecture against baseline models such as CTGAN (Xu et al., 2019). They show that larger language models yield superior downstream classification utility and maintain competitive performance even at smaller scales. Furthermore, Barr et al. (2025) show that GPT-4o is capable of generating clinical tabular data in a zero-shot manner and achieving high fidelity in terms of the statistical properties within each column. This performance is especially high when the model is guided by descriptive statistics. However, the authors did not directly evaluate downstream utility or the risks of duplication and memorization. They highlighted these areas as important directions for further study.

Beyond the imitation of individual rows, frameworks such as LLM-TabFlow use the reasoning capabilities of LLMs to capture logical relationships between columns and synthesize data within a latent space. This approach helps optimize the trade-off between fidelity, utility, and privacy (Long et al., 2025). Together, these findings indicate that synthesis driven by LLMs is becoming a practical way to share sensitive tables under strict regulatory constraints.

**Data Augmentation.** In situations characterized by data scarcity or a lack of rare feature combinations, generation driven by LLMs provides a way to increase training distributions. This approach helps to improve class balance and the ability of models to generalize to new data (Seedat et al., 2024; Tran & Xiong, 2024; Kim et al., 2025; Yang et al., 2024b).

Recent research is moving beyond simple oversampling methods toward more controllable ways of creating tabular data. In some cases, these methods include formal privacy guarantees. For instance, DP-LLMTGen ensures differential privacy by using a two-stage fine-tuning pipeline. This process first involves learning the data format and then performing fine-tuning with differential privacy using a loss function designed for tabular data, and finally samples the private model to create synthetic tables (Tran & Xiong, 2024).

In a related direction, P-TA uses an optimization scheme based on Proximal Policy Optimization to include feedback from a discriminator. This improves the alignment between the generated data and the real tabular

distributions (Yang et al., 2025). As a result, approaches in this field are transitioning from basic oversampling techniques to generation frameworks that are both controllable and aware of privacy. These developments make generators based on LLMs flexible tools for improving the performance of downstream models in difficult data environments.

# 5 Semi-structured Data

Semi-structured data can be thought of as an intermediate modality between rigid relational schemas and unstructured text. This modality is characterized by flexible schemas and hierarchical organization. In this survey, we unify **Graph**, **JSON**, and **Log** data under this category.

Graph data consist of nodes and edges that define topological relationships. Recent literature demonstrates that LLMs can synthesize such structures by serializing them into text formats such as edge lists (Yao et al., 2024).

JSON data represent nested structures of objects consisting of name-value pairs and arrays (Bray, 2017). However, reliable JSON generation requires strict schema adherence and syntax verification to ensure the output is executable by downstream tools (Agarwal et al., 2025a).

Log data consist of sequential event messages where each entry typically combines a timestamp with a log template and variable parameters. This format allows for automatic parsing into structured templates (He et al., 2017).

## 5.1 Generation Methods

### 5.1.1 Graph Data

Recent advancements in LLM-driven graph synthesis can be classified based on the necessity of parameter updates into training-free generation versus learning-based generation.

**Training-free Graph Generation.** A growing body of work explores the capability of LLMs to produce syntactically valid and structurally plausible graphs without gradient-based tuning. Yao et al. (2024) introduce LLM4GraphGen, demonstrating that models such as GPT-4 can generate graph structures directly from natural language prompts. This work covers both rule-based and distribution-based tasks. While direct prompting offers a domain-agnostic and deployment-friendly approach, existing evaluations indicate limitations in structural fidelity. This is particularly evident when targeting complex distributional parameters such as specific motif counts (Yao et al., 2024). To move beyond naive prompting, Generate-on-Graph or GraphLingo tackle Incomplete Knowledge Graph Question Answering through a training-free exploration procedure. This method retrieves knowledge graph evidence and generates additional factual triples dynamically when crucial triples are missing (Xu et al., 2024b; Le et al., 2024a). In parallel, ontology-grounded Knowledge Graph construction leverages a Wikidata-schema-aligned ontology to ground relation extraction and improve interoperability with existing ontologies (Feng et al., 2024).

**Learning-Based and Multi-Agent Generation.** Alternative approaches enhance the quality of graph construction through supervised fine-tuning or multi-agent collaboration frameworks (Huang et al., 2025a; Le et al., 2024a). GraphJudge (Huang et al., 2025a) employs supervised fine-tuning to train LLMs as discriminators for graph quality. This approach significantly reduces noise during Knowledge Graph enrichment by filtering generated triples. Beyond the use of individual models, multi-agent systems such as GAG and GraphMaster leverage collaborative refinement to improve global consistency. GAG formulates graph synthesis as a scalable and simulation-based multi-agent process. This method enables the generation of large text-attributed graphs that adhere to macroscopic network properties (Ji et al., 2025). Similarly, GraphMaster introduces an evaluation-driven iterative loop among agents. This system is specifically designed to enhance the structural integrity and semantic coherence of the synthesized graphs (Du et al., 2025).

### 5.1.2 JSON Data

LLMs have been adapted to generate schema-compliant JSON through two representative methodological classes. These include inference time constraint control, often referred to as guided decoding, and learning based alignment such as reinforcement learning to improve schema adherence (vLLM Project, 2024; Lu et al., 2025; Agarwal et al., 2025a).

**Constrained Decoding for JSON Generation.** In this regime, model parameters remain unchanged and structural compliance is enforced at inference time using decoding time constraints. These constraints include token filtering guided by schemas or regular expressions (vLLM Project, 2024; Gat, 2025; dottxt-ai, 2025). As a complementary step, LLMs can also be used to infer and enrich JSON Schemas from existing corpora. This can be achieved by generating natural language descriptions for schema elements and identifying potentially noisy properties (Mior, 2024). Once a target schema is available, practical frameworks such as vLLM structured outputs and toolkits like Outlines and LM Format Enforcer implement constrained decoding by masking invalid tokens. These systems ensure that the final output conforms to the target schema and prevent malformed JSON that may arise under unconstrained prompting. Furthermore, JSONSchemaBench provides large-scale evidence and analysis of compliance, coverage, and efficiency for these constrained decoding systems across 10,000 real world JSON schemas (vLLM Project, 2024; dottxt-ai, 2025; Gat, 2025; Geng et al., 2025). Recent work also begins to examine diversity among valid structured outputs. For example, automata-based steering improves structural and lexical diversity by encouraging exploration over valid automaton paths while preserving decoding efficiency (Luan et al., 2025). However, while these approaches excel at enforcing surface form validity, satisfying stricter requirements may still require additional validation, post processing, or learning based alignment. Such requirements include fine-grained field typing and cross field semantic consistency (Lu et al., 2025).

**Learning-Based Generation.** When stricter schema adherence is required, reinforcement learning based methods optimize the model policy with reward signals tied to schema correctness. These rewards are typically provided via schema validators or verifier style rewards (Lu et al., 2025; Agarwal et al., 2025a). For instance, Lu et al. (2025) introduce SchemaBench which contains approximately 40,000 JSON schemas. They improve structured generation by incorporating reinforcement learning with a fine-grained schema validator. This approach outperforms standard supervised fine-tuning baselines (Lu et al., 2025). Meanwhile, Agarwal et al. (2025a) apply Group Relative Policy Optimization to train smaller models with custom rewards for strict schema adherence. This work demonstrates effective improvements in enforcing schema consistency.

### 5.1.3 Log Data

Research in log generation highlights a discrepancy between the semantic prediction of log components and the syntactic realization of the final log message. These components include log levels and variables. Li et al. (2024d) observe that while LLMs can accurately determine necessary logging attributes, they often fail to produce full log statements that mimic human-written code. On their benchmark, LogBench, the best-performing models achieved a BLEU score of only 0.249. This indicates limited surface-form similarity (Li et al., 2024d). Furthermore, evaluations on semantically equivalent but transformed code contexts, referred to as LogBench-T, show consistent performance degradation. This is especially true for variable prediction and log-text generation, suggesting that general-purpose LLMs remain brittle under semantics-preserving code transformations (Li et al., 2024d).

To bridge this performance gap, recent studies advocate for model specialization via post-training on domain-specific corpora. Zhang et al. (2025a) propose AUCAD, which is a framework that automatically constructs an alignment dataset called AucadLog derived from log-related software issues. By leveraging this dataset to post-train open-source LLMs, the authors demonstrate that the resulting specialized models significantly outperform existing model-based solutions in log statement generation. These results are confirmed by both human evaluation and quantitative metrics (Zhang et al., 2025a). Related work also studies structured event log generation rather than source-code log statements. In this setting, FSM-constrained GFlowNets generate logs under finite-state constraints and are evaluated by distributional similarity, behavioral diversity, and downstream intent classification (Samanta, 2025).

### 5.2 Quality Metrics for Semi-structured Data

### 5.2.1 Graph Data

**Validity.** Validity represents the proportion of generated graphs that comply with specific task rules or logical constraints. This metric reflects whether the output of the model is effective in a topological or physical sense.

The fraction of generated graphs that satisfy task-specific rule-based constraints is given by:

$$\text{Valid}_{\text{rule}} = \frac{1}{|\mathcal{G}_{\text{gen}}|} \sum_{G \in \mathcal{G}_{\text{gen}}} \mathbb{I}\{\text{passes\_rules}(G)\},$$

where $\mathcal{G}_{\text{gen}}$ denotes the set of generated graphs where $\mathcal{G}_{\text{gen}} \subseteq \mathcal{G}$. The function passes\_rules is a task-specific rule checker that returns 1 if $G$ satisfies the rules and 0 otherwise. These rules may include constraints specified by the generation task (Yao et al., 2024).

**Fidelity.** Fidelity measures the structural similarity between generated graphs and reference data.

The similarity between generated graphs and reference structures can be assessed using kernel based Maximum Mean Discrepancy computed on graph descriptor features:

$$\text{MMD}^2(\mathcal{X}, \mathcal{Y}) = \frac{1}{m^2} \sum_{i,i'} k(x_i, x_{i'}) + \frac{1}{n^2} \sum_{j,j'} k(y_j, y_{j'}) - \frac{2}{mn} \sum_{i,j} k(x_i, y_j),$$

where $\mathcal{X} = \{x_i\}_{i=1}^m$ and $\mathcal{Y} = \{y_j\}_{j=1}^n$ are descriptor sets extracted from generated and real graphs respectively, and $k$ is a chosen kernel (You et al., 2018; Liao et al., 2020). Typical descriptors include degree distributions, clustering coefficient distributions, orbit or motif counts, and spectral statistics such as Laplacian eigenvalue histograms (You et al., 2018; Liao et al., 2020).

For molecular graphs, we include the **Frechet ChemNet Distance** (Preuer et al., 2018):

$$\text{FCD} = \|\mu_1 - \mu_2\|_2^2 + \text{Tr}\Big(\Sigma_1 + \Sigma_2 - 2\big(\Sigma_1 \Sigma_2\big)^{1/2}\Big).$$

This metric measures the distance between embedding distributions of generated and reference molecules. $\mu$ and $\Sigma$ represent the mean and covariance of these distributions where embeddings are obtained from a pretrained ChemNet model (Preuer et al., 2018).

**Diversity.** Diversity evaluates the variety and originality among the generated graphs. This includes the novelty of the graphs relative to the prompt and the non-repetition within the generated set. To evaluate variety, we compute Novelty. A generated graph is considered novel if it is different from every example graph provided in the prompt for the same task:

$$\text{Novelty} = \frac{1}{|\mathcal{G}_{\text{gen}}|} \sum_{G \in \mathcal{G}_{\text{gen}}} \mathbb{I}\{G \not\equiv \mathcal{G}_{\text{ex}}\},$$

where $\mathcal{G}_{\text{ex}} \subseteq \mathcal{G}$ denotes the set of example graphs in the prompt. The condition $G \not\equiv \mathcal{G}_{\text{ex}}$ means $G$ is not identical to any example graph under the same equality criterion used by the evaluation pipeline (Yao et al., 2024).

Uniqueness is defined as the fraction of valid generated graphs that are not duplicates:

$$\text{Uniq} = \frac{|\text{unique}(\mathcal{G}_{\text{valid}})|}{|\mathcal{G}_{\text{valid}}|}, \qquad \mathcal{G}_{\text{valid}} = \{G \in \mathcal{G}_{\text{gen}} : \text{passes\_rules}(G)\},$$

where the function unique removes duplicates under the same equality criterion used in evaluation (Yao et al., 2024).

**Utility.** Utility assesses the practical value of the generated data for downstream applications. This is typically measured by how much the synthetic data contributes to the performance of a model on a real world task. When a Graph Neural Network is trained on enhanced graphs and evaluated on a fixed split, we report Accuracy and F1 Score. Higher values for Accuracy and F1 indicate greater utility of the synthetic graphs for downstream tasks (Du et al., 2025).

### 5.2.2 JSON Data

For semi-structured outputs such as JSON responses, quality metrics commonly focus on three aspects. These dimensions include validity, fidelity, and utility.

**Validity.** Validity represents the requirement that outputs are machine-parseable and conform to the expected schema. This ensures that the generated data is effective for automated processing. Let $\mathcal{J} = \{J_1, \dots, J_N\}$ be the set of generated JSON objects and $\mathcal{S}$ be the target schema.

The Correctness Indicator for a generated object $J$ under schema $\mathcal{S}$ is defined as follows:

$$V(J, \mathcal{S}) = \mathbb{I}[\text{parsable}(J) \wedge \text{schema}(J, \mathcal{S})].$$

In this equation, $\mathbb{I}[\cdot]$ is the indicator function that returns 1 if $J$ is syntactically valid and schema-compliant, and 0 otherwise.

The average Correctness Rate is given by:

$$\text{CorrectnessRate} = \frac{1}{|\mathcal{J}|} \sum_{J \in \mathcal{J}} V(J, \mathcal{S}).$$

In addition, a purely syntactic metric, referred to as the Valid JSON rate or parsability rate, can be written as follows. This metric measures whether the output is parseable as JSON while ignoring specific schema constraints:

$$\text{ValidJSONRate} = \frac{\left|\{J \in \mathcal{J} \mid \text{IsParsable}(J) = \text{true}\}\right|}{|\mathcal{J}|}.$$

These two metrics capture distinct aspects of structured output quality. The first is a purely syntactic parsability metric referred to as the ValidJSONRate. The second is a schema level validity metric referred to as the CorrectnessRate. Parsability metrics are explicitly reported in strict structured output evaluations (Agarwal et al., 2025a). At the same time, schema validation protocols are commonly used in settings involving schema-constrained generation (Lu et al., 2025).

**Fidelity.** Fidelity measures how closely the generated content matches the semantic information and distribution of the target data. This dimension reflects the quality of the content within the structured JSON framework. Fidelity metrics evaluate whether the model produces the correct fields, values, and semantics beyond simple syntactic validity.

For structured JSON where ground truth instances denoted as $J_i^{\text{gt}}$ are aligned to generated instances $J_i$, the **Mean Match Percentage** compares generated fields against the ground truth instantiations:

$$MMP = \frac{1}{N} \sum_{i=1}^{N} \frac{|\text{fields}(J_i) \cap \text{fields}(J_i^{\text{gt}})|}{|\text{fields}(J_i^{\text{gt}})|}.$$

This metric can be extended to checks at the level of values or constraints when ground truth values and constraints are available (Agarwal et al., 2025a).

Schema focused evaluations also examine whether the schema itself is meaningful. For example, researchers may assess the quality of generated definitions and names through similarity based on embeddings. Additionally, property selection is evaluated as a classification problem between useful and noisy properties. This is typically reported using accuracy (Mior, 2024).

**Diversity.** Diversity captures the breadth and non-redundancy of generated samples beyond validity and fidelity, and is increasingly recognized as a critical yet under-explored dimension in synthetic data evaluation (Zhu et al., 2025b; Chen et al., 2024a). For structured outputs such as JSON, diversity can be quantified at multiple complementary levels. At the surface level, lexical diversity is commonly measured using the Distinct-$n$ metric:

$$\text{Distinct-}n(D) = \frac{\left|\text{Unique}\big(\text{n-grams}(\text{Concat}(D))\big)\right|}{|\text{n-grams}(\text{Concat}(D))|},$$

which evaluates the proportion of unique $n$-grams in serialized outputs. At the representation level, diversity can be characterized by dispersion in embedding space, such as K-means inertia:

$$\text{Inertia}(D) = \sum_{c_k \in C,\ h_j \in H_{c_k}} (h_j - c_k)^2,$$

and spectral entropy-based measures like VendiScore:

$$\text{VS}(D) = \exp\left(-\sum_{i=1}^{n} \lambda_i \log \lambda_i\right),$$

where $\lambda_i$ are eigenvalues of a normalized similarity kernel. Building on this perspective, recent work proposes DCScore, which models diversity as a classification problem over sample similarities. Given a kernel matrix $K$, the probability matrix is defined as:

$$P[i,j] = \frac{\exp(K[i,j]/\tau)}{\sum_j \exp(K[i,j]/\tau)},$$

and diversity is summarized as:

$$\text{DCScore}(D) = \text{tr}(P) = \sum_{i=1}^{n} P[i,i],$$

which reflects the effective number of distinguishable samples in the dataset (Zhu et al., 2025b). Beyond embedding-based metrics, model-based approaches such as LLM-driven clustering quantify diversity via:

$$D = \frac{1}{N} \sum_{i=1}^{N} \frac{C_i}{S_i},$$

where $C_i$ and $S_i$ denote cluster counts and sizes across repeated runs (Chen et al., 2024a).

**Utility.** Utility evaluates the practical effectiveness of the generated JSON data in solving a specific task or its usefulness as an input for downstream processes.

To evaluate the utility of generated JavaScript Object Notation responses for later use, each response is required to follow a predefined schema so that the data can be read reliably. Task success can be measured by calculating the exact match accuracy between the extracted answer and the ground truth. Let the set of generated responses be denoted as $\{J_i\}_{i=1}^{N}$ and the reference answers be denoted as $A_{\text{gt},i}$. Task Accuracy is defined as the average rate at which the extracted answer matches the reference:

$$\text{TaskAcc} = \frac{1}{N} \sum_{i=1}^{N} \mathbb{I}\left(\text{ExtractAnswer}(J_i) = A_{\text{gt},i}\right).$$

This measurement reflects how well the synthetic data supports the successful completion of the target application. This methodology aligns with evaluations based on task accuracy for schema-constrained structured outputs (Geng et al., 2025).

### 5.2.3 Log Data

Logs are typically semi-structured in nature. Each message consists of a fixed template combined with runtime variables or parameters, which may also be accompanied by metadata such as severity levels and components. Accordingly, evaluation should separately address two distinct aspects. The first is structural validity under a log parser, which involves template and variable extraction as well as grouping. The second is the content fidelity of the generated text, variables, and metadata.

**Validity.** For log data, validity depends not only on surface well-formedness but also on whether the logs can be correctly structured into templates and variables by a parser. A common message-level metric used for this purpose is Parsing Accuracy, which measures the fraction of log messages that are parsed exactly into the correct template and variables:

$$\mathrm{PA} = \frac{|\mathcal{L}_{\mathrm{correct}}|}{|\mathcal{L}|},$$

where $\mathcal{L}$ represents the set of all log messages $\{\ell_i\}$, and $\mathcal{L}_{\mathrm{correct}}$ refers to the subset whose predicted template, consisting of static tokens, and variable spans, consisting of dynamic tokens, match the ground truth exactly (Khan et al., 2022; Jiang et al., 2024a; Ma et al., 2024b).

Beyond per-message correctness, Grouping Accuracy evaluates whether messages are assigned to the correct template group:

$$\mathrm{GA} = \frac{|\mathcal{L}_{\mathrm{grouped}}|}{|\mathcal{L}|},$$

where $\mathcal{L}_{\mathrm{grouped}}$ denotes messages whose predicted grouping is consistent with the ground-truth template partition. This means that messages belonging to the same true template are grouped together and those from different templates are separated (Khan et al., 2022; Ma et al., 2024b). Grouping Accuracy is particularly important when downstream pipelines rely on template clusters rather than individual parses.

Since Parsing Accuracy and Grouping Accuracy are based on message counts, they can be overly influenced by highly frequent templates, such as heartbeat messages. This motivates the use of template-level evaluation (Khan et al., 2022; Jiang et al., 2024a). Let $\mathcal{T}_{gt}$ and $\mathcal{T}_p$ be the sets of ground-truth and predicted templates, and let $\mathcal{T}_{\mathrm{correct}}$ be the set of correctly identified templates. This is often reported with oracle template correction following common evaluation guidelines (Khan et al., 2022). Define template-level precision and recall as

$$P_{\mathrm{TA}} = \frac{|\mathcal{T}_{\mathrm{correct}}|}{|\mathcal{T}_p|}, \qquad R_{\mathrm{TA}} = \frac{|\mathcal{T}_{\mathrm{correct}}|}{|\mathcal{T}_{gt}|}.$$

The F1-score of Template Accuracy is defined as

$$\mathrm{FTA} = 2 \cdot \frac{P_{\mathrm{TA}} \cdot R_{\mathrm{TA}}}{P_{\mathrm{TA}} + R_{\mathrm{TA}}},$$

which complements Parsing Accuracy and Grouping Accuracy by emphasizing template coverage and uniqueness (Khan et al., 2022; Jiang et al., 2024a). Recent large-scale benchmarks further recommend additional template-level grouping measures, such as the F1-score of Group Accuracy, to mitigate the sensitivity of message-level grouping metrics when template frequencies are imbalanced (Jiang et al., 2024b). Overall, these metrics capture whether generated logs support faithful structural parsing beyond mere syntactic correctness.

**Fidelity.** For logs, fidelity emphasizes how well generated templates, variables, and metadata preserve linguistic realism and operational plausibility. When models generate or reconstruct variables, Variable Precision, Variable Recall, and the Variable F1-score assess whether the bound runtime variables are correct:

$$\mathrm{VP} = \frac{|V_p \cap V_t|}{|V_p|}, \quad \mathrm{VR} = \frac{|V_p \cap V_t|}{|V_t|}, \mathrm{VF1} = 2 \cdot \frac{\mathrm{VP} \cdot \mathrm{VR}}{\mathrm{VP} + \mathrm{VR}},$$

where $V_p$ and $V_t$ are the predicted and true variable sets (Li et al., 2024d).

For template and text realism, overlap-based metrics such as BLEU and ROUGE can be used between generated and reference log templates or texts. In addition, embedding-based semantic similarity is often reported to reduce sensitivity to paraphrases and mixed natural language or code phrasing (Li et al., 2024d).

Metadata fidelity is also crucial. For example, Log Level Accuracy measures the fraction of logs with exactly correct severity levels:

$$\text{L-ACC} = \frac{N_{\text{correct\_level}}}{N}$$

Furthermore, the Average Ordinal Distance captures how far predicted levels deviate on an ordinal severity scale:

$$\text{AOD} = \frac{1}{N} \sum_{i=1}^{N} \left( 1 - \frac{\text{Dis}(l_{\text{pred}}^{(i)}, l_{\text{gt}}^{(i)})}{\text{MaxDis}} \right),$$

where $\text{Dis}(\cdot)$ is the ordinal distance between levels, such as the ranking from error to trace, and $\text{MaxDis}$ is the maximum possible distance on the chosen scale (Li et al., 2024d).

**Diversity.** For log data, diversity captures the variability of generated log templates, parameters, and event sequences, ensuring that synthetic logs do not collapse into a small number of repetitive patterns. Following prior work on synthetic data diversity (Zhu et al., 2025b; Chen et al., 2024a), diversity can be evaluated at both lexical and structural levels. At the surface level, logs can be serialized into sequences and evaluated using the Distinct-$n$ metric, which measures the proportion of unique token patterns and detects template repetition. To further quantify distributional diversity, entropy over log events or tokens can be computed:

$$H = - \sum_{e \in \mathcal{E}} p(e) \log p(e),$$

where $\mathcal{E}$ denotes the set of event types or tokens and $p(e)$ is the empirical frequency, capturing how evenly the generator explores the event space. At the structural level, diversity can be assessed through pairwise similarity across generated logs. Given a set of structured logs with representations $T_i$, diversity is inversely related to average similarity:

$$C(O) = \frac{2}{n(n-1)} \sum_{i<j} s(T_i, T_j),$$

where $s(\cdot, \cdot)$ measures structural similarity (e.g., template or sequence alignment), and lower values indicate higher diversity. These formulations align with existing structured-output evaluations, where diversity reflects both variation in token sequences and heterogeneity in structural patterns across generated logs.

**Utility.** For logs, downstream utility reflects their contribution to system understanding and diagnosis. A common evaluation approach is to determine whether synthetic or augmented logs improve or preserve performance on downstream log analysis tasks, such as parsing or anomaly detection, under standard pipelines and fixed test sets (Huo et al., 2023).

A direct way to quantify utility is the change in downstream model performance when synthetic logs are used for training or augmentation:

$$\Delta_{\mathcal{M}} = \mathcal{M}\big(f(\mathcal{D}_{\text{real}} \cup \mathcal{D}_{\text{syn}})\big) - \mathcal{M}\big(f(\mathcal{D}_{\text{real}})\big),$$

where $\mathcal{M}$ is a task metric, such as the F1 score or accuracy, and $f$ is the downstream model or procedure evaluated on a fixed test set.

In human-centered evaluations, developers may additionally rate the usefulness and readability of generated logs and explanations on Likert scales to capture practical diagnostic value (Liu et al., 2024d).

### 5.3 Trustworthy Metrics for Semi-structured Data

### 5.3.1 Graph Data

**Privacy.** Graph-structured data raises distinct privacy risks because nodes and edges often correspond to individual users and their relationships. These risks are formalized using differential privacy notions specialized to graph adjacency (Kasiviswanathan et al., 2013).

The formal definition of privacy in graph generation is based on the framework of differential privacy, which is characterized by two key parameters, $\varepsilon$ and $\delta$. A randomized mechanism $M$ satisfies the $(\varepsilon, \delta)$- differential privacy inequality if for any two neighboring datasets $D$ and $D'$ that differ by a single record, and for any possible output set $S$, the probability that the mechanism produces an output in $S$ for dataset $D$ is bounded by the following condition:

$$\Pr[M(D) \in S] \le e^{\varepsilon} \Pr[M(D') \in S] + \delta.$$

In this mathematical framework, $\varepsilon$ represents the privacy budget or privacy loss, which quantifies the maximum allowable change in the output probability distribution when one individual's information is modified. A smaller $\varepsilon$ value indicates a stronger privacy guarantee because it ensures that the outputs are more similar regardless of whether a specific record is present. The parameter $\delta$ represents the probability that the privacy constraint might be violated, and it is typically required to be a very small value to ensure that the guarantee remains robust (Dwork & Roth, 2014).

When this definition is applied to graph structured data, the concept of neighboring datasets is specialized to either node or edge adjacency (Kasiviswanathan et al., 2013). Node level differential privacy defines neighbors as two graphs where one is obtained from the other by removing a single node along with all its incident edges. In contrast, edge level differential privacy considers graphs to be neighbors if they differ by exactly one edge. Under these adjacency notions, the parameters $\varepsilon$ and $\delta$ serve as quantitative metrics for the risk that a specific user or relationship could be inferred from the generated graph (Kasiviswanathan et al., 2013; Dwork & Roth, 2014).

Similarly, edge-level central differential privacy treats neighboring graphs as those differing by exactly one edge, a concept known as edge adjacency (Kasiviswanathan et al., 2013). Under this adjacency notion, the same $(\varepsilon, \delta)$-differential privacy inequality applies (Dwork & Roth, 2014):

$$\Pr[M(G) \in S] \le e^{\varepsilon} \Pr[M(G') \in S] + \delta, \ \forall S \subseteq \mathcal{O}.$$

In this case, the presence or absence of any single relationship or edge is protected.

For more granular protection without a trusted curator, we can work in the local model of differential privacy, where each participant randomizes their contribution before sharing. A standard formalization of local differential privacy is given via a likelihood-ratio bound on the per-user privatization channel (Duchi et al., 2014). Instantiating this standard definition for per-edge reports, such as an edge indicator or local adjacency information, we use the following notion of edge-level local differential privacy. A mechanism $M$ is $\varepsilon$-edge-level local differential privacy if for any two possible edge values $e$ and $e'$ and any output set $S \subseteq \mathcal{O}$, the following holds:

$$\Pr[M(e) \in S] \le e^{\varepsilon} \Pr[M(e') \in S] \qquad \forall e, e', S \subseteq \mathcal{O}.$$

In many local differential privacy settings, one focuses on the pure case described above (Duchi et al., 2014). One can also consider an $(\varepsilon, \delta)$ extension by adding a $\delta$ term as in approximate differential privacy (Dwork & Roth, 2014), but we maintain the common pure form for clarity.

A common implementation in the central model uses global sensitivity and Laplace noise (Dwork & Roth, 2014):

$$\Delta f = \max_{G \sim G'} \|f(G) - f(G')\|_1,$$

$$\eta \sim \text{Lap}\left(0, \tfrac{\Delta f}{\varepsilon}\right),$$

where $f$ is a numeric query, such as degree-histogram bins. The Laplace mechanism yields pure $\varepsilon$-differential privacy, which means $\delta$ is equal to zero, under the chosen adjacency notion (Dwork & Roth, 2014). In graph

settings, $\Delta f$ can be very large for degree-related queries under node adjacency, since adding or removing a node can affect many incident edges and consequently many degree counts. Practical approaches therefore impose bounded-degree assumptions or apply degree clipping and projection to a bounded-degree graph family. These techniques keep $\Delta f$ and the required noise level manageable (Kasiviswanathan et al., 2013).

The privacy parameters $\varepsilon$ and $\delta$ for central differential privacy, and $\varepsilon$ for pure local differential privacy, serve as quantitative privacy metrics. A smaller $\varepsilon$ and a smaller $\delta$ indicate stronger privacy protection, which typically comes at the cost of utility (Dwork & Roth, 2014).

Finally, it is common to require $\delta$ to be negligible in an appropriate problem or security parameter. It is important to avoid regimes where $\delta$ is non-trivially large, such as on the order of one over $n$ for population size $n$ (Dwork & Roth, 2014). In graph settings where nodes correspond to protected individuals, a practical heuristic is to select $\delta$ to be far smaller than one over the number of vertices in the graph when using approximate differential privacy. The exact choice remains dependent on the specific application and threat model.

**Robustness.** Beyond privacy, graph generation must be robust to structural hallucinations, which refer to the production of plausible but incorrect graphs (Richardeau et al., 2025). Complementary checks can be used to capture this risk.

The Syntactic Correctness Rate is defined as the fraction of generated outputs that can be parsed into valid graph structures under the required grammar or format:

$$\sigma_{SCR} = \frac{1}{|\mathcal{G}_{\text{gen}}|} \sum_{G \in \mathcal{G}_{\text{gen}}} \mathbb{1}\{\text{PARSEOK}(G) = 1\}.$$

A low value of $\sigma_{SCR}$ indicates format-breaking outputs and sensitivity to changes in prompting or decoding. This is particularly relevant when the generator is prompted to output a concrete graph representation, such as edge lists, which must be parsed into a graph object for downstream evaluation. Such generate-then-parse pipelines are common in research involving graph generation with LLMs (Richardeau et al., 2025).

The Graph Atlas Distance is employed to directly measure topological deviation for well-specified targets, such as canonical atlas graphs. Following the methodology proposed in (Richardeau et al., 2025), this metric is defined as follows:

$$\text{GAD} = \frac{1}{k} \sum_{i=1}^{k} d_{\text{GED}}(G_i, a_i),$$

where $G_1, \ldots, G_k$ represent the outputs generated from a fixed set of prompts whose ground-truth targets are the corresponding canonical atlas graphs $a_1, \ldots, a_k$. In this formulation, $d_{\text{GED}}$ denotes the graph edit distance as described in (Richardeau et al., 2025). A larger value for the Graph Atlas Distance implies more severe structural hallucination in the generated results.

To reduce sensitivity to rare and extreme edit distances which might otherwise dominate the mean, we additionally report a capped version of the Graph Atlas Distance known as robust aggregation. This metric is calculated as follows:

$$\text{GAD}^{\text{cap}} = \frac{1}{k} \sum_{i=1}^{k} \min\big\{ d_{\text{GED}}(G_i, a_i),\, C \big\},$$

where $C$ caps extreme values. In this context, the uncapped average corresponds to the standard definition of the Graph Atlas Distance proposed in (Richardeau et al., 2025), while the cap serves as a robustification measure during the evaluation process.

The Degree-distribution Deviation captures distributional drift by measuring a lightweight deviation between normalized degree histograms. This metric is defined by the following expression:

$$D_{L_2}(G) = \big\| h_{\text{gen}}(G) - h_{\text{ref}} \big\|_2,$$

where $h_{\text{gen}}(G)$ represents the normalized degree histogram of the generated graph $G$ and $h_{\text{ref}}$ is the target histogram. This target may be derived from the ground-truth graph or a reference dataset. A higher value

of this deviation indicates that the generated graphs may appear reasonable but diverge statistically from the reference model. Comparing degree-based statistics through distributional distances, such as Maximum Mean Discrepancy, Kullback Leibler divergence, or Kolmogorov-Smirnov tests, is standard practice in the evaluation of graph generation (Liao et al., 2020; Liu et al., 2024b). In this study, an $L_2$ histogram deviation is used for simplicity. It is further noted that degree-based deviations provide useful although not sufficient auxiliary signals. Consequently, they should be interpreted alongside topology-sensitive measures such as the Graph Atlas Distance or the graph edit distance (Richardeau et al., 2025).

### 5.3.2 JSON Data

**Privacy.** Privacy is a critical concern for generated JSON data, such as electronic health record resources in JSON format, that contains personally identifiable information or other sensitive data. In pipelines based on sanitization, privacy can be evaluated through established de-identification methods. Furthermore, the quality of privacy-enhancing post-processing is essential. This includes techniques like entity relexicalization, which ensures consistent surrogate substitutions to maintain coherence for downstream applications (Singh et al., 2025).

These document-level and corpus-level scores measure the accuracy of sensitive entity detection and removal. Since even a single leak of sensitive data can lead to a major privacy breach, researchers often adopt stricter definitions of recall (Singh et al., 2025). All-or-Nothing Recall represents this strictness by treating the completeness of detection within a document as a binary outcome. The value is 1 only if no instance of the target entity type is missed and it is 0 otherwise (Scaiano et al., 2016; Singh et al., 2025).

$$\text{Recall}_{\text{AON}}(t, d) = \mathbb{I}\Big(E_{t,d} \subseteq \hat{E}_{t,d}\Big) = \mathbb{I}\Big(E_{t,d} \setminus \hat{E}_{t,d} = \emptyset\Big).$$

In this equation, for an entity type $t$ and document $d$, $E_{t,d}$ represents the set of ground-truth sensitive entity instances of type $t$. Additionally, $\hat{E}_{t,d}$ denotes the set of instances that are correctly detected and redacted, and $\mathbb{I}(\cdot)$ is the indicator function. This document-level definition reflects the fact that a single missed instance can invalidate privacy protection for that specific entity type.

Clinical Model Consistency measures whether relexicalized data avoids introducing systematic biases when it is used in clinical decision-making models. These biases may include factors such as race, ethnicity, region, or age group. This metric also assesses whether the data preserves real-world model behavior (Singh et al., 2025). Specifically, if a clinical model achieves a performance metric value $X$ when trained and evaluated on real data, poor relexicalization may result in a metric of $X + \delta_X$:

$$|\delta_X| = |X_{\text{relex}} - X_{\text{real}}|.$$

A smaller value of $|\delta_X|$ indicates higher-quality relexicalization. This ensures that the downstream clinical utility and behavior remain close to those of models that are trained and evaluated on real data.

### 5.3.3 Log Data

**Privacy.** Evaluating privacy in logs by explicitly capturing the residual disclosure risk that remains in the released log text and the utility loss induced by anonymization. This metric reflects the practical trade-off between privacy and utility that is widely observed in log anonymization and task-aware anonymization benchmarking (Aghili et al., 2025; Loiseau et al., 2025). Inspired by evaluation practices in text anonymization, we organize privacy evaluation for logs along three complementary axes. These axes include exposure-oriented privacy risk, downstream utility degradation, and human-centered effectiveness (Pilán et al., 2022; Ren et al., 2025; Loiseau et al., 2025). Logs are naturally related to text anonymization because they are categorized as semi-structured text.

Sensitive Attribute Exposure measures an observable indicator of privacy risk by counting the number of sensitive attribute instances that remain in a log message after anonymization. Let $S_S$ be a predefined set of sensitive attribute types, such as IP addresses and MAC addresses. This set is derived from regulations,

empirical analysis, or industry consensus for software logs (Aghili et al., 2025). Let $\psi(\ell)$ be a sensitive-information detector that extracts a multiset of spans from the log message $\ell$, and let $\text{type}(x)$ return the attribute type of span $x$. We define the exposure as follows:

$$\text{Exposure}(\ell; S_S, \psi) = \sum_{x \in \psi(\ell)} \mathbb{I}\big(\text{type}(x) \in S_S\big).$$

Optionally, a dataset-level exposure score can be computed by averaging over a log subset $\mathcal{D} \subseteq \mathcal{L}$:

$$\text{Exposure}(\mathcal{D}; S_S, \psi) = \frac{1}{|\mathcal{D}|} \sum_{\ell \in \mathcal{D}} \text{Exposure}(\ell; S_S, \psi).$$

Exposure provides a lightweight and annotation-free signal of remaining identifiable spans. This aligns with identifier-removal effectiveness measures commonly used in the evaluation of anonymized text. When token or span-level ground truth annotations exist, researchers may further report precision, recall, and F1 scores for sensitive-span removal (Ren et al., 2025; Pilán et al., 2022). Note that Exposure is dependent on the detector. That is, it reflects both the quality of anonymization and the quality of detection, and it should be interpreted accordingly.

To quantify the utility cost of anonymization, we can measure the performance drop on a downstream task Downstream Model Performance Degradation. Such tasks include anomaly detection, failure diagnosis, and log parsing. Let $\mathcal{A}$ be a fixed learning algorithm and $\mathcal{M}$ be a task metric, such as F1 or Accuracy. A task-aware evaluation can be conducted by training the downstream model once on original training logs and then swapping the original and anonymized inputs at evaluation time. This follows the task-sensitivity perspective in anonymization benchmarking (Loiseau et al., 2025):

$$g = \text{Train}(\mathcal{A}, \mathcal{D}_{\text{orig}}^{\text{train}}),$$

$$\Delta_{\mathcal{M}} = \mathcal{M}\big(g; \mathcal{D}_{\text{test, orig}}\big) - \mathcal{M}\big(g; \mathcal{D}_{\text{test, anon}}\big).$$

Optionally, one may also evaluate a retraining-oriented setting by training on anonymized logs and comparing the results against the original training baseline, depending on specific deployment constraints.

The Qualitative Effectiveness Score captures expert judgment on how well anonymization protects privacy while keeping logs usable in practice. Examples of usability include readability for debugging and sufficiency for incident response. Such human-centered assessments are commonly collected through Likert-scale surveys in log-privacy studies (Aghili et al., 2025). The average score is calculated as:

$$\text{Score}_{\text{Qualitative}} = \frac{1}{|R|} \sum_{i=1}^{|R|} r_i,$$

where $R$ is the set of responses for a question, such as perceived privacy effectiveness or perceived utility preservation, and $r_i$ is the numerical value of response $i$, such as a value from 1 to 5.

### 5.4 Evaluation Practice Gap

Within the representative audit summarized in Table 5, privacy is the least consistently reported evaluation dimension, while utility is reported more often but with heterogeneous protocols across graph, JSON, and log settings. These patterns should be interpreted as recurring under-covered dimensions within the audited sample rather than as field-wide prevalence estimates.

**Privacy.** In Table 5, only three of the seventeen audited semi-structured works explicitly report privacy evaluation, and these are primarily governance-oriented papers. This property is typically measured only in

Table 5: Whether representative semi-structured generation methods explicitly evaluate each dimension in their experimental sections, together with their method family and functional role. ✓: explicitly evaluated; △: partially or indirectly covered; ×: not reported or not applicable.

| Representative Work | Family | Role | Validity | Fidelity | Diversity | Utility | Privacy |
|---|---|---|---|---|---|---|---|
| LLM4GraphGen (Yao et al., 2024) | Graph Data | Generation | ✓ | △ | ✓ | × | × |
| Generate-on-Graph (GoG) (Xu et al., 2024b) | Graph Data | Generation | × | × | × | △ | × |
| Ontology-grounded constrained decoding (Feng et al., 2024) | Graph Data | Generation | △ | △ | × | △ | × |
| GraphJudge (Huang et al., 2025a) | Graph Data | Evaluation | △ | △ | × | △ | × |
| GAG (Ji et al., 2025) | Graph Data | Generation | ✓ | ✓ | × | ✓ | × |
| GraphMaster (Du et al., 2025) | Graph Data | Generation | △ | ✓ | × | ✓ | × |
| PrivGraph (Yuan et al., 2023) | Graph Data | Governance | × | △ | × | △ | ✓ |
| JSON Schema discovery (Mior, 2024) | JSON Data | Generation | × | ✓ | × | △ | × |
| JSONSchemaBench (Geng et al., 2025) | JSON Data | Benchmark | ✓ | × | × | ✓ | × |
| Schema Reinforcement Learning (Lu et al., 2025) | JSON Data | Generation | ✓ | × | × | ✓ | × |
| ThinkJSON (Agarwal et al., 2025a) | JSON Data | Generation | ✓ | ✓ | × | × | × |
| RedactOR (Singh et al., 2025) | JSON Data | Governance | △ | △ | × | ✓ | ✓ |
| Automata-Based Steering (Luan et al., 2025) | JSON Data | Generation | × | × | ✓ | ✓ | × |
| LogBench (Li et al., 2024d) | Log Data | Benchmark | × | ✓ | × | × | × |
| AUCAD (Zhang et al., 2025a) | Log Data | Generation | △ | △ | × | △ | × |
| Protecting Privacy in Software Logs (Aghili et al., 2025) | Log Data | Governance | △ | △ | × | ✓ | ✓ |
| Log Generation using FSM-GFlowNets (Samanta, 2025) | Log Data | Generation | △ | ✓ | ✓ | ✓ | × |

studies where the main contribution is explicitly focused on preserving privacy. For example, this includes graph generation based on differential privacy or clinical de-identification. This pattern shows that most papers on general-purpose generation prioritize demonstrating feasibility and fidelity while leaving privacy risks unmeasured. A likely explanation is that rigorous privacy evaluation requires specific experimental commitments that many authors consider to be outside the scope of their work. These commitments include an explicit threat model as well as a concrete and reproducible protocol such as the use of differential privacy budgets with $\varepsilon$ and $\delta$ parameters. As a result, privacy measurement remains an optional feature that is limited to literature focused on privacy rather than becoming a standard reporting dimension for semi-structured generation.

**Utility.** Utility is reported more frequently than privacy in Table 5, with eight of the seventeen audited works providing explicit utility evaluation and six more offering partial coverage; however, its operationalization remains highly inconsistent across graph, JSON, and log settings. Graph synthesis papers typically measure utility through downstream learning performance such as training graph neural networks. Similarly, methods related to knowledge graph question answering treat end-task accuracy as the main evidence. In contrast, research on generating JSON and log data often emphasizes structural correctness or semantic similarity without systematically showing improvements in downstream applications. This variety shows that utility is naturally difficult to standardize because it depends on the task and requires complex experimental controls. These controls include data mixing strategies, fixed training and testing splits, and specific downstream models to allow for fair comparisons. As a result, many studies choose to use validity or fidelity measures that are easier to calculate instead of using a consistent and complete utility protocol. This leads to evidence that is fragmented and can only be compared partially across different research areas.

## 5.5   Usage

**Graph Construction and Analysis.**   LLM-guided graph synthesis supports four main applications and an emerging evaluation method.

First, in validator-in-the-loop knowledge graph curation, candidate triples such as facts proposed by extraction pipelines or completion models are verified by validators based on LLMs. These systems perform consistency checks and use retrieval to verify information against external sources. This process filters noisy statements and reduces the need for large-scale human validation (Boylan et al., 2024).

Second, for text-conditioned graph simulation, frameworks such as the GraphAgent-Generator, which is also known as GAG, use large-scale agent simulations based on LLMs to create dynamic social graphs with text attributes. These approaches improve structural accuracy at both small and large scales and can handle large networks through parallel processing (Ji et al., 2025).

Third, synthetic graphs and latent structures derived from LLMs serve as data augmentation for downstream graph neural networks, which are often called GNNs. For example, DemoGraph uses black-box LLMs to generate latent knowledge graphs from text prompts. It integrates these structures into the original training graph to solve problems with limited data and noise in real-world applications. The value of this augmentation is usually measured using performance metrics from downstream tasks (Feng et al., 2025b; Ji et al., 2025; Le et al., 2025; Zhong et al., 2024; Le et al., 2024b).

Finally, graph-structured hallucination analysis examines the reliability of LLMs when they produce structured outputs. A study by Richardeau et al. (2025) prompts these models to reproduce standard real-world networks such as Zachary's karate club and Les Miserables. The study also asks models to generate random graphs like Erdos Renyi graphs. Researchers measure graph hallucinations using structural metrics including degree sequence statistics, spectral distances, and the Graph Atlas Distance which is based on graph edits. This work connects structural differences with external leaderboards for hallucinations.

**JSON for Tool Use and Schema Automation.**   LLMs frequently generate JSON as a structured interface for tool invocation, workflow automation, and data exchange between different services. These are contexts where outputs must strictly follow predefined schemas. Constrained decoding is also known as grammar-constrained generation or schema-constrained generation. This process enforces adherence to specifications such as JSON Schema, regular expressions, or context-free grammars during the decoding stage. This mechanism reduces parse failures and improves the reliability of outputs that are consumable by machines in agent pipelines. Recent benchmarks on real-world schemas further describe the trade-offs between efficiency, coverage, and quality in these systems for constrained decoding (Geng et al., 2025).

JavaScript Object Notation is widely used for data interchange across application programming interfaces and data integration workflows. When schemas are missing or incomplete, large language models can improve automatically discovered schemas for this format. These models generate natural language descriptions and assign meaningful names to components that are reusable. They also filter properties that were inferred but are not useful. This process helps with validation and the use of the data in later stages (Mior, 2024).

**Log for Observability and Developer Assistance.**   Semi-structured log generation is used to create realistic and parsable telemetry for incident drills and anomaly benchmarks. This technology also assists developers with automatic logging tasks such as decisions about placement, logging levels, and message content. Furthermore, it helps share exemplars that preserve privacy. However, recent benchmarks reveal critical limitations. While LLMs often produce log messages that seem correct semantically, they frequently fail to meet strict quality requirements and project conventions at a large-scale. LogBench organizes the evaluation of log statement generation and identifies significant gaps under common automatic metrics and generalization settings (Li et al., 2024d). Similarly, AL Bench emphasizes real-world constraints through dynamic evaluation. It demonstrates that code containing generated log statements often fails to compile and that log outputs at runtime can differ significantly from the ground truth (Tan et al., 2025).

These findings encourage adaptation for specific domains instead of using generic prompting. To address this, research on the AUCAD framework shows that training on alignment datasets created specifically for log

generation can perform much better than generic solutions based on LLMs. This provides a practical path for adoption in software engineering (Zhang et al., 2025a).

# 6 Vision–Language Data

Vision-language data consists of multimodal artifacts where visual signals are naturally paired with linguistic descriptions. This category forms the foundational training corpus for modern Vision-Language Models and Multimodal LLMs. Recent methodologies increasingly use both real and synthetic data to reduce costs, privacy concerns, and the scarcity associated with large-scale aligned corpora (Mohammadkhani et al., 2025). Specifically, vision-language corpora include image-text pairs, interleaved documents such as webpages containing mixed images and text, and video-text sequences. All of these formats provide paired visual content and textual descriptions (Sun et al., 2024).

In the context of LLMs, multimodal models such as Emu (Sun et al., 2024) encode visual signals into continuous embeddings and interleave them with text tokens. These models then train a single transformer using a unified autoregressive objective, which involves predicting the next text token or regressing the next visual embedding within the multimodal sequence (Sun et al., 2024). Recent architectures such as Emu3 further advance this paradigm by tokenizing images, text, and videos into a shared discrete token space. This approach applies pure next-token prediction over the mixed multimodal stream (Wang et al., 2024c).

Cross-modal alignment operates at multiple levels of granularity. Fundamentally, vision-language data provides weak or global pairing between a visual unit such as an image or video clip and a textual unit such as a caption, transcript, or surrounding narrative. Beyond this global alignment, certain datasets encode fine-grained grounding signals that connect specific regions or spatio-temporal segments to linguistic entities. This allows models to connect localized perception with compositional semantic reasoning. Representative examples include region-to-phrase correspondences with bounding boxes in Flickr30k Entities (Plummer et al., 2015), dense region descriptions and object-level grounding in Visual Genome (Krishna et al., 2016), and spatio-temporal tube grounding within the VidSTG benchmark for the Spatio-Temporal Video Grounding task (Zhang et al., 2020).

## 6.1 Generation Methods

### 6.1.1 Image-Text Data

LLM-based image-text generation can be broadly categorized into two paradigms based on the generation mechanism. The first paradigm is native autoregressive generation. In this approach, the multimodal model produces visual representations within a unified token or embedding stream. This includes methods such as continuous visual-embedding regression in Emu or discrete multimodal token prediction in Emu3 (Sun et al., 2024; Wang et al., 2024c). The second paradigm is external diffusion control. In this setup, the LLM functions as a controller for a separate diffusion backbone. The model iteratively diagnoses mismatches and guides edits to improve how well the output follows the prompt (Wu et al., 2023).

**Native Autoregressive Generation.** This paradigm treats multimodal generation as a sequence modeling task using mixed visual and textual representations. In this framework, images and other data types are represented as continuous embeddings or discrete tokens. These elements are combined with text within a single autoregressive stream (Sun et al., 2024; Wang et al., 2024c; Team, 2025).

For example, Emu encodes images into visual embeddings that are mixed with text tokens. The model uses a transformer to predict the next text token or estimate the next visual embedding in a unified sequence (Sun et al., 2024). Emu3 further develops this approach by converting images, text, and videos into a shared space of discrete tokens. This allows the model to work entirely through next-token prediction over the mixed sequence (Wang et al., 2024c).

These unified designs support both multimodal understanding and generation within one autoregressive model. By starting with different multimodal prefixes, a single model can perform various tasks such as image captioning, visual question answering, and image generation (Sun et al., 2024; Wang et al., 2024c).

Team (2025) also shows the effectiveness of early-fusion modeling when applied directly to long sequences of interleaved image and text tokens.

**External Diffusion Control.**   In contrast to generating visual representations directly within the LLM, this paradigm uses the model as a high-level controller. This controller routes intent and conditioning signals to specialized generative tools such as diffusion models (Pan et al., 2024; Koh et al., 2023).

Systems such as Kosmos-G (Pan et al., 2024) first align the outputs of a multimodal model to a CLIP-anchored conditioning interface through supervised alignment. They then apply score-distillation instruction tuning where a frozen diffusion decoder provides training signals. This process enables the controller to produce representations that guide high-fidelity image synthesis.

Similarly, GILL (Koh et al., 2023) connects a frozen text-only LLM with pretrained image generation backbones using lightweight mapping modules. These modules translate the hidden representations of the model into the embedding space of the generator. This allows downstream decoders such as diffusion-based generators to render the requested images.

This modular design separates intent understanding and planning from high-fidelity rendering. In this setup, the LLM manages the planning while a diffusion backbone executes the rendering. This separation enables precise control through an explicit conditioning interface. In systems like Kosmos-G, the visual generator can be upgraded or replaced while the controller remains largely the same. In mapping-based designs such as GILL, replacing the generator typically requires retraining the bridging module rather than the entire LLM (Pan et al., 2024; Koh et al., 2023).

In summary, while native autoregressive generation provides a unified space for both reasoning and rendering, external diffusion control makes better use of specialized vision models and established generation tools.

### 6.1.2   Video-Text Modeling

Video-text generation extends the scope of LLM synthesis from static visual domains to spatiotemporal media. Similar to the image domain, contemporary methodologies generally fall into two distinct paradigms. The first is native spatiotemporal generation, where the model directly synthesizes video and often audio representations within a unified multimodal sequence. The second is planner-based diffusion control, where the LLM manages high-level video structure for execution by a separate rendering engine.

**Native Spatiotemporal Generation.**   Approaches such as VideoPoet (Kondratyuk et al., 2024) and Emu3 (Wang et al., 2024c) treat video generation as a native language modeling task over spatiotemporal tokens. In the case of VideoPoet, this also includes audio. VideoPoet uses a decoder-only transformer to process multimodal inputs including images, video clips, text, and audio. The model is trained using multimodal generative objectives in an autoregressive manner (Kondratyuk et al., 2024). Similarly, Emu3 converts video frames into a discrete latent space and learns to predict the next token. This approach effectively unifies video understanding and generation under a single backbone (Wang et al., 2024c).

By representing video and sometimes audio as discrete tokens and training a decoder-only transformer with autoregressive next-token objectives, these methods unify multimodal conditioning and generation within a language-model style backbone. This strategy avoids the use of diffusion-based generation heads. In practice, these systems still rely on specific tokenizers or decoders and sometimes use additional modules such as token-space super-resolution. Understanding capabilities such as captioning depend on the mixture of tasks and the post-training setup. Furthermore, the models learn long-range temporal dependencies, such as motion coherence and scene continuity, end-to-end through sequence modeling.

**Planner-based Diffusion Control.**   In planner-based architectures such as FlowZero (Lu et al., 2023), LVD (Lian et al., 2024), and VideoDirectorGPT (Lin et al., 2024), the LLM functions as a director rather than a renderer. It translates natural language prompts into structured and interpretable intermediate representations. These representations include dynamic scene syntaxes or spatiotemporal layouts such as per-frame scene descriptions, object layouts, bounding-box trajectories, and background motion patterns. These plans then condition a separate diffusion-based video generator.

For instance, FlowZero uses a dynamic scene syntax generated by an LLM to guide frame-wise diffusion and improve temporal smoothness. This includes coherent object motion as well as controllable background and camera motion patterns (Lu et al., 2023). LVD similarly uses the LLM as a spatiotemporal planner that outputs dynamic scene layouts, which are typically frame-consistent bounding-box sequences. These layouts are injected into the video diffusion model to enforce spatial relations and motion consistency (Lian et al., 2024). VideoDirectorGPT further decomposes long prompts into editable multi-scene video plans and consistency groupings. It generates scene-level and frame-level layouts before invoking downstream diffusion modules, which enables multi-scene narrative structure and character consistency (Lin et al., 2024).

By separating high-level planning from low-level frame synthesis, planner-based control provides transparent and editable handles over visual content and motion dynamics. This paradigm also offers natural anchor points for integrating external constraints such as timeline editing, user edits, or guardrails at the planning stage before invoking the diffusion generator.

Across both paradigms, there is a converging trend to treat structural integrity and provenance as native components of the generation pipeline rather than retroactive additions. First, temporal alignment and long-horizon consistency are increasingly treated as primary objectives in long-form video generation. Second, to address generative authenticity, researchers are integrating provenance mechanisms such as Content Credentials with cryptographically signed metadata following the C2PA standard (Coalition for Content Provenance and Authenticity, 2025). They are also including robust invisible watermarking for video. This involves watermarking designed for latent video diffusion models (Jang et al., 2025) as well as visual-audio watermarking for manipulation localization and copyright protection (Zhang et al., 2024e). These tools enable machine-checkable attribution and manipulation tracing. Consequently, modern video-text systems are evolving from loosely coupled toolchains into more integrated frameworks that jointly optimize expressivity, structural consistency, and governance.

### 6.2 Quality Metrics for Vision-Language Data

#### 6.2.1 Image-Text

**Validity.** Validity ensures that generated vision-language content is both well-formed and verifiable. This means that the content can be parsed under a predefined machine-readable structure and checked for basic consistency constraints. In practice, a valid output must follow the target schema and maintain consistent internal references. This includes features such as stable identifiers or pointers between turns when the format requires them. Common structural failures such as malformed schemas can be reduced through formally constrained decoding. For instance, grammar-based engines such as DOMINO align grammatical constraints with the subword vocabulary of the model to guarantee the structural correctness of machine-readable outputs (Beurer-Kellner et al., 2024).

Given a set of vision-language generations $\mathcal{M}_{\text{gen}} = \{m_1, \ldots, m_N\}$ and a schema $S$, we define the Well-Formed Rate as:

$$\text{WFR} = \frac{1}{|\mathcal{M}_{\text{gen}}|} \sum_{m \in \mathcal{M}_{\text{gen}}} V(m, S),$$

where $V(m, S) \in \{0, 1\}$ indicates whether $m$ conforms to $S$.

**Fidelity.** Semantic-level fidelity measures the logical and contextual alignment between text and non-text modalities within a generated interleaved sequence $m = \{(t_\ell, v_\ell)\}_{\ell=1}^L$. In this notation, $\ell$ represents the individual steps within a single vision-language sample. A common approach is to score each pair using an alignment function $\mathcal{S}_{\text{align}}$. In practice, this function is implemented in one of two ways. The first uses a strong multimodal judge model such as GPT-4o. The second uses embedding-space alignment, such as cosine similarity between CLIP text and image features, which is also known as text alignment (Ham et al., 2024). Step-wise alignment can then be aggregated as follows:

$$\text{ITA}(m) = \frac{1}{L} \sum_{\ell=1}^L \mathcal{S}_{\text{align}}(t_\ell, v_\ell).$$

The CoMM evaluation framework also reports an Image-Text Alignment score using strong judge models for a similar purpose (Chen et al., 2025c).

Beyond global alignment, instance-level fidelity measures how accurately specific attributes are preserved. Following standard protocols for evaluating subject-driven generation (Ruiz et al., 2023), Subject Fidelity can be computed between a source subject image $V_{\text{subj}}$ and a generated image $V_{\text{gen}}$. This is typically calculated using an image embedding extractor $E_{\text{img}}(\cdot)$ such as the CLIP image encoder or DINO:

$$\text{Fidelity}_{\text{subj}} = \text{sim}\big(E_{\text{img}}(V_{\text{gen}}),\, E_{\text{img}}(V_{\text{subj}})\big),$$

In this equation, $\text{sim}(\cdot, \cdot)$ is typically cosine similarity. Prompt Fidelity, which refers to how well the model follows the prompt, can be captured using CLIP-style alignment between text and images (Ruiz et al., 2023):

$$\text{Fidelity}_{\text{prompt}} = \text{sim}\big(E_{\text{img}}(V_{\text{gen}}),\, E_{\text{text}}(T_{\text{prompt}})\big).$$

When reference data are available, standard text metrics such as ROUGE and METEOR are commonly reported. For image evaluation, metrics such as the Fréchet Inception Distance and the Inception Score are widely used to measure image quality. The Structural Similarity Index is used for style consistency, and the Peak Signal-to-Noise Ratio is used for reconstruction when applicable. These metrics are typically measured against a reference text $T_{\text{ref}}$ or a reference image $V_{\text{ref}}$ (Chen et al., 2025c).

The Caption Hallucination Assessment with Image Relevance (CHAIR) metric is commonly used to measure how often models mention objects that are not in the image (Rohrbach et al., 2019). Mentions are extracted from captions and compared to eighty object categories from the Microsoft Common Objects in Context dataset. A mention is considered a hallucination if it does not appear in the ground truth labels.

The metric provides two scores. The object level rate is the fraction of all mentions that are hallucinated. The sentence-level rate is the fraction of all sentences with at least one hallucination. Lower scores indicate higher grounded fidelity.

$$\text{CHAIR}_i = \frac{\text{Number of hallucinated objects}}{\text{Total object mentions}}$$

$$\text{CHAIR}_s = \frac{\text{Number of sentences with hallucinations}}{\text{Total number of sentences}}$$

**Utility.** Utility measures the value of a model for downstream tasks such as image captioning and visual question answering. It also applies to tool-augmented workflows. This is typically assessed through performance on downstream tasks and evaluations by automatic or human judges. Unified early-fusion models such as Chameleon demonstrate interleaved reasoning and generation over mixed-modal sequences (Team, 2025). Similarly, Anole provides an open-source autoregressive model along with a training framework (Chern et al., 2024). Interleaved image and text training data such as CoMM (Chen et al., 2025c) and instruction-driven multi-turn dialogues such as InterSyn (Feng et al., 2025a) further support supervised instruction tuning and evaluation.

Let $D = \{(q_i, a_i)\}_{i=1}^N$ denote a benchmark dataset consisting of $N$ question–answer pairs, where $q_i$ is the $i$th question and $a_i$ is its corresponding ground truth answer. The notation $|D|$ denotes the size of the dataset, which is equal to $N$. Let $M$ denote the model or inference function that maps an input question $q_i$ to a predicted answer.

An indicator function is used, which equals one if its argument is true and zero otherwise. The equality predicate $M(q_i) = a_i$ indicates an exact-match comparison between the predicted answer and the reference answer. This process may optionally be performed after applying a standard normalization procedure, such as lowercasing and punctuation stripping, when the task is open-ended. The accuracy for question answering is then computed as the average indicator value over all samples in the dataset:

$$\text{Acc}_{\text{QuestionAnswering}} = \frac{1}{N} \sum_{i=1}^{N} \mathbb{I}(M(q_i) = a_i)$$

For human evaluation of utility, a pairwise preference protocol is commonly adopted, following the relative evaluation setting in Chameleon (Team, 2025). Given the same input, annotators are shown two anonymized responses from a target model $M$ and a baseline $B$ in randomized order. The annotators then select one of the following options: $M$ is better, $B$ is better, or about the same. A win is defined as the case where annotators prefer the output of $M$ due to better task fulfillment and usefulness. Similarly, a loss occurs when $B$ is preferred, and a tie occurs when the two outputs are judged to be comparable. The win rate is computed by awarding one point for a win and half a point for a tie (Team, 2025):

$$\text{Win}_{\text{Rate}} = \frac{W + 0.5 \times T}{W + T + L},$$

where $W$, $T$, and $L$ denote the number of wins, ties, and losses of $M$ against $B$, respectively.

**Diversity.** Diversity measures the range of entities, attributes, styles, and layouts while preserving the agreement between text and images. It is examined at three levels. The first level is distributional diversity over the entire set. The second level is intra-sequence diversity within a single sample. The third level is judge-based diversity over multimodal outputs.

From an architectural perspective, MM-Interleaved improves multi-image interleaved generation by synchronizing access to detailed visual features during the generation process (Tian et al., 2024). Chameleon uses a unified token space (Team, 2025). InterSyn increases coverage with multi-turn dialogues driven by instructions. These dialogues are created through a process known as iterative refinement during data creation (Feng et al., 2025a).

Let $\mathcal{V}$ be the set of generated images. Following Salimans et al. (2016), sample quality and diversity are evaluated using the Inception Score. Specifically, a pretrained Inception classifier is applied to each generated image $v \in \mathcal{V}$ to obtain a conditional label distribution $p(y \mid v)$.

Images that contain meaningful and recognizable objects tend to yield low entropy conditional label distributions where the classifier shows high confidence. In contrast, a generator that avoids mode collapse should produce varied images such that the marginal label distribution $p(y) = \mathbb{E}_{v \in \mathcal{V}}[p(y \mid v)]$ has high entropy. This ensures that predictions are spread across many different classes (Salimans et al., 2016). Combining these two goals, the Inception Score is defined as:

$$\text{IS}(\mathcal{V}) = \exp\left( \mathbb{E}_{v \in \mathcal{V}}\left[ D_{\text{KL}}(p(y \mid v) \,\|\, p(y)) \right] \right).$$

Equivalently, $\log \text{IS}(\mathcal{V}) = H(p(y)) - \mathbb{E}_{v \in \mathcal{V}}[H(p(y \mid v))]$. The second term encourages sharpness and recognizability through low conditional entropy. The first term encourages diversity by rewarding marginal distributions with high entropy and $D_{KL}$ means KL Divergence. This represents broad coverage over the semantic categories predicted by the Inception classifier (Salimans et al., 2016).

Let $m_v = \{v_\ell\}_{\ell=1}^L$ be a visual sequence such as a series of generated images or video frames. In this sequence $v_\ell$ is the visual element at position $\ell$ and $L$ represents the total length of the sequence. Let $E$ be a feature extractor that transforms a visual input into an embedding vector. This extractor can be an image encoder such as Contrastive Language Image Pretraining or Self Distillation with no Labels. The term sim refers to a similarity function between embeddings and this value is typically determined by cosine similarity.

Intra-Sequence Diversity is defined as one minus the average pairwise similarity among all unordered pairs in the sequence:

$$\text{Div}_{\text{seq}}(m_v) = 1 - \frac{2}{L(L-1)} \sum_{\ell=1}^{L} \sum_{r=\ell+1}^{L} \text{sim}\big( E(v_\ell), E(v_r) \big).$$

Let $\mathcal{M}_{\text{generations}}$ be a set of multimodal generations, and let $|\mathcal{M}_{\text{generations}}|$ represent the total number of items in that set. For each generated sample $m$ in the set of generations, the term $\mathcal{J}_m$ denotes the diversity score. This score is assigned by a judge model, a human rater, or an automatic rater to evaluate the diversity of the output. Judge-based diversity is then aggregated by averaging these scores over the full set of generations, yielding the final diversity score.

$$\text{Score}_{\text{diversity}} = \frac{1}{|\mathcal{M}_{\text{generations}}|} \sum_{m \in \mathcal{M}_{\text{generations}}} \mathcal{J}_m$$

### 6.2.2 Video-Text

**Validity.** For video-text samples, we define validity using a set of verifiable sub-properties. These properties include schema-level well-formedness, temporal correctness, and cross-modal binding consistency. Let $\mathcal{M}$ be the set of $N$ generated video and text samples where the size of the set is $N$. Each sample $m_i$ consists of a video $v_i$ and a paired text $t_i$ and the number of modalities is two.

Let $\mathbb{S}$ denote a machine-readable schema, such as a typed JavaScript Object Notation or Yet Another Markup Language schema with field constraints used to validate structured outputs. The function conforms returns whether a sample $m_i$ adheres to schema $\mathbb{S}$. Let $\mathbb{I}$ be the indicator function, which returns one if its argument is true and zero otherwise. The Schema Adherence Rate is defined as the fraction of generated samples that pass schema validation:

$$\text{SchemaAdherenceRate} = \frac{1}{|\mathcal{M}|} \sum_{m_i \in \mathcal{M}} \mathbb{I}[\text{conforms}(m_i, \mathbb{S}) = \text{true}]$$

In the evaluation of temporal correctness, established protocols from temporal language grounding are commonly followed, such as TALL (Gao et al., 2017). Temporal correctness is typically measured using recall at various Intersection over Union thresholds. In this framework, $T_{\text{pred},i}^{(j)}$ represents the $j$-th ranked predicted interval for sample $i$ based on the model score, and $T_{\text{gt},i}$ represents the ground truth interval. The Recall @ K metric for a given Intersection over Union threshold $\delta$ is defined as follows:

$$\text{R@K}(\delta) = \frac{1}{|\mathcal{M}|} \sum_{m_i \in \mathcal{M}} \mathbb{I}\left( \max_{1 \leq j \leq K} \text{IoU}\left(T_{\text{pred},i}^{(j)}, T_{\text{gt},i}\right) \geq \delta \right).$$

In practice, this metric is often reported for standard choices such as $K$ values of 1, 5, or 10 and threshold values of 0.3, 0.5, or 0.7. Additionally, a comprehensive temporal score can be obtained by taking the average across a specific set of thresholds $\Delta$:

$$\text{R@K-Avg} = \frac{1}{|\Delta|} \sum_{\delta \in \Delta} \text{R@K}(\delta).$$

Regarding cross-modal binding consistency, cross-modal binding can be treated as a validity property that can be examined through representation agreement. Let $f_T$ and $f_V$ be text and video encoders, and let $\phi_{\text{sim}}$ represent cosine similarity. For each sample, it is assumed that a grounded video segment $v_i^{\text{seg}}$ is referenced by text $t_i$. The Cross-Modal Consistency score is then defined as follows:

$$\text{CMC} = \frac{1}{|\mathcal{M}|} \sum_{m_i \in \mathcal{M}} \phi_{\text{sim}}\left(f_T(t_i), f_V(v_i^{\text{seg}})\right).$$

This equation naturally extends to aligned speech or text regions by using automatic speech recognition transcripts or optical character recognition snippets when they are available.

**Fidelity.** Regarding video fidelity, this property reflects both spatio-temporal coherence and visual realism. Let $F$ be the number of frames in $v_i$, and let $\tau$ represent the frame indices. To evaluate object identity consistency over tracked object regions $\{o_{i,\tau}\}$, a visual feature extractor $f_E$ such as CLIP and a similarity function $\phi_{\text{sim}}$ are introduced. The identity consistency score is defined as follows:

$$\mathcal{F}_{\text{ID}}(m_i) = \frac{1}{F-1} \sum_{\tau=1}^{F-1} \phi_{\text{sim}}\left(f_E(o_{i,\tau}), f_E(o_{i,\tau+1})\right).$$

Holistic realism and distributional faithfulness are often summarized by the Frechet Video Distance. This metric was originally proposed for the evaluation of generative video models (Unterthiner et al., 2019). It is now widely reported in modern text-to-video evaluations including VideoPoet and various planner-based systems (Kondratyuk et al., 2024; Lin et al., 2024).

Additionally, frame-level or clip-level quality scores from specialized predictors can be aggregated to define a general video quality metric:

$$\mathcal{F}_{\text{VQ}}(m_i) = \frac{1}{F} \sum_{\tau=1}^{F} \Psi(v_{i,\tau}),$$

In this equation, $\Psi$ represents one or more automated evaluators that target dimensions such as imaging and aesthetics. These evaluators are typically organized in standardized suites such as VBench (Huang et al., 2024b).

**Utility.** Utility represents how reliably synthetic video and text data can support controllable generation and downstream usage. For planner-style pipelines that generate structured plans or prompts, such as VideoDirectorGPT (Lin et al., 2024), let $P$ represent a set of plans or prompts and $v_{\text{out}}^{(j)}$ denote the output video conditioned on a specific plan $p_j$. Inspired by planner-based pipelines such as VideoDirectorGPT, a controllability-oriented Task Success Rate can be defined as follows:

$$\mathcal{U}_{\text{Control}} = \frac{1}{|P|} \sum_{p_j \in P} \mathbb{I}\big(\text{eval}_{\text{task}}(v_{\text{out}}^{(j)}, p_j) = \text{true}\big),$$

In this formulation, the task evaluation function can be implemented through automatic checkers, learned judges, or human evaluation. The choice of method depends on the controllability constraints that are encoded by the plan $p_j$.

For cases involving specific styles or functions, an embedding-based proxy for usefulness can be used:

$$\mathcal{U}_{\text{Func}} = \phi_{\text{sim}}\big(f_T(t_{\text{style}}), f_V(v_{\text{out}})\big),$$

This metric captures whether the generated video matches a target style or condition within a shared representation space.

**Diversity.** Regarding video diversity, this metric reflects both motion variety and semantic breadth. Let $\mathcal{V} = \{v \mid (v,t) \in \mathcal{M}\}$ denote the set of generated videos.

To evaluate dynamic variety, motion magnitude can be measured using a motion score based on optical flow (Teed & Deng, 2020). This score is defined using the following equation:

$$\text{motion\_score}(v) = \frac{1}{T-1} \sum_{\tau=1}^{T-1} \frac{1}{|\Omega|} \sum_{\mathbf{x} \in \Omega} \|\mathbf{f}_{\tau \to \tau+1}(\mathbf{x})\|_2,$$

In this formulation, $f_{\tau \to \tau+1}$ represents the optical flow field from frame $\tau$ to frame $t+1$ over the pixel domain $\Omega$, and $T$ represents the number of frames. In practice, optical flow is typically computed using methods such as RAFT, and the average flow magnitude is aggregated over all frames to obtain the motion score for each video (Liao et al., 2024).

In terms of semantic diversity, a metric analogous to the Inception Score can be used for videos. This approach is widely adopted in the evaluation of video generation. Following the standard Inception Score formulation (Salimans et al., 2016), the metric can be defined as follows:

$$\mathcal{D}_{\text{sem}} = \exp\big(\mathbb{E}_{v \sim \mathcal{V}} \left[ D_{KL}\big(p(y|v) \,\|\, p(y)\big)\right]\big),$$

In this definition, $p(y|v)$ is the label distribution predicted by a pretrained classifier (Saito et al., 2017). The term $p(y)$ represents the marginal label distribution.

### 6.3 Trustworthy Metrics for Vision-Language Data

#### 6.3.1 Image-Text

**Safety.** Safety aims to prevent the generation of harmful content across different modalities. It also focuses on reducing multimodal attack surfaces such as image-based prompt attacks and jailbreaking conditioned on images (Liu et al., 2024c). The MM-SafetyBench framework investigates these prompt attacks and demonstrates that images relevant to a query can significantly increase the success rate of an attack. This finding supports the hypothesis that query-relevant images activate the alignment module for vision and language. Since this module is often trained without specific safety constraints, its activation can weaken the ability of the model to identify harmful requests (Liu et al., 2024c).

Furthermore, the benchmark notes that a low attack success rate is not always clear. It might indicate that the model is properly aligned for safety, or it could simply mean that the model fails to understand the malicious query. To address this uncertainty, MM-SafetyBench emphasizes refusal behavior. This approach helps to distinguish between a model that recognizes and refuses a harmful request and one that simply fails to comply because of a lack of understanding. In practice, achieving a low attack success rate while keeping the expected refusal behavior remains a difficult task (Liu et al., 2024c).

The benchmark also shows that adding a safety prompt can significantly lower attack success rates for models that are better at following instructions. For other models, these improvements are smaller. The authors suggest that the effectiveness of safety prompts depends on how well the underlying model follows instructions (Liu et al., 2024c).

Following MM-SafetyBench, Attack Success Rate (ASR) and Refusal Rate (RR) are commonly reported on a query set $D$:

$$\text{ASR} = \frac{\sum_{q \in D} I(q)}{|D|}, \qquad \text{RR} = \frac{\sum_{q \in D} R(q)}{|D|}.$$

Here $I(q) = 1$ if and only if the model engages with the disallowed request under query $q$, for example by providing prohibited content or actionable instructions; otherwise $I(q) = 0$ (Liu et al., 2024c). Similarly, $R(q) = 1$ if and only if the model begins with a refusal to satisfy the unsafe query, using the refusal definition in MM-SafetyBench; otherwise $R(q) = 0$ (Liu et al., 2024c).

In implementation, refusal detection can be instantiated with a predefined refusal-pattern matcher, and can be further strengthened with a judge $\Phi_S$ for more robust refusal identification. The metric RR is reported over all queries in $D$. To quantify over-sensitivity at the system level, a benign subset $D_{\text{benign}}$ can additionally be evaluated when constructed separately, and $\text{RR}_{\text{benign}}$, the refusal rate on benign queries, can be reported. This provides a practical system-level proxy inspired by safety-awareness benchmarks such as MMSafeAware (Wang et al., 2025), which include benign subsets to capture harmless requests that are mistakenly treated as unsafe. MMSafeAware focuses on safety awareness and recognition, whereas $\text{RR}_{\text{benign}}$ captures the downstream refusal behavior under benign inputs.

**Provenance.** Regarding provenance, this property tracks the verifiable origin, attribution claims, and modification history of multimodal outputs. C2PA version 2.2 provides signed content credentials that support an auditable provenance chain (Coalition for Content Provenance and Authenticity, 2025). In addition, SynthID-Image enables invisible watermarking for imagery generated by artificial intelligence to facilitate detection and traceability (Gowal et al., 2025; Google DeepMind, 2025). The credential validation rate over a set of generated multimodal samples is defined as follows:

$$\text{Rate}_{\text{validate}} = \frac{1}{|\mathcal{M}_{\text{gen}}|} \sum_{m \in \mathcal{M}_{\text{gen}}} V_{\text{c2pa}}(m),$$

In this definition, the indicator function $V_{\text{c2pa}}$ takes a value of either zero or one. The value is zero when no C2PA manifest or credentials are present. The value is one when the validator returns a manifest state that is at least Valid, or Trusted under a stricter setting, depending on the chosen trust model. To clarify, the manifest states of Well-Formed, Valid, and Trusted reflect three distinct properties. These include the

well-formedness and cryptographic integrity of the manifest, the verification of the signature, and the trust status of the signing credential under the adopted trust configuration (Coalition for Content Provenance and Authenticity, 2025).

### 6.3.2 Video-Text

**Safety.** The Harmful Content Rate measures the failure to prevent video generation that violates policies during adversarial evaluation. This approach is inspired by T2VSafetyBench, which constructs malicious prompts such as real-world prompts, prompts generated by GPT-4, and prompts based on jailbreak attacks. That framework evaluates videos using a GPT-4-based assessment together with manual review for validation. The Harmful Content Rate is defined through a safety evaluation pipeline $\Phi_S$. This pipeline may involve LLM-based judging together with optional human review and detector-based checks (Miao et al., 2024).

This metric is defined as follows:

$$\text{HCR} = \frac{|\{v \in \mathcal{V}_{\text{adv}} \mid \text{is\_unsafe}(v, \Phi_S) = \text{true}\}|}{|\mathcal{V}_{\text{adv}}|},$$

In this equation, $\mathcal{V}_{\text{adv}}$ represents the set of all successfully generated videos under adversarial prompts. Beyond measuring the number of unsafe generations, T2VSafetyBench reports three findings that are useful for system-level safety evaluation. First, no single model performs well across all safety aspects. Second, assessments by GPT-4 correlate well with manual reviews, which supports the use of large-scale automatic judging. Finally, a trade-off exists between model capability and safety. This suggests that safety risks may increase as video generation technology improves (Miao et al., 2024).

**Provenance.** Watermark-based provenance evaluates robustness as well as the trade-off between payload and quality for generated videos. When an original watermark $w$ consisting of $L$ bits, an attack $\mathcal{A}$, and a recovery procedure $\mathcal{R}$ are given, robustness is measured using bit accuracy. For a generated video $v$, this quantity is defined as follows:

$$\mathcal{P}_{\text{Rob}} = 1 - \frac{d_H\big(w, \mathcal{R}(\mathcal{A}(v))\big)}{L}.$$

Payload and fidelity cost are defined using the following equations:

$$\mathcal{P}_{\text{Load}} = L, \qquad \mathcal{P}_{\text{Cost}} = \Delta\big(V_{\text{ori}}, v\big).$$

In these expressions, $\mathcal{P}_{\text{Load}}$ represents the payload per video in bits. The variable $V_{\text{ori}}$ represents the corresponding baseline video that was generated without a watermark using the same prompt and seed. The term $\Delta$ quantifies the perceptual distortion introduced by the watermark embedding process. For example, this quantity can be computed by averaging frame-wise Learned Perceptual Image Patch Similarity over sampled frames to capture frame-level distortion. Frechet Video Distance can also be used to assess quality degradation at the distribution level.

These estimates can be used together with signed credentials and invisible watermarking for video content. Specific examples include LVMark for latent video diffusion model watermarking and V2A-Mark for visual and audio watermarking with tamper localization. These tools support traceability and detection in downstream tasks. Furthermore, they can complement provenance mechanisms that are cryptographically verifiable such as C2PA (Coalition for Content Provenance and Authenticity, 2025; Jang et al., 2025; Zhang et al., 2024e).

### 6.4 Evaluation Practice Gap

We analyze representative methods from Section 6.1 and categorize their reported evaluation protocols in Table 6.

**Diversity.** Within the representative audit in Table 6, diversity is explicitly evaluated in only one of the six audited vision-language works and is therefore less consistently reported than fidelity in this sample. A practical reason is that commonly reported diversity proxies such as the Inception Score combine quality and coverage. However, stronger measures such as intra-sequence embedding diversity, judge-based diversity, or

Table 6: Whether representative vision–language methods explicitly evaluate each dimension in their experimental sections, together with their method family and functional role. ✓: explicitly evaluated; △: partially or indirectly covered; ×: not reported or not applicable.

| Representative Work | Family | Role | Validity | Fidelity | Diversity | Utility | Safety | Provenance |
|---|---|---|---|---|---|---|---|---|
| Kosmos-G (Pan et al., 2024) | Image–Text External Diffusion Control | Generation | × | ✓ | × | △ | × | × |
| GILL (Koh et al., 2023) | Image–Text External Diffusion Control | Generation | × | ✓ | × | ✓ | × | × |
| LVD (Lian et al., 2024) | Video–Text Planner-based Diffusion | Generation | △ | ✓ | × | ✓ | × | × |
| FlowZero (Lu et al., 2023) | Video–Text Planner-based Diffusion | Generation | △ | ✓ | × | △ | × | × |
| VideoDirectorGPT (Lin et al., 2024) | Video–Text Planner-based Diffusion | Generation | ✓ | ✓ | ✓ | ✓ | × | × |
| SynthID-Image (Gowal et al., 2025) | Image–Text External Diffusion Control | Governance | × | ✓ | × | × | × | ✓ |

video diversity based on motion entropy require additional modeling choices. These measures also require reliable feature extractors or judges and careful control of prompt distributions. In addition, diversity is closely linked with alignment. Increasing variety in a simple way can reduce the agreement between text and images. For this reason, many papers choose to prioritize faithful rendering over broad coverage. As a result, diversity is frequently treated as a qualitative claim or a secondary proxy in the audited sample. This leaves room for standardized and reproducible diversity panels that separate breadth from alignment.

**Safety.** Within Table 6, none of the six audited representative methods explicitly reports safety evaluation, even though most of them measure fidelity metrics such as realism and alignment. In contrast, safety is often treated as a secondary concern. It is usually addressed only through indirect measures such as brief qualitative discussions or data filtering rather than being measured in a systematic way. One major challenge is that multimodal safety involves a wide range of potential attacks. Examples include visual prompt injections and jailbreaks that are conditioned on images. Furthermore, a low success rate for attacks can be difficult to interpret. This result might mean that the model is safe or it might simply mean that the model failed to understand the malicious query. Finally, safety evaluation has two goals. These include stopping harmful responses and avoiding the refusal of safe requests. This balance makes it both difficult and costly to standardize safety reports.

**Provenance.** Provenance is explicitly evaluated in only one of the six audited works in Table 6. This is mainly because measuring provenance requires additional infrastructure such as cryptographic credentials or watermarking systems. It also requires testing the model against changes in the real world such as compression, cropping, re-encoding, or editing. However, as generative models become more capable and synthetic content spreads more widely, establishing verifiable provenance through robust and measurable watermarks is becoming essential for tracking content, ensuring accountability, and building trust.

## 6.5 Usage

Recent methodologies increasingly use synthetic vision and language data across four primary functional paradigms. This trend is largely independent of the underlying generation backbone. These paradigms include supervised fine-tuning, preference based alignment and reward modeling, data curation and remediation for weak supervision, and evaluation centered on safety or structure. In these contexts, synthetic image and text, video and text, and document vision and language samples serve not only as supplementary data but also as scalable supervision sources, explicit alignment signals, and controllable probes for assessing robustness and safety.

**Supervised Fine-tuning.** Significant research effort has been directed toward constructing large-scale and structured multimodal instruction datasets through synthetic captioning and automated prompt curation to drive supervised fine-tuning. Broadly speaking, these initiatives scale supervised fine-tuning along two principal axes. The first axis is modality coverage, which extends from static images to video. The second axis is semantic granularity, which progresses from generic captions to instance-level and OCR-aware instructions. Advancements along both of these axes have demonstrated strong correlations with improvements in downstream performance.

In the video domain, LLaVA-Video-178K introduces a largely synthetic instruction-following corpus that was assisted by GPT-4o and included human involvement in the pipeline. This dataset comprises detailed captions, open-ended question answering, and multiple-choice tasks, which yields consistent improvements in video large multimodal model training (Zhang et al., 2025e). To address text-rich imagery, TextSquare, also known as Square-10M, leverages closed-source multimodal LLMs to scale synthetic image and text instructions to the order of tens of millions. When this approach is utilized for large-scale supervised fine-tuning, it exhibits near-monotonic performance gains relative to the scale of the data (Tang et al., 2025).

Similarly, LLaVAR-2 integrates human-annotated captions with filtered instructions generated by GPT-4o to curate a high-quality and text-centric supervised fine-tuning dataset (Zhou et al., 2024). For the purpose of targeting instance-level grounding, Inst-IT synthesizes explicit visual-prompted instructions for continuous supervised fine-tuning. This method enhances instance-grounded understanding and demonstrates effective transfer to generic benchmarks (Peng et al., 2026). Concurrently, dense and high-fidelity synthetic captions have proven effective for supervision. For example, ShareGPT4V and ShareGPT4Video demonstrate that large-scale synthetic captioning directly benefits both supervised fine-tuning and unified large vision-language model pre-training (Chen et al., 2023b; 2024b).

**Preference alignment and reward modeling.** Beyond standard Supervised Fine-tuning, synthetic data plays a pivotal role in constructing pairwise preferences and training reward models. For video generation, VideoDPO proposes an automatic pipeline that generates multiple videos for each prompt and scores them using a multi-dimensional metric called OmniScore. This pipeline forms score-ranked preference pairs consisting of the best and worst samples. These pairs are further re-weighted during Direct Preference Optimization to prioritize distinctive and high-impact samples (Liu et al., 2024a).

Similarly, the work titled Improving Video Generation with Human Feedback leverages multi-dimensional human preferences regarding synthetic videos to train a VideoReward model. This reward model then guides generators through both Direct Preference Optimization objectives and reward-weighted inference (Liu et al., 2025a). In the domain of vision and language, VLFeedback provides more than 82,000 feedback instances annotated by artificial intelligence. These instances include preference labels and rationales across helpfulness, visual faithfulness, and ethical and safety aspects. Applying Direct Preference Optimization to VLFeedback to train the Silkie model improves helpfulness, visual faithfulness, and safety-related metrics. Experimental results show that applying this optimization method to such feedback significantly enhances the helpfulness, visual faithfulness, and safety of the models (Li et al., 2024b).

**Data curation, error correction, and failure-targeted synthesis.** Synthetic data is also instrumental in refining noisy real-world data, repairing weak labels, and generating hard negatives to expose model failures. For example, CapFusion leverages an LLM to consolidate and refine information from both noisy web-based image and text pairs and model-generated synthetic captions. This process produces higher-quality and more scalable supervision for multimodal pre-training (Yu et al., 2024b).

In the area of structured graphics, CHOCOLATE provides a human-annotated benchmark and a typology of factual errors in chart captions. This work also establishes the task of factual error correction for chart captions. It further introduces ChartVE, which is a reference-free visual entailment metric for chart and caption factuality. Additionally, the authors propose C2TFec, an interpretable two-stage framework that converts charts into tables before using an LLM to rectify factual inconsistencies (Huang et al., 2024a).

For document understanding, SynthDoc synthesizes bilingual document images containing text, tables, and charts. This study demonstrates that training OCR-free parsers similar to the Donut model on such data

improves pre-training read tasks and downstream robustness, even when language inconsistencies are present (Ding et al., 2024).

**Safety alignment and evaluation.** Beyond general capabilities, synthetic data is essential for red teaming techniques. These techniques involve generating harmful and helpful responses along with preference pairs to stress-test the safety of vision and language models. For instance, SPA-VL creates 100,000 image-instruction quadruples that feature chosen and rejected responses across six specific harm domains. This dataset is designed to support safety training based on Proximal Policy Optimization or Direct Preference Optimization (Zhang et al., 2025d). Beyond alignment based on preferences red teaming also requires stress testing failure modes that could lead to behavior that is not safe in an indirect manner. This includes visual scenes containing a large amount of text where optical character recognition and misunderstandings of the layout can result in interpretations that are harmful. Regarding evaluation, OCRBench v2 extends text-rich scenarios with human-verified questions to strictly examine the optical character recognition and structure understanding capabilities of large multimodal models (Fu et al., 2025).

Based on the discussion above, combining these distinct roles reveals that synthetic vision-language data have evolved into a fundamental infrastructure for post-training. This data not only scales supervised instruction learning but also provides explicit signals for alignment. Furthermore, it enables the repair of weak cross-modal supervision and supports rigorous safety and structural evaluations within an ecosystem driven by LLMs.

## 7 Agent Data

Within the framework of digital twins and embodied artificial intelligence, we analyze agent data generated by LLMs from the perspective of the final data product utilized in practical applications. This approach is motivated by the perspective that world models function as internal simulators that capture environment dynamics and enable forward as well as counterfactual rollouts to support perception, prediction, and decision making (Li et al., 2025c).

We classify the practical data products used in digital twins and embodied artificial intelligence into three primary categories. The first category is environment and task data, which describes world setups and task-oriented scenario configurations (Ruan et al., 2025). The second category consists of control and decision data, which captures policies, action traces, and other control decisions for steering agents and simulations. This includes LLM multi-agent parametrization trajectories and sequential control plans in digital twin simulations (Xia et al., 2024). The third category is perception and telemetry data, which focuses on observed sensor streams and logs.

A key feature of this taxonomy is that it classifies data based on its primary use. For instance, mixed artifacts such as trajectory logs are assigned to a category based on whether they primarily serve to design testbeds, train agent behaviors, or monitor system health. Based on this data-product perspective, egocentric streams that are typical of embodied agents and exocentric telemetry common in digital twins can be integrated into a single taxonomy according to their main application. This classification remains compatible with the broader world model perspective that connects perception, dynamics, and control across different domains (Li et al., 2025c).

### 7.1 Generation Methods

**Environment and task data.** Environment and task data include the specifications required to initialize an episode. These specifications define the appearance of the environment, the entities or agents that are present, and the goals and constraints that constitute a valid simulation run.

In urban traffic simulation settings, LLMs parse free-form requests into structured keywords such as a dictionary of scenario fields. These keywords then drive the generation of scriptable configurations and parameters for urban mobility testing (Li et al., 2024c). Multimodal pipelines extend this concept to rare and difficult situations by generating realistic corner cases along with runnable tests that target the long tail

of driving scenarios (Lu et al., 2024). Other complementary pipelines convert operational design domain descriptions into scripts compatible with ScenarioRunner for simulation-based testing (Danso & Büker, 2025).

Beyond traffic applications, semantic digital twins use LLMs to connect domain concepts to mission-level plans and recovery strategies. In these systems, the digital twin provides the domain concepts and interaction rules, while the LLMs generate structured action descriptions and recovery behaviors that can be executed and monitored within the twin (Naeem et al., 2025). At the enterprise level, work oriented toward Industry 5.0 proposes an Interactive-DT framework. In this framework, LLMs act as interactive interfaces and intelligent agents across the edge, digital twin, and service layers. This supports the construction and operation of digital twins, collaboration between cloud and edge systems, and advanced data analytics. This research also identifies integration challenges such as unreliability driven by hallucinations, which may have safety impacts, as well as bias, inference speed constraints, interoperability, and secure deployment according to standards (Chen et al., 2025a).

In the area of embodied artificial intelligence, long-horizon planners such as L3M+P generate PDDL problems and maintain world-state graphs that can be instantiated into many concrete tasks and goals (Agarwal et al., 2025b). Safety-focused planners such as SELP translate natural language tasks into temporal logic specifications and use equivalence voting to improve the robustness and consistency of the mapping from natural language to logic. Following this, constrained planning enforces the resulting safety constraints during execution (Wu et al., 2025). Self-correcting frameworks such as the T3 Planner verify plans against spatio-temporal logic and repair them when needed. This process produces verified plan and trajectory traces that can be recorded as supervision for evaluation or learning under explicit constraints (Li & Zhao, 2025). Large-scale multi-agent benchmarks such as PARTNR also belong to this category. In these benchmarks, LLMs help design and decompose collaboration tasks, ground them into simulators, and export them as standard task collections together with planner baselines (Chang et al., 2024).

**Control and decision data.** Control and decision data describe how agents act once an environment and task have been fixed. In digital twin settings, LLM agents can operate in a closed loop with the simulator. These agents read simulation data through a data interface and output parameterized application programming interface or function calls, which are often serialized as JSON, through a control interface to adjust simulation parameters. They then iteratively summarize outcomes for the next cycle.

Running such agents yields cycle-level traces that include simulation data, control parameters, and agent summaries. These traces are recorded as simulation logs and can be compiled into a concise sequential control or parametrization plan for offline analysis and what-if evaluation (Xia et al., 2024). In safety-critical infrastructures such as power grids, control policies coordinated by LLMs are executed inside digital twin sandboxes where numerical solvers validate stability and safety. The resulting control sequences and trajectories support stress tests and control studies without affecting the real system (Zhang et al., 2025c).

For human-centered digital twins, persona simulations based on LLMs can act as silicon samples for digital experimentation. Academics may use them for pilot experiments to identify impactful stimuli, while firms may explore strategies for customer insight and product development (Toubia et al., 2025). Persona-based behavior-chain benchmarks such as BehaviorChain quantify the extent to which current LLMs can faithfully simulate continuous human behavior (Li et al., 2025a).

In embodied artificial intelligence, LLMs can also augment demonstration and trajectory data more directly. For instance, LLM Trainer performs offline demonstration annotation and online keypose retargeting to adapt demonstrations to new scenes and generate additional imitation trajectories with minimal human input (George & Farimani, 2025). Frameworks such as ELLMER integrate high-level language-driven task decomposition with low-level execution using vision or force feedback and retrieval-augmented code examples to solve long-horizon tasks. The resulting executions naturally produce rich interaction traces that can be recorded for analysis (Mon-Williams et al., 2025). Program-structured approaches such as Instruct2Act and ProgPrompt leverage executable or program-like representations to ground instructions in available actions as well as perception and planning loops in some systems. This enables a more modular analysis of which components are necessary (Huang et al., 2023; Singh et al., 2022). When execution logs contain both actions

and rich observations, we treat them as control and decision data in this survey if their main use is to study or improve behavior rather than perception.

**Perception and telemetry data.** Perception and telemetry data focus on what is observed and recorded in digital twin and embodied environments, as well as on how these observations are generated and labeled. In digital twin pipelines for defect inspection, DefectTwin integrates a LLM driven multimodal pipeline to analyze multimodal inputs such as images. This system generates detailed defect descriptions and utilizes user interaction and feedback loops to support defect analysis and maintenance workflows (Ferdousi et al., 2024). In digital twins for autonomous driving based on camera and LiDAR sensors, a LLM interface enables online scenario editing through natural language prompts. These prompts allow for actions such as adding or removing assets, changing positions, or modifying appearances. This supports photorealistic and physically interactive simulations with sensor visualization and real-time interfacing with autonomous driving software stacks (Samak et al., 2025).

Similar workflows support egocentric data categories for embodied agents. LLM agents can produce Python scripts executable in Blender that retrieve and arrange 3D assets under spatial constraints derived from scene graphs. These agents render images and iteratively refine scenes through feedback from vision language models (Hu et al., 2024). LLMs can also be trained to generate Blender scripts for programmatic 3D object creation. The pipeline can render multiple views such as several images from different angles to increase visual diversity without manual modeling effort (Du et al., 2024). End-to-end pipelines based on mobile LiDAR enable rapid reconstruction of digital twin assets and can incorporate semantic enhancement guided by large vision language models into the reconstructed 3D representation. The resulting twins can be exported through formats such as OpenUSD for immersive inspection and downstream editing in platforms such as NVIDIA Omniverse (Gholizadeh HamlAbadi et al., 2025).

Across these examples, perception and telemetry data are primarily used for transfer learning, domain adaptation, and scalable evaluation using methods such as language-conditioned scoring or anomaly descriptions. In embodied setups, data consist of interaction traces in which observations are explicitly paired with agent actions, often following a partially observable Markov decision process formulation (Li et al., 2025c). In contrast, exocentric telemetry in digital twins is often structured for tracking past behaviors, monitoring current behaviors, and predicting future behaviors to support decision making as well as operations and maintenance (Deng et al., 2021; Bofill et al., 2023).

### 7.2 Quality Metrics for Agent Data

This section formalizes four metric families for agent data, which include validity, fidelity, diversity, and utility, and provides explicit definitions for each. We draw evaluation semantics from multi-agent embodied benchmarks such as PARTNR, text-to-traffic-scene evaluation on CARLA (Dosovitskiy et al., 2017), and world model or video assessments (Chang et al., 2024; Ruan et al., 2025; Li et al., 2025c). We consider trajectories defined as:

$$\tau = (s_0, a_0, r_1, s_1, \ldots, a_{T-1}, r_T, s_T)$$

and a dataset of generated trajectories denoted by:

$$\mathcal{A}_{\text{gen}} = \{\tau_i\}_{i=1}^N, \qquad \tau_i = (s_0^{(i)}, a_0^{(i)}, r_1^{(i)}, s_1^{(i)}, \ldots, a_{T_i-1}^{(i)}, r_{T_i}^{(i)}, s_{T_i}^{(i)})$$

In this formulation, $s_t$ represents the state or observation at step $t$, $a_t$ represents the action including tool or application programming interface calls, and $r_{t+1}$ denotes the reward or feedback associated with the transition from $(s_t, a_t)$ to $s_{t+1}$.

**Validity.** Validity measures whether generated agent data respect basic syntax, structural constraints, and executable preconditions. The action executability rate over all generated trajectories is defined as follows:

$$\text{ExecRate} = \frac{1}{\sum_{i=1}^N T_i} \sum_{i=1}^N \sum_{t=0}^{T_i-1} \mathbf{1}\Big\{\text{action } a_t^{(i)} \text{ executes without error in } s_t^{(i)}\Big\}.$$

Task success is a binary indicator of whether all goal predicates are satisfied at the end of an episode. Over $N$ episodes, the success rate is calculated by the following formula:

$$\text{SR}_{\text{valid}} = \frac{1}{N} \sum_{i=1}^{N} \mathbf{1}\{\text{all task goals satisfied in episode } i\}.$$

To capture partial completion, the percent complete metric reports the achieved fraction of goal predicates at termination. For a goal set $G$ and an achieved subset $\hat{G}(\tau_i)$, the metric is defined as:

$$\text{PC}(\tau_i) = \frac{|\hat{G}(\tau_i)|}{|G|}.$$

The PARTNR benchmark defines task evaluation functions using propositions together with dependencies and constraints such as temporal constraints. It computes percent completion as the ratio of satisfied propositions. Success is achieved when the percent completion value equals 1.0 (Chang et al., 2024). In text-to-traffic-scene generation for CARLA, generation correctness is evaluated through text matching. This method uses a binary matched or unmatched criterion between the input prompt and the rendered scene, and the results are averaged over repeated generations (Ruan et al., 2025).

**Fidelity.** Fidelity measures the degree to which synthetic observations match reference distributions and structures. This metric is widely used in evaluations related to world models and video prediction (Li et al., 2025c).

The Frechet Inception Distance compares Gaussian fits of real and generated feature embeddings (Heusel et al., 2018). We let $\mu_X$ and $\Sigma_X$ represent the mean and covariance of embeddings for real samples, $\text{Tr}(\cdot)$ denotes the matrix trace and we let $\mu_Y$ and $\Sigma_Y$ represent the mean and covariance for synthetic samples. The distance is calculated as follows:

$$\text{FID} = \|\mu_X - \mu_Y\|_2^2 + \text{Tr}\Big(\Sigma_X + \Sigma_Y - 2\big(\Sigma_X^{1/2}\Sigma_Y\Sigma_X^{1/2}\big)^{1/2}\Big).$$

The Frechet Video(FVD) applies the same mathematical form to video embeddings that capture temporal dynamics, where lower values indicate higher fidelity (Unterthiner et al., 2019).

Structural Similarity compares luminance, contrast, and structure between an image $x$ and a reference image $y$ (Wang et al., 2004). The metric is defined by the following equation:

$$\text{SSIM}(x,y) = \frac{(2\mu_x\mu_y + C_1)(2\sigma_{xy} + C_2)}{(\mu_x^2 + \mu_y^2 + C_1)(\sigma_x^2 + \sigma_y^2 + C_2)},$$

where $C_1$ and $C_2$ are small positive constants introduced to avoid numerical instability when the denominators are close to zero.

Mean Squared Error and Peak Signal-to-Noise Ratio are expressed as:

$$\text{MSE} = \frac{1}{M} \sum_{i=1}^{M} (x_i - y_i)^2,$$

$$\text{PSNR} = 10 \log_{10}\left(\frac{\text{MAX}^2}{\text{MSE}}\right).$$

In these formulas, $M$ represents the number of pixels or dimensions, and MAX denotes the maximum possible value of the signal.

Learned Perceptual Image Patch Similarity computes a perceptual distance between deep feature activations (Zhang et al., 2018). For layer-wise normalized features $\hat{f}_{h,w,x}^l$ and $\hat{f}_{h,w,y}^l$ as well as learned weights $w_l$, the distance is calculated as:

$$\text{LPIPS}(x,y) = \sum_l \frac{1}{H_l W_l} \sum_{h,w} \left\| w_l \odot \big(\hat{f}_{h,w,x}^l - \hat{f}_{h,w,y}^l\big) \right\|_2^2.$$

where the symbol $\odot$ denotes the Hadamard element-wise product. This operation represents channel-wise scaling performed by the learned weight vector $w_l$.

The evaluation metrics mentioned above assume either distributional access to real samples, as in the cases of Frechet Inception Distance and Frechet Video Distance, or the availability of paired references such as Structural Similarity, Mean Squared Error, Peak Signal-to-Noise Ratio, and Learned Perceptual Image Patch Similarity. However, in many conditional generation settings, paired pixel-level ground truth is not available. In these situations, semantic prompt and outcome alignment is utilized as a reference-free proxy.

When pixel-level ground truth is unavailable, prompt–outcome semantic alignment can serve as a reference-free proxy for fidelity that is broadly applicable. This metric captures whether a synthesized outcome is consistent with the conditioning prompt at the level of high-level semantics. Given a textual prompt or specification $p$ and a generated outcome $o$, such as a rendered image or keyframe produced from a generated scenario, cross-modal embeddings are typically obtained using a pre-trained vision–language model such as CLIP (Radford et al., 2021).

Let $f_T(\cdot)$ and $f_V(\cdot)$ denote the text and vision encoders, and let

$$e_p = f_T(p), \qquad e_o = f_V(o),$$

denote the resulting embeddings.

The prompt–outcome semantic alignment score can be expressed using cosine similarity:

$$\text{SemAlign}(p, o) = \frac{e_p^\top e_o}{\|e_p\|_2 \, \|e_o\|_2},$$

where $\| \cdot \|_2$ is the $\ell_2$ norm and higher values indicate stronger semantic agreement.

Over $N$ prompt and outcome pairs $\{(p_i, o_i)\}_{i=1}^N$, the mean semantic fidelity is calculated as:

$$\text{SemFid} = \frac{1}{N} \sum_{i=1}^N \text{SemAlign}(p_i, o_i).$$

The mean semantic fidelity summarizes the semantic consistency between conditions and synthesized outcomes without requiring a reference image for each prompt. This approach is closely related to CLIP-based and reference-free evaluation metrics used in vision and language generation (Hessel et al., 2022).

When a discrete correctness signal is preferred, such as a matched or unmatched criterion, one may threshold the semantic alignment score and report the resulting match rate:

$$\text{MatchRate}(\tau) = \frac{1}{N} \sum_{i=1}^N \mathbf{1}\{\text{SemAlign}(p_i, o_i) \geq \tau\},$$

where $\tau$ is a similarity threshold and $\mathbf{1}\{\cdot\}$ is the indicator function.

**Diversity.** Diversity captures the coverage of modes, long-tail concepts, and non-templated variety within the generated data. In the context of text-to-traffic-scene generation, diversity can be quantified using agent diversity (AD) and road diversity (RD). Both of these metrics are computed as the ratio of unique elements to total elements across repeated generations. Specifically, agent diversity accounts for variations in agent type, action, and relative position, while road diversity is calculated based on unique road identifiers (Ruan et al., 2025).

For observation-level evaluation of generated videos or world model rollouts, the VBench framework decomposes video generation assessment into complementary axes such as subject consistency and temporal quality. An example of temporal quality is motion smoothness. This framework also separately evaluates the consistency between the video and the condition, which refers to how well the output adheres to the given conditions or prompts. These axis-wise scores provide fine-grained evidence regarding strengths and weaknesses across the prompt and condition space (Huang et al., 2024b; Li et al., 2025c).

**Utility.** Utility measures downstream effectiveness during training or evaluation with synthetic agent data. The success rate over $N$ evaluation episodes with binary success indicators $s_i$ taking values of zero or one is given by the following equation:

$$\text{SR}_{\text{eval}} = \frac{1}{N} \sum_{i=1}^{N} s_i.$$

To quantify efficiency, let $L_i$ denote the number of environment steps used in episode $i$:

$$\text{SimSteps} = \frac{1}{N} \sum_{i=1}^{N} L_i.$$

In collaborative settings, task offloading measures the division of labor. Let $n_i^{(r)}$ be the number of sub-tasks or propositions completed by the robot in episode $i$ and $n_i$ be the total number of completed sub-tasks. The metric is expressed as:

$$\text{Offloading} = \frac{1}{N} \sum_{i=1}^{N} \frac{n_i^{(r)}}{n_i}.$$

For partial task completion at the episode level, the percent complete metric can be reused:

$$\text{PC}(\tau_i) = \frac{|\hat{G}(\tau_i)|}{|G|}.$$

In SafeBench-style CARLA driving evaluation, the collision rate (CR) is defined as the expected collision indicator (or count) over evaluation scenarios $\tau$:

$$\text{CR} = \mathbb{E}_{\tau \sim P}[c(\tau)],$$

where $c(\tau)$ denotes the collision signal in scenario $\tau$. In many episodic driving setups where an episode terminates upon the first collision (or collisions are binarized), $c(\tau) \in \{0, 1\}$ and CR can be interpreted as the proportion of episodes that contain at least one collision. Lower values indicate better safety performance (Xu et al., 2022; Zhang et al., 2024b). The overall score is a composite metric that aggregates driving metrics related to safety, functionality, and etiquette by using weights specified by the benchmark. In practice, the overall score is commonly interpreted as a measure of ego-vehicle driving performance where higher values are preferred. Conversely, adversarial scenario generation or selection may instead aim to reduce the overall score by constructing more challenging conditions.

Reinforcement learning studies also report the discounted return per episode:

$$G_t = \sum_{k=0}^{\infty} \gamma^k \, r_{t+k+1},$$

This return, as well as the average return across episodes, is used to assess whether synthetic data improve policy performance or sample efficiency.

### 7.3 Trustworthy Metrics for Agent Data

In practice, the evaluation of agent data generated by LLMs has primarily focused on safety. This focus examines whether trajectories, scenarios, and plans avoid dangerous states and behaviors that break established rules. Other dimensions of trustworthiness, such as fairness and privacy, are significantly less developed in current benchmarks for embodied agents and digital twins. Therefore, we treat these aspects as open directions for future research and focus our current discussion on safety metrics.

**Safety.** Safety measures whether trajectories, scenarios, and plans generated by LLMs avoid hazardous states and rule-breaking behaviors and satisfy explicit safety constraints, independent of task completion. In practice, safety is often assessed through infraction and collision metrics defined by simulators in driving benchmarks, such as suites based on CARLA like SafeBench. It is also evaluated through the satisfaction of formal specifications in pipelines for constraint-enforced planning and motion planning (Xu et al., 2022; Wu et al., 2025; Li & Zhao, 2025). Systems for text-to-scenario generation further emphasize the production of standardized scenario files and simulator testing reports that include monitored evaluation indicators (Cai et al., 2025).

Rule violations include basic traffic infractions such as running red lights, driving the wrong way, lane-keeping violations, and speeding. A generic per-episode violation rate can be calculated as:

$$\text{RVR} = \frac{1}{N} \sum_{i=1}^{N} \frac{V_i}{E_i},$$

where $V_i$ represents the total number of rule violations in episode $i$, and $E_i$ represents an exposure term such as the number of decision steps $T_i$ or the distance traveled.

In practice, CARLA-based benchmarks frequently report specific categories of violation metrics such as collisions, red-light running, stop-sign running, out-of-road distance, and lane invasion rather than a single aggregated count (Xu et al., 2022). An exposure-normalized violation rate for a specific infraction type $t$ can be defined as follows:

$$\text{RVR}^{(t)} = \frac{\sum_{i=1}^{N} V_i^{(t)}}{\sum_{i=1}^{N} \text{dist}_i},$$

where $V_i^{(t)}$ is the number of infractions of type $t$ in episode $i$. This style of reporting is also used in CARLA Leaderboard results where multiple infraction types are shown as infractions per kilometer (CARLA Autonomous Driving Leaderboard, 2025b). Fine-grained violation counts for each specific type additionally enable a more detailed diagnosis than aggregate success or return (Cai et al., 2025; Xu et al., 2022).

Route incompleteness measures the extent to which the planned route remains unfinished at the end of a scenario:

$$\text{RI} = 1 - \frac{\text{distance completed}}{\text{planned route length}}.$$

Higher values for this metric indicate early termination or a failure to follow the designated route. In CARLA-style evaluations, route progress or completion is commonly reported together with violation and collision metrics. This approach helps to distinguish safe task completion from instances of early stopping or unsafe driving (Xu et al., 2022).

Speed-related compliance and infractions capture whether an agent drives too slowly or too fast relative to legal or context-dependent bounds. A simple minimum-speed compliance rate is defined as follows:

$$\text{MSCR} = \frac{1}{T} \sum_{t=1}^{T} \mathbb{I}(v(t) \geq v_{\min}(t)),$$

where $v_{\min}(t)$ is a context-dependent minimum-speed requirement such as a speed determined by nearby traffic. The CARLA Leaderboard explicitly penalizes the failure to maintain minimum speed as part of its evaluation criteria (CARLA Autonomous Driving Leaderboard, 2025a). More generally, one may also track speed-limit compliance within specific bounds when such information is available in the simulator or map.

Comfort and smoothness often serve as indicators for aggressive or risky maneuvers. CARLA-based diagnostic reports may include kinematics-based indicators such as average acceleration and yaw velocity (Xu et al., 2022). For example, if $a(t)$ represents the acceleration and $\omega(t)$ represents the yaw velocity, these can be quantified as follows:

$$\text{ACC} = \mathbb{E}_{\tau \sim P}[\text{acc}(\tau)],$$
$$\text{YV} = \mathbb{E}_{\tau \sim P}[y(\tau)].$$

Furthermore, more sensitive smoothness indicators can be computed, such as root-mean-square jerk and the hard-braking rate:

$$\text{Jerk}_{\text{RMS}} = \sqrt{\frac{1}{T-2} \sum_{t=2}^{T-1} \left\| \frac{a(t+1) - a(t-1)}{2\Delta t} \right\|^2},$$

$$\text{HardBrakeRate} = \frac{1}{T} \sum_{t=1}^{T} \mathbb{I}(a_x(t) < -\tau),$$

where $a_x(t)$ is longitudinal acceleration and $\tau$ is a braking threshold. These smoothness metrics help to characterize the trade-off between safety and efficiency while complementing collision and completion metrics.

Formal safety satisfaction measures the fraction of executions that satisfy a temporal-logic specification $\phi$, such as those defined by Linear Temporal Logic (LTL) or Signal Temporal Logic (STL). If $\sigma^{(i)}$ represents the execution trace for episode $i$, then the safety satisfaction rate is defined as follows:

$$\text{SafetySat} = \frac{1}{N} \sum_{i=1}^{N} \mathbb{I}(\sigma^{(i)} \models \phi).$$

The SELP framework improves the safety rate by mapping natural language to Linear Temporal Logic and enforcing constrained decoding, while the $\text{T}^3$ Planner utilizes Signal Temporal Logic verification in the loop to increase the satisfaction rates of motion plans (Wu et al., 2025; Li & Zhao, 2025).

Hazard rejection and risk are safety-specific metrics for embodied LLM agents that capture how these agents respond to explicitly hazardous tasks. Given a set of labeled hazardous tasks $\mathcal{H}$ and a set of safe tasks $\mathcal{S}$, the hazard rejection rate and risk rate are defined as follows:

$$\text{Rejection} = \frac{1}{|\mathcal{H}|} \sum_{h \in \mathcal{H}} \mathbb{I}(\text{agent refuses } h),$$

$$\text{Risk} = \frac{1}{|\mathcal{H}|} \sum_{h \in \mathcal{H}} \mathbb{I}(\text{agent executes } h).$$

SafeAgentBench reports a low rejection rate and non-trivial risk for current embodied LLM agents, which motivates the explicit tracking of these quantities (Yin et al., 2025).

Time-to-Collision is a widely used proximity-based surrogate safety indicator in traffic safety assessment. Given the relative longitudinal distance $d(t)$ and closing speed $\dot{d}(t) < 0$, the instantaneous Time-to-Collision and its minimum over an episode are defined as follows:

$$\text{TTC}(t) = \begin{cases} \dfrac{d(t)}{-\dot{d}(t)}, & \dot{d}(t) < 0, \\ +\infty, & \dot{d}(t) \geq 0, \end{cases}$$

$$\text{minTTC} = \min_t \text{TTC}(t).$$

Lower values for the minimum Time-to-Collision indicate a higher risk of collision (Ward et al., 2015; Sharath & Mehran, 2021).

Minimum Distance to Collision is another commonly used proximity-based indicator. It is defined by using the positions of the ego agent $\mathbf{p}_{\text{ego}}(t)$ and the other agent $\mathbf{p}_{\text{other}}(t)$ as follows:

$$\text{MDC} = \min_t \left\| \mathbf{p}_{\text{ego}}(t) - \mathbf{p}_{\text{other}}(t) \right\|.$$

Lower values for the Minimum Distance to Collision indicate higher collision risk (Gao et al., 2025).

These proximity measures based on Time-to-Collision and Minimum Distance to Collision have also been adopted when evaluating traffic scenarios generated or selected with the aid of LLMs (Gao et al., 2025).

Violation diagnosis accuracy evaluates the reliability of automated detectors, including auditors based on multimodal LLMs, when used to identify safety violations or accidents from logs, images, or narratives. This accuracy is typically summarized by standard classification metrics such as precision, recall, and the F1 score (Skender et al., 2025). Additionally, the SeeUnsafe framework proposes a multimodal LLM agent for traffic accident analysis and introduces an Information Matching Score to align structured model responses with ground truth data (Zhang et al., 2025b).

### 7.4 Evaluation Practice Gap

We analyze representative methods from Section 7.1 and categorize their reported evaluation protocols in Table 7.

Table 7: Whether representative agent-data generation methods explicitly evaluate each dimension in their experimental sections, together with their method family and functional role. ✓: explicitly evaluated; △: partially or indirectly covered; ×: not reported or not applicable.

| Representative Work | Family | Role | Validity | Fidelity | Diversity | Utility | Safety |
|---|---|---|---|---|---|---|---|
| TTSG (Ruan et al., 2025) | Environment & Task Data | Generation | ✓ | △ | ✓ | ✓ | ✓ |
| PARTNR (Chang et al., 2024) | Environment & Task Data | Generation | ✓ | × | × | ✓ | × |
| SELP (Wu et al., 2025) | Environment & Task Data | Governance | ✓ | × | × | ✓ | ✓ |
| Grid-Agent (Zhang et al., 2025c) | Control & Decision Data | Generation | ✓ | × | × | ✓ | △ |
| $T^3$ Planner (Li & Zhao, 2025) | Environment & Task Data | Governance | ✓ | △ | × | ✓ | ✓ |
| SceneCraft (Hu et al., 2024) | Perception & Telemetry Data | Generation | ✓ | △ | × | ✓ | × |

**Diversity.** Within the representative audit in Table 7, only one of the six audited agent-data methods explicitly reports diversity metrics. Our analysis shows that only generation from text to traffic scenes explicitly reports diversity metrics. In contrast, other representative methods do not provide quantitative measures of coverage or variation that is not based on templates under repeated sampling. Consequently, although these methods allow for comparisons of success rates or efficiency, it remains difficult to determine from this audited sample whether the generated data meaningfully expands the coverage of behavior modes, tasks, or scenarios.

**Fidelity.** Within Table 7, none of the six audited methods explicitly evaluates distribution-level fidelity, and three provide only partial proxy coverage. Evaluations are therefore mostly focused on consistency or correctness proxies rather than direct tests of distributional alignment. In our representative set, the generation from text to traffic scenes reports prompt scene matching as a consistency proxy. In contrast, methods centered on planning and control focus mainly on success rates, completion, and constraint satisfaction. These methods include SELP (Wu et al., 2025), PARTNR (Chang et al., 2024), and Grid Agent (Zhang et al., 2025c). These studies generally do not report metrics that compare synthetic trajectories or plans to reference data distributions. As a result, while consistency proxies are partially covered, the statistical proximity of synthetic agent data to reference distributions is still mostly unvalidated.

**Safety.** Concerning the diagnosis of safety issues, current reporting relies heavily on broad benchmark metrics such as those related to collisions in evaluations using the CARLA simulator. It also relies on the implicit satisfaction of specifications such as adherence to linear temporal logic or signal temporal logic. These methods are used instead of detailed and specific diagnostics for different types of violations. In our representative set, the generation of traffic scenes reports signals related to collisions. Meanwhile, the SELP (Wu et al., 2025) and $T^3$ Planner (Li & Zhao, 2025) methods mainly focus on the satisfaction of formal constraints. In contrast, other approaches mainly report success at the task level. This includes the completion of tasks or the resolution of violations. However, they do not provide an independent breakdown of safety violations for each specific type. As a result, although current evaluations can show whether safety

constraints are met, they provide limited diagnostic detail for identifying and explaining specific modes of safety failure.

## 7.5 Usage

Having discussed the generation of agent data, we now focus on its practical applications. This section examines how trajectories, task specifications, and telemetry streams are used to train, evaluate, and refine agents based on LLMs in both embodied and digital twin environments. We follow the same three-part structure established previously by categorizing these applications into environment and task data for testbeds, control and decision data for supervision, and perception and telemetry data for evaluation signals.

**Environment and task data usages.** Environment and task data primarily serve as a rigorous testbed for reasoning, planning, and safety verification. In the context of lifelong planning, maintaining a persistent memory of the world through world-state graphs is crucial. This approach enables agents to repeatedly instantiate and solve PDDL planning problems from natural language tasks as the environment evolves, which supports the repeatable evaluation of long-horizon symbolic planning (Agarwal et al., 2025b).

When safety is paramount, these data specifications act as strict constraints. Safety-focused pipelines convert natural language instructions into formal temporal logic, such as Linear Temporal Logic, and then utilize constrained decoding guided by automata to ensure that generated plans adhere to these formal constraints step by step (Wu et al., 2025).

Beyond static enforcement, this data also drives self-correction. Systems often pair planners based on LLMs with logic-based verifiers that iteratively reject and repair unsafe actions (Li & Zhao, 2025). This iterative loop can be logged to form trial-and-error traces consisting of failed plans, verifier diagnostics such as constraint violations or robustness scores, and corrected revisions. These traces are useful for evaluation and may be repurposed for future training.

For multi-agent systems, collaboration benchmarks provide grounded and simulator-verifiable tasks. These datasets effectively transform individual environments into collaborative evaluation suites designed to test how well agents coordinate, track task progress, and recover from errors through language (Chang et al., 2024). Similarly, in digital twin settings, formal task descriptions define the distinct operational boundaries for system controllers. These specifications are used to stress-test governance policies and verify that fallback strategies function correctly under diverse conditions.

**Control and decision data usages.** Control and decision data provide direct supervision for language-driven policies and are reused in both imitation and feedback-based learning. On the demonstration side, recent frameworks ground decisions from LLMs in collections of trajectories. For example, combining language understanding with value estimates learned from robot interaction data allows a model to choose actions that are consistent with successful past behavior (Ahn et al., 2022). Programmatic control follows a related pattern where natural language is translated into executable policies that can be directly run on robotic platforms (Liang et al., 2023). More generally, executing such policies naturally produces traces including states, actions, and failures that could be repurposed for analysis, dataset construction, or distillation into smaller controllers. Large and heterogeneous robot logs have been pooled to train instruction-following visuomotor policies that generalize across different robots, which enables analyses of how dataset diversity and composition affect language-conditioned generalization (Collaboration et al., 2025).

The same data also drive reinforcement-style updates that are mediated by language. Automated reward design utilizes LLMs to write and refine reward functions, which are frequently formatted as executable code. These functions are validated through downstream policy optimization and evaluation in the target environment, which is typically performed in simulation to enable rapid iteration over reward hypotheses (Ma et al., 2024a; Xie et al., 2024).

In offline settings, policies can be conditioned on language embeddings derived from LLMs and trained via offline reinforcement learning on static logs. This approach enables generalization to unseen commands without the need for additional interaction (Morad et al., 2024). To reduce the effort required for human labeling, artificial intelligence feedback protocols use multimodal critics to score trajectories after they occur.

This process converts archived trials into training data for behavior shaping. For example, recent work adapts video and language models into language-conditioned robotic reward functions that score executions directly from video (Yang et al., 2024c). Complementary to critic-based feedback, methods for one-video reward inference can compute dense rewards from a single demonstration video using methods such as semantic point correspondence and automated point tracking. This allows for trajectory evaluation and dataset filtering for downstream policy synthesis (Shi et al., 2025a).

In digital twins, control logs from simulation play a similar role. These logs can be relabeled, scored, and reused to refine control policies and to analyze how controllers driven by LLMs behave under distribution shifts or rare events.

**Perception and telemetry data usage.** Perception and telemetry data focus less on direct control and more on evaluation and reward learning. Streams aligned with actions, such as egocentric or first-person observations from cameras and proprioception, provide raw material for judging behavior and system state. Similarly, exocentric system events, including third-person logs from external cameras, map-based states, and simulator records, serve a similar purpose. Methods that utilize LLMs as judges evaluate diverse artifacts and multimodal inputs along multiple criteria. The reliability of these assessments is supported by careful design of prompts and inputs as well as strategies for consistency and bias mitigation (Gu et al., 2025; Li et al., 2024a).

Vision and language reward learning can adapt video and language models into language-conditioned reward functions using successful and failed execution videos. This process produces reward scores and signals that can guide planning or reinforcement learning (Yang et al., 2024c). Furthermore, LLMs can propose parameterized reward features and iteratively refine reward parameters using execution feedback. This refinement is achieved by minimizing ranking inconsistency between the model and the learned reward function (Zeng et al., 2024). In the domain of driving evaluation, frameworks based on LLMs can transform multi-source driving logs, such as surround-view videos and CAN bus signals, into structured driving contexts. These systems then output structured assessments of driving behaviors across safety, intelligence, and comfort, which are validated in simulation (You et al., 2025).

Large and multi-domain observation corpora also support 4D world modeling, including geometric understanding and camera-conditioned video generation. These corpora provide temporally aligned video streams with rich multimodal signals such as depth maps, camera poses, optical flow, and foreground masks. Such data enables models to learn scene dynamics and generate videos that adhere to specified camera trajectories. OmniWorld presents a unified resource for 4D world modeling that spans reconstruction and future-oriented prediction needs. It introduces a benchmark centered on 3D geometric prediction and camera-controlled video generation, which is built from a newly collected game subset called OmniWorld-Game and curated public datasets across diverse domains (Zhou et al., 2025).

# 8 Cross-Modal Synthesis of Evaluation Dimensions and Gaps

## 8.1 Cross-Modal Synthesis of the Unified Evaluation Space

Table 1 outlines the unified evaluation framework of this survey, detailing the quality and trustworthiness dimensions used to assess LLM-generated synthetic data across six data types. Table 8 serves a complementary purpose. Rather than providing another modality-by-dimension inventory or a simple count summary, it adds a cross-modal interpretive layer to the same evaluation space. By regrouping these dimensions into foundational, transferred, and modality-concentrated roles, Table 8 makes comparative patterns more explicit than the modality-by-modality presentation alone.

Specifically, foundational dimensions provide the most consistent dimension across the six data types, although their concrete operationalization remains modality-specific. Transferred dimensions remain broadly relevant across modalities, but their operational definitions shift with the artifact being evaluated. Finally, modality-concentrated dimensions highlight criteria that become more explicit or structurally central in particular data types due to their representational formats, deployment contexts, or governance risks. In this way, the cross-modal synthesis extends the unified evaluation space beyond parallel organization alone by foundational

Table 8: Cross-modal synthesis across the six data types. The table organizes recurring evaluation patterns into three cross-modal roles—foundational, transferred but modality-dependent, and modality-concentrated—and summarizes how each role is characteristically instantiated for each data type.

| Data Type | Foundational
*Validity; Fidelity* | Transferred, but
Modality-Dependent
*Diversity; Utility; Faithfulness;*
*Safety; Robustness* | Modality-Concentrated
*Privacy; Fairness; Provenance;*
*Benchmark Integrity and*
*Contamination* |
|---|---|---|---|
| **Text** | linguistic well-formedness and semantic fidelity | semantic diversity, instruction utility, and safety filtering | benchmark contamination and provenance controls |
| **Reasoning** | answer correctness and verifiable intermediate support | trajectory diversity, process faithfulness, and OOD robustness | benchmark leakage and contamination auditing |
| **Tabular** | schema-valid records and distributional fidelity | distribution coverage, TSTR utility, and robustness trade-offs | privacy leakage and subgroup fairness |
| **Semi-Structured** | schema validity and structural fidelity | structural novelty, downstream utility, and perturbation robustness | privacy-sensitive structures and governance controls |
| **Vision–Language** | content validity and cross-modal fidelity | aligned diversity, QA utility, and multimodal safety checks | provenance validation and credential tracing |
| **Agent** | execution validity and trajectory fidelity | scenario diversity, task utility, and safety-constrained control | not a primary governance axis in current audits |

dimensions, transferred concerns, and modality-concentrated dimensions explicit across the six data types covered in this work.

## 8.2 Shared Foundations and Transferred Instantiations

The most stable shared dimensions in Table 8 are Validity and Fidelity. Across all six data types, these two dimensions evaluate whether synthetic data are well-formed for the modality it claims to represent and whether these preserve the content, structure, or correspondence required for meaningful use. Their concrete instantiation remains modality-specific. In text, they concern linguistic well-formedness and semantic fidelity; in reasoning, they extend to answer correctness together with verifiable intermediate support. In tabular and semi-structured generation, they are reflected in schema validity, structural fidelity, and distributional alignment. In vision–language and agent data, they appear through cross-modal content validity and trajectory-level execution fidelity, respectively. Taken together, Validity and Fidelity form the most consistent cross-modal dimensions because they capture whether the generated artifact is appropriate for its modality and remains aligned with the target content or structure it is meant to preserve.

Beyond this shared core, our synthesis treats Diversity, Utility, Faithfulness, Safety, and Robustness as transferable evaluation concerns, albeit through highly contextualized instantiations. While diversity and utility remain broadly relevant across the six data types covered here, their practical targets shift—capturing semantic and lexical breadth in text, valid trajectory variations in reasoning, downstream effectiveness (e.g., TSTR) in tabular and semi-structured formats, and scenario coverage or task success in agent environments. Faithfulness, safety, and robustness exhibit similar modality-specific adaptations. Text evaluation prioritizes attribution and safety filters, whereas reasoning focuses on process faithfulness and out-of-distribution stability. In multimodal and agentic settings, these dimensions further extend to alignment-sensitive failure modes, physical safety constraints, and environment-specific control requirements. Consequently, what is unified in this survey is the evaluation lens, rather than a single metric formula shared identically across modalities. Table 8 therefore reframes these dimensions not as isolated add-ons or perfectly uniform constructs, but as transferred evaluative inquiries that must be re-instantiated for each distinct data form.

## 8.3 Modality-Concentrated Dimensions

In contrast to the more consistently shared foundations discussed in Section 8.2, the third column of Table 8 highlights dimensions that become more explicit in only a subset of the six data types. Privacy, Fairness, Provenance, and Benchmark Contamination are not uniformly foregrounded across modalities; instead, they become most visible where the underlying data representation, evaluation protocol, or governance objective provides a more direct audit surface. In this sense, their concentration reflects the current limits of operational evaluability in the representative literature surveyed here, rather than rigid boundaries of conceptual relevance.

This pattern is especially visible for privacy and fairness in structured domains. In tabular generation, records, attributes, and subgroup memberships make both dimensions more directly auditable. For example, in Train-on-Synthetic-Test-on-Real (TSTR) settings, privacy leakage and subgroup disparities can be reflected in measurable downstream outcomes. Semi-structured formats such as graphs, JSON records, and logs inherit part of this logic when they preserve identifiable structural metadata, although privacy evaluation there is typically activated under narrower governance-oriented threat models.

Provenance is most explicitly foregrounded in vision–language generation, where authenticity and source tracing are more readily connected to dedicated technical mechanisms such as credentialing and watermarking. By contrast, benchmark integrity and contamination are more prominently surfaced in text and reasoning, where close coupling to public evaluation sets and reference solutions makes overlap-based or split-based audits more natural to define.

# 9 Open Challenges and Future Directions

While this survey establishes a unified taxonomy for evaluating data generation driven by LLMs across modalities, the field is rapidly transitioning from static dataset creation to dynamic and self-improving

synthesis ecosystems. Current metrics were largely designed for the era of fixed human-annotated corpora and they face significant challenges in this new regime. Building on our established framework of quality and trustworthiness, we highlight directions where methodological advancement is required to ensure the sustainability and reliability of synthetic data.

## 9.1 From Static Snapshots to Dynamic Feedback-Loop Evaluation

Most metrics reviewed in this survey, such as diversity measured through self-similarity or fidelity measured through distributional distance, provide only a static snapshot of a single generation round. However, practical applications are increasingly moving toward recursive training loops where synthetic data are used to train models that subsequently generate new data.

The main challenge is that these static snapshots fail to capture long-term system dynamics. A dataset may achieve high scores for diversity in the first iteration yet still drive model collapse. This is a degenerative process where distributional tails disappear and modes are over-amplified after several training and generation cycles.

A key future direction is to move from point-wise evaluation to longitudinal trajectory monitoring. There is a need for dynamic metrics that track the derivatives of quality over time. Such metrics would detect the loss of support coverage, the contraction of the feature space, or the drift of error patterns across iterations. These temporal metrics could serve as early warning signals that trigger interventions, such as mixing in real data, rebalancing domains, or adjusting sampling strategies, before model collapse becomes irreversible. More broadly, this shift requires meta-evaluation protocols to test whether existing metrics are sufficiently sensitive to serve as stability controllers in feedback loops.

## 9.2 Redefining Fidelity: From Surface Mimicry to Process Verifiability

In our taxonomy, fidelity has traditionally measured how closely generated data match the distribution of real world human data. For open-ended natural language tasks such as dialogue, this human-centric distributional similarity remains central. However, for reasoning-intensive domains, this definition is becoming a bottleneck. As reasoning-oriented LLMs begin to match or surpass average human performance, enforcing strict adherence to human distributions may penalize correct but novel solutions. In fields such as mathematics, programming, or symbolic logic, sounding human is less important than being objectively correct.

For such modalities, fidelity metrics must evolve from measuring mimicry, which is the distributional similarity to human artifacts, toward measuring verifiability. Promising directions include scalable and automated process-level rewards. These rewards may include execution feedback for code, formal proof checkers for mathematics, or logical consistency probes for chain-of-thought traces. The goal is to assess the correctness and internal coherence of the reasoning process rather than its stylistic resemblance to human baselines.

## 9.3 Navigating the Trust and Utility Pareto Frontier

Our framework distinguishes between Quality, which includes downstream utility, and Trustworthiness, which covers privacy, safety, and fairness. While these are often treated as separate pillars in a taxonomy, they frequently act as competing objectives during deployment.

Mechanisms designed to increase trustworthiness often impose an alignment tax on synthetic corpora. For instance, aggressive safety filtering may not only remove harmful content but also disproportionately eliminate rare concepts, which effectively shrinks tail coverage and reduces diversity. Similarly, adding differential privacy noise to tabular data can improve privacy guarantees but degrade predictive performance. Strict alignment in conversational agents may also reduce harmful outputs at the cost of elevated refusal rates on benign queries, which is a phenomenon known as over-refusal.

Future work should move beyond optimizing individual metrics in isolation and instead quantify and navigate these trade-offs. Concretely, we need frameworks that characterize the Pareto optimal frontier over pipeline configurations. This is the set of operating points where no trust metric can be improved without degrading utility or vice versa. This would enable explicit and application dependent choices, such as estimating how

much downstream performance must be sacrificed to attain a target privacy guarantee defined by $\epsilon$ and $\delta$, or how much refusal rate slack is acceptable to achieve a desired safety level. More broadly, this calls for multi-objective optimization and selection strategies that treat trust and utility as co-dependent variables rather than independent checkboxes.

## 10    Conclusion

Current research efforts are predominantly directed toward leveraging Large Language Models (LLMs) for data generation, while the critical role of the **"Data Auditor"**—responsible for evaluating the quality of synthetic data—has been relatively overlooked. Ensuring high data quality is essential for transforming scarce data into a controllable resource, suitable not only for model training but also for direct real-world applications.

In this survey, centered on synthetic data, we aim to bridge fragmented progress and provide a systematic understanding of evaluation methods through our proposed LLM Data Auditor framework. Beyond summarizing typical generation methods across various modalities, we categorize representative metrics into two primary dimensions: Quality and Trustworthiness. By applying this evaluation system to representative works across modalities, we highlight under-covered evaluation dimensions within our audited sample. For instance, within our representative audit of tabular data generation, fairness-oriented evaluation appears in only 2 of the 7 audited works, suggesting that fairness is less consistently reported than several other evaluation dimensions in this modality. Consequently, our framework serves as both a comprehensive reference and a diagnostic tool to pinpoint missing evaluation dimensions across different data modalities.

Our findings suggest that, regardless of the modality, there is an urgent need to refine post-generation evaluation metrics. A holistic assessment is necessary to prevent synthetic data from excelling in one dimension while failing in others. Furthermore, we highlight inherent trade-offs between certain metrics, such as the tension between privacy and fidelity in tabular data, which further underscores the necessity of a multi-dimensional evaluation system. This need becomes even more pressing as synthetic data is increasingly used in settings where its impact is not limited to benchmark performance. In such cases, shortcomings in fairness, privacy, or safety should not be regarded as purely technical deficiencies. For example, if synthetic data distorts subgroup proportions or label-conditional distributions, it may introduce or reinforce disparities in downstream deployment. This is particularly important in socially sensitive applications, where insufficient auditing may obscure harms behind strong aggregate utility or fidelity scores. At the same time, this survey is intended primarily as a conceptual and analytical synthesis of fragmented evaluation practices, rather than an end-to-end empirical validation of the proposed framework on a specific synthetic dataset.

For future work, an important next step is to operationalize the evaluation framework proposed in this survey in end-to-end empirical auditing settings, which may also serve as a foundation for developing comprehensive benchmarks to assess existing generation methods. Additionally, exploring the transition from static to dynamic evaluation systems will be crucial for the continued advancement of data generation. While we acknowledge that our collection of metrics may not be exhaustive, the Quality-Trustworthiness framework is designed to be open and extensible, allowing for the integration of new and valuable metrics as the field evolves.

We hope this survey serves as a foundation for high-quality and trustworthy LLM-based data generation, enabling the community to develop more robust and reliable data generation systems.

## Acknowledgments

We thank the anonymous reviewers and the action editor for their constructive feedback. This work was supported in part by the U.S. National Science Foundation under Award Nos. 2431515 and 2450662. The views expressed are those of the authors and do not necessarily reflect the views of the National Science Foundation.

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

# A  Notation Table

In this section, we provide the main notations we used in the survey.

Table 9: Unified notation for core data objects across modalities.

| Symbol | Meaning |
|--------|---------|
| **Shared** | |
| $E(\cdot)$ | Encoder / embedding function |
| $\text{sim}(\cdot, \cdot)$ | Similarity (e.g., cosine) |
| **Text / Dialogue** | |
| $\mathcal{T}$ | Set of generated texts |
| $t_i$ | $i$-th generated text, $t_i \in \mathcal{T}$ |
| $p_i$ | Prompt / instruction for $t_i$ |
| $\mathcal{D}$ | Labeled evaluation dataset |
| $y_i, \hat{y}_i$ | Gold / predicted label |
| $w_m, L$ | $m$-th token; sequence length |
| $\tau$ | Acceptability threshold |
| $\mathcal{X}$ | Corpus (set of instances) |
| **Symbolic / Reasoning** | |
| $\mathcal{R}$ | Set of reasoning examples $(q_i, c_i, y_i)$ |
| $q_i$ | Question / problem for example $i$ |
| $c_i$ | Chain-of-thought / rationale for example $i$ |
| $c_{i,t}$ | $t$-th step in chain $c_i$ |
| $y_i$ | Final answer / conclusion for example $i$ |
| $c_{i,k}$ | $k$-th sampled chain for the same question $q_i$ |
| $y_{i,k}$ | Final answer induced by chain $c_{i,k}$ |
| $K$ | Number of sampled chains per question |
| **Tabular** | |
| $X_{\text{real}}, X_{\text{syn}}$ | Real / synthetic tables |
| $x_i$ | Row (record) in a table |
| $y_i$ | Label / target for $x_i$ (if any) |
| $d$ | Number of features (columns) |
| **Graphs / JSON / Logs** | |
| $\mathcal{G}$ | Set of graphs (real or generated) |
| $G$ | A graph; $V(G), E(G)$: nodes / edges |
| $\mathcal{J}$ | Set of JSON / structured records |
| $J$ | A JSON key–value object |
| $\mathcal{L}$ | Set of log entries or sequences |
| $\ell_i$ | A log entry or a log sequence |
| **Vision-Language** | |
| $\mathcal{M}$ | Set of multi-modal samples |
| $m_i$ | $i$-th sample, e.g., $(v_i, t_i)$ |
| $v_i$ | Visual / audio / video input paired with text |
| $k$ | Number of modalities in $m_i$ |
| **Agent / Interaction** | |
| $\mathcal{A}$ | Set of agent trajectories / episodes |
| $\mathcal{A}_{\text{gen}}$ | Set of generated trajectories $\{\tau_i\}_{i=1}^N$ |
| $N$ | Number of trajectories / episodes in $\mathcal{A}_{\text{gen}}$ |
| $\tau_i$ | $i$-th trajectory $(s_0, a_0, r_1, s_1, \ldots, a_{T_i-1}, r_{T_i}, s_{T_i})$ |
| $T_i$ | Horizon (number of decision steps) of trajectory $\tau_i$ |
| $s_t$ | State / observation at step $t$ |
| $a_t$ | Action (incl. tool/API call) at step $t$ |
| $r_{t+1}$ | Reward / feedback for transition $(s_t, a_t, s_{t+1})$ |

## B    Related Work

We summarize related works and conduct a comparison with LLM Data Auditor across several dimensions, including Primary Focus, Organization, Quality, Trustworthy Evaluation, and Scope.

Table 10: Comparative Analysis of Surveys in Synthetic Data. While prevailing surveys structure the field by engineering workflows (Long et al., 2024), training lifecycles (Wang et al., 2024a), or specific modalities (Shi et al., 2025b), this work establishes a metric-centric taxonomy focused on the intrinsic evaluation and governance of cross-modal data.

| Dimension | Engineering Workflows (Long et al., 2024) | Training Lifecycles (Wang et al., 2024a) | Specific Modalities (Shi et al., 2025b) | This Work |
|---|---|---|---|---|
| Primary Focus | **The Engineering Pipeline**. Focuses on the operational workflow of constructing data, emphasizing generation techniques, curation strategies, and downstream application. | **The Model Lifecycle**. Focuses on the utility of synthetic data across distinct training stages, spanning pre-training, supervised fine-tuning, and alignment. | **The Generative Method**. Focuses on methodological families within the structured data domain, contrasting GANs, Variational Autoencoders, and Diffusion models. | **The Data Artifact**. Focuses on the intrinsic properties of the generated product, prioritizing rigorous auditing standards and data governance protocols. |
| Organization | **Procedure-Oriented**. Taxonomy defined by operational modules including prompt engineering, task decomposition, and heuristic filtering. | **Stage-Oriented**. Taxonomy structured around the LLM development lifecycle and downstream competencies such as reasoning and coding. | **Model-Centric**. Taxonomy categorized by the underlying generative architecture and associated post-processing techniques for tabular structures. | **Metric-Oriented**. Taxonomy defined by a unified evaluation coordinate system separating Quality dimensions from Trustworthiness dimensions. |
| Quality Evaluation | **Extrinsic Utility**. Quality is frequently judged by downstream gains, complemented by lightweight intrinsic proxies. | **Benchmark-Based**. Effectiveness is assessed via success rates on capability-specific evaluation suites and standard public benchmarks. | **Statistical Fidelity**. Evaluation prioritizes distributional resemblance to real data, machine learning efficacy, and column-wise statistical alignment. | **Intrinsic Verification**. Quality is defined through proactive audits of Validity, Fidelity, Diversity, and Utility prior to model training. |
| Trustworthy Evaluation | **Challenge-Oriented**. Hallucination and bias are discussed primarily as open challenges or limitations. | **Alignment-Oriented**. Safety is framed within the context of RLHF alignment. | **Privacy-Specific**. Analysis heavily concentrates on privacy guarantees. | **Foundational Pillar**. Elevates mainstream trustworthiness to a primary dimension orthogonal to utility metrics. |
| Scope | **Text-Dominant**. Primarily covers natural language processing tasks. | **Broad**. This work covers multiple modalities such as text, code, and vision. | **Tabular Data Specialized**. Addresses the constraints inherent to tabular data. | **Cross-Modality**. Applies a single framework to introduce 6 modal data. |

