# OpenReview forum: "A Survey on Evaluating Quality and Trustworthiness in LLM-Generated Data"
_TMLR — Accepted by TMLR_

### Review · Reviewer_xkeb · 2026-03-15

**Summary Of Contributions:**

- While the extrinsic utility of synthetic data (i.e., improving downstream performance) remains the primary goal, the paper provides a well-motivated argument forthe necessity of evaluating its intrinsic quality.
- The survey covers diverse modalities beyond text, including tabular, image, and agent data, reflecting the expanded task range of modern LLMs.
- The manuscript is well-structured, applying a consistent format to each data modality by discussing generation methods, evaluation metrics, the evaluation-practice gap, and relevant use cases.
- The focused analysis on Quality and Trustworthiness provides depth but results in the omission of Diversity, which is an essential desideratum for synthetic data. Since a dataset's utility is defined by its coverage of the underlying distribution, the limited discussion on diversity metrics narrows the evaluation framework; high-fidelity samples are of limited use if they lack the variety required for robust applications.

**Audience:**

Yes

**Audience Explanation:**

- Yes. Synthetic data generation and its evaluation are central topics in current machine learning research. Researchers working across different modalities will find the survey a useful reference for understanding the current landscape of synthetic data.

**Broader Impact Concerns:**

- While the manuscript incorporates ethical considerations by discussing trustworthiness (including fairness and privacy), these discussions primarily focus on technical metrics. It would be better to add a brief discussion to the Conclusion regarding broader societal implications (such as the potential risks of synthetic data in reinforcing systemic biases).

**Claims And Evidence:**

Yes

**Claims Explanation:**

- The manuscript provides an organized review across multiple modalities, and the evidence for identified evaluation-practice gaps is consistently drawn from a broad range of existing literature.
- Although the "Evaluation Gap Analysis" is a literature-based systematic review rather than an experimental re-validation, the current manuscript is sufficient for a survey to provide a clear overview of the research landscape.

**Requested Changes:**

- The evaluation axes in Tables 2–7 vary across modalities. While modality-specific metrics are necessary, a unified evaluation taxonomy would provide a more coherent framework for the survey. For example, diversity is currently omitted for (3) symbolic and (5) semi-structured data, despite being a general requirement for synthetic data. I recommend applying a consistent set of evaluation dimensions across all tables; if a specific dimension is inapplicable to a certain modality, this exclusion should be explicitly stated and justified.
- I would encourage the authors to consider whether Diversity could be elevated to a primary evaluation axis alongside Quality and Trustworthiness. While the current taxonomy well-defines the latter two, diversity is only partially addressed for certain modalities. In my view, the core value of synthetic data lies in providing a large, diverse volume of high-quality and trustworthy samples, and acknowledging diversity as a third pillar would better reflect these fundamental requirements. (Note: This is merely a suggestion to potentially strengthen the conceptual framework. I recognize that such a change might require substantial restructuring; as an alternative, simply complementing the missing discussions on diversity for the specific modalities where it is currently omitted would be sufficient.)

---

> ### Author Response · Authors · 2026-04-07
> **Thank you! (1/1)**
>
> **[W1]:** The evaluation axes in Tables 2–7 vary across modalities. While modality-specific metrics are necessary, a unified evaluation taxonomy would provide a more coherent framework for the survey. For example, diversity is currently omitted for (3) symbolic and (5) semi-structured data, despite being a general requirement for synthetic data. I recommend applying a consistent set of evaluation dimensions across all tables; if a specific dimension is inapplicable to a certain modality, this exclusion should be explicitly stated and justified.
>
>
> **[Answer 1]:** Thank you for this valuable suggestion. We agree that the common evaluation taxonomy can be more conceptually consistent across modalities.
>
> To address the concern, we have made the following targeted updates in the revised manuscript:
>
> * **Standardized "Diversity" Across Modalities:** We fixed the specific inconsistency you pointed out by adding diversity metrics to the previously missing sections. Specifically, symbolic/logical reasoning now includes $Div(q_i)$, and semi-structured data now includes *Novelty* and *Uniq*. Diversity is now explicitly represented across all modality sections.
> * **Added Cross-Modal Synthesis (Section 8 & Table 11):** We introduced a new section to clarify the unified structure underlying the modality-specific tables. We now explicitly categorize evaluation dimensions into three roles: *foundational* (e.g., Validity, Fidelity), *transferred but modality-dependent* (e.g., Diversity, Utility), and *modality-concentrated* (e.g., Privacy, Fairness).
>
> These revisions clarify that our unification lies in the overarching evaluation *lens*, justifying structural differences as modality-specific operationalizations rather than accidental omissions.
>
>
> **[W2]:** I would encourage the authors to consider whether Diversity could be elevated to a primary evaluation axis alongside Quality and Trustworthiness. While the current taxonomy well-defines the latter two, diversity is only partially addressed for certain modalities. In my view, the core value of synthetic data lies in providing a large, diverse volume of high-quality and trustworthy samples, and acknowledging diversity as a third pillar would better reflect these fundamental requirements. (Note: This is merely a suggestion to potentially strengthen the conceptual framework. I recognize that such a change might require substantial restructuring; as an alternative, simply complementing the missing discussions on diversity for the specific modalities where it is currently omitted would be sufficient.)
>
>
> **[Answer 2]:** Thank you for this thoughtful suggestion. We opted to preserve our core two-axis architecture (Quality and Trustworthiness) to maintain the structural consistency of the framework, figures, and contributions. Instead, we have substantially strengthened the explicit role of Diversity within the existing framework:
>
> * **Expanded Coverage:** We added Diversity to the previously missing modalities (symbolic/logical reasoning and semi-structured data). It is now explicitly evaluated across all six modality sections.
> * **Elevated Cross-Modal Role (Section 8 & Table 11):** In our new cross-modal synthesis, we explicitly highlight Diversity as a critical *transferred but modality-dependent* dimension across all data types.
>
>
> **[W3]:**  It would be better to add a brief discussion to the Conclusion regarding broader societal implications (such as the potential risks of synthetic data in reinforcing systemic biases).
>
> **[Answer 3]:** Thank you for this valuable comment. We have added a brief discussion to the Conclusion and highlighted the changes in red.

---

### Review · Reviewer_2DvE · 2026-03-22

**Summary Of Contributions:**

The paper introduces a LLM Data Auditor framework, a metric-oriented survey that evaluates the quality and trustworthiness of synthetic data generated by LLMs across six modalities. I think one of the paper's strongest contributions is its attempt to shift the conversation from how to generate synthetic data to how to evaluate synthetic data, especially by evaluating trustworthiness dimensions alongside quality.

However, I want to note that I am not an expert in this area, so I will focus on my general impressions here:

Strength:
   * The overall structure is unified and the coverage is quite broad.
   * Table 2 to 7 are quire informative, although there are some issues with mixing object types (see weaknesses).
   * The paper goes beyond standard quality metrics to include trustworthiness metrics.

Weakness:
   * I think one of the main issues here is taxonomy drift. The paper claims a unified framework, but the categories are not stable across sections. For example, the framework figure (Figure 1) describes trustworthiness primarily through fairness, robustness, and faithfulness, while the contribution bullet points emphasize fairness, robustness, and privacy.
   * The paper also frequently uses language like "we use", "we adopt", which read as though the authors are proposing and applying a evaluation framework. However, as a survey, there are not actual experiments or empirical validation. I think if the framework is prescriptive, it should be validated on at least one modality, if descriptive, the language should be adjusted accordingly.
   * The unified cross-modal claim is a bit overclaimed. The actual metrics used across modalities are largely different and modality-specific. For example, it's unclear how "fidelity" for tabular data (KST) relates conceptually to "fidelity" for graphs (MMD).
   * Some tables appear to mix generation methods with metrics. For example, in table 2, I think RARR and PMI-FAITH are both metrics, yet they are listed alongside generation methods.

**Audience:**

Yes

**Audience Explanation:**

I think this is a timely and broad survey on an important topic, and I do think the general area is relevant to TMLR.

**Claims And Evidence:**

Yes

**Claims Explanation:**

I think the claims are partially supported but not yet fully convincing. Please refer to the weaknesses.

**Requested Changes:**

Please see the weaknesses for main concerns. Some additional suggestions:
   * Page 6, “ CoLA benchmark.Building on this practice,”, space in front of building
   * I found the metrics sections quite hard to read. It is difficult to tell which metrics are established ones and which are newly proposed or adapted. Adding a summary table at the end of each section would strengthen this.
   * Table 9 (the last table) seems unfinished? It has duplicated citations.

---

> ### Author Response · Authors · 2026-04-07
> **Thank you! (1/2)**
>
> **[W1]:** I think one of the main issues here is taxonomy drift. The paper claims a unified framework, but the categories are not stable across sections. For example, the framework figure (Figure 1) describes trustworthiness primarily through fairness, robustness, and faithfulness, while the contribution bullet points emphasize fairness, robustness, and privacy.
>
>
>
> **[Answer 1]:** We thank the reviewer for this insightful comment. We agree that the previous presentation could inadvertently create an impression of "taxonomy drift," as the Introduction, the overview surrounding Figure 1, and the summary text referenced different illustrative subsets of trustworthiness.
>
> To resolve this concern, we have carefully aligned these descriptions in the revised manuscript. We now make it explicitly clear that our unification occurs at the level of a **metric-oriented evaluation framework** anchored by two major pillars—*Quality* and *Trustworthiness*—rather than requiring every overview component to exhaustively enumerate the exact same list of sub-dimensions.
>
> This approach reflects the intended structural design of our paper. To ensure this is transparent to the reader, we have clarified the specific roles of our figures and tables:
>
> * **Clarified the Role of Figure 1:** We have revised the high-level presentation to explicitly state that **Figure 1** serves strictly as a schematic overview of the two-pillar architecture.
> * **Pointed to the Exhaustive Taxonomy (Table 1):** We direct readers to **Table 1** for the complete, unified evaluation taxonomy across all modalities.
> * **Integrated the Cross-Modal Synthesis (Section 8 & Table 11):** We emphasize that **Section 8** and **Table 11** provide a complementary cross-modal synthesis over this exact same evaluation space. By regrouping dimensions into foundational, transferred, and modality-concentrated roles, we demonstrate that the concrete instantiation of trustworthiness dimensions can vary by modality and use case *without* implying any change or drift in the underlying taxonomy.
>
> We believe these revisions successfully eliminate any appearance of taxonomy drift while firmly preserving the central contribution of our paper: delivering a unified, metric-oriented framework for organizing and auditing the evaluation of LLM-generated synthetic data across diverse modalities.
>
> **[W2]:** The paper also frequently uses language like "we use", "we adopt", which read as though the authors are proposing and applying a evaluation framework. However, as a survey, there are not actual experiments or empirical validation. I think if the framework is prescriptive, it should be validated on at least one modality, if descriptive, the language should be adjusted accordingly.
>
> **[Answer 2]:** Thank you for this insightful comment. We agree with reviewer that some of our earlier phrasing (e.g., expressions such as “we use” or “we adopt”) may have inadvertently caused the manuscript to read more prescriptively than intended.
>
>
> To directly address your concern and align the text with the true nature of the paper, we have made the following revisions:
>
> * **Adopted Neutral, Survey-Oriented Language:** We have thoroughly reviewed and revised the manuscript to employ neutral, descriptive language throughout. We systematically replaced prescriptive formulations (such as “we use” or “we adopt”) with phrasing that accurately reflects our role as a survey. For instance, we now use expressions like “can be evaluated,” “can be utilized,” or “the literature commonly evaluates.” These textual edits ensure it is explicitly clear that our framework organizes and summarizes existing practices, rather than prescribing a newly validated experimental procedure.
> * **Clarified the Descriptive Nature of the Contribution:** We have further refined our phrasing to emphasize that the core contribution of this work is purely organizational and descriptive. We provide a unified evaluation space for analyzing prior literature and utilize representative audits to identify under-reported dimensions, rather than claiming the experimental validation of a novel benchmark or protocol.

---

> ### Author Response · Authors · 2026-04-07
> **Thank you! (2/2)**
>
> **[W3]:** The unified cross-modal claim is a bit overclaimed. The actual metrics used across modalities are largely different and modality-specific. For example, it's unclear how "fidelity" for tabular data (KST) relates conceptually to "fidelity" for graphs (MMD).
>
> **[Answer 3]:** We thank the reviewer for providing this comment and example. We agree with the reviewer that concrete metric formulas are inherently modality-specific.
>
> To clarify, our intended cross-modal claim is *not* that identical metric formulas apply uniformly across all data types. Instead, the unification we propose operates strictly at the macro-level of **evaluation dimensions**. As we now explicitly outline in **Section 8**, our framework organizes the evaluation space into shared conceptual dimensions, while acknowledging that their concrete operationalization must remain highly modality-dependent.
>
> Using the exact example you raised: *Fidelity* in tabular data is instantiated via distribution-matching measures such as KST or TVD, whereas in graph or semi-structured settings, it necessitates metrics like MMD or FCD. We fully agree that these are not interchangeable metrics. Rather, they serve as distinct, modality-specific realizations of the exact same higher-level conceptual dimension (Fidelity).
>
> To ensure this nuance is perfectly clear and to prevent any misinterpretation, we have made the following refinements in the revised manuscript:
>
> * **Clarified the Scope of Unification (Section 8):** We have explicitly distinguished between *dimension-level conceptual unification* and *metric-level operationalization*, making it clear that shared dimensions require tailored, modality-specific mathematical instantiations.
>
>
>
>
>
>
> **[W4]:** Some tables appear to mix generation methods with metrics. For example, in table 2, I think RARR and PMI-FAITH are both metrics, yet they are listed alongside generation methods.
>
> **[Answer 4]:** Thank you for this comment. To resolve this confusion, we have made the following structural and specific corrections in the revised manuscript:
>
> * **Corrected the Entry:** We replaced the metric PMI-FAITH with the actual representative work **PMI-DECODE** in the audit table.
> * **Renamed Table Headers:** We changed the table heading from "Generation Methods" to **"Representative Work"** to accurately reflect that not every entry is a generation pipeline.
> * **Added "Family" and "Role" Columns (Section 1.1):** We introduced explicit columns to indicate each work's *Family* and *Role* (e.g., Generation, Benchmark, Evaluation). Under this new presentation, entries like RARR and PMI-DECODE are clearly tagged as *Evaluation* works rather than generation methods.
>
>
>
>
>
> **[W5]:** Page 6, “ CoLA benchmark.Building on this practice,”, space in front of building
>
> **[Answer 5]:** Thanks for this comment. We have modified this typo.
>
>
>
> **[W6]:** I found the metrics sections quite hard to read. It is difficult to tell which metrics are established ones and which are newly proposed or adapted. Adding a summary table at the end of each section would strengthen this.
>
> **[Answer 6]:** Thank you for this helpful comment. We clarify that this survey is descriptive rather than prescriptive: we do not propose new metrics, but summarize and organize established metrics from prior work under a unified taxonomy. The equations and formulations are included only to present these existing metrics in a consistent survey notation for cross-modal comparison. To reduce any ambiguity, we also revised the metric sections to use more neutral wording, so that they read as literature synthesis rather than newly proposed evaluation methods.
>
>
> **[W7]:** Table 9 (the last table) seems unfinished? It has duplicated citations.
>
> **[Answer 7]:** Thank you for this comment. We added the comparison columns Engineering Workflows, Training Lifecycles, and Specific Modalities, and also cleaned up the duplicated citation issue. This makes the comparison much clearer.

---

### Review · Reviewer_z3Pq · 2026-03-28

**Summary Of Contributions:**

As LLMs become the dominant engine for synthetic data generation, ensuring the quality of the resulting data is critical. Yet current research overwhelmingly focuses on generation methodologies, with limited attention to systematic evaluation of the data itself. Most existing evaluations are extrinsic and modality-specific, lacking a unified intrinsic perspective. This survey proposes the "LLM Data Auditor" framework to bridge this gap. The framework covers six data modalities including text, symbolic/logical reasoning, tabular, semi-structure, vision-language, and agent data (embodied AI, autonomous driving, decision-making) under a unified five-stage pipeline: (1) LLM-based generation methods, (2a) quality metrics organized into validity, fidelity, diversity, and utility, (2b) trustworthiness metrics covering safety, fairness, privacy, faithfulness, and robustness, (3) evaluation gap analysis auditing representative works against these dimensions, and (4) practical data usage patterns. For each modality, the paper provides formal mathematical definitions of representative metrics, surveys generation techniques (from corpus curation and prompt-driven methods to alignment-based and inference-time approaches), and produces gap analysis tables that identify which evaluation dimensions are missing in current literature. Key findings include that validity and safety are rarely evaluated with explicit metrics in text generation, fairness evaluation in tabular data generation is notably underdeveloped, and trustworthiness metrics are broadly neglected across most modalities. The paper concludes with forward-looking discussions on model collapse detection, redefining fidelity from distributional mimicry to process verifiability for reasoning-intensive domains, and navigating the trust-utility Pareto frontier. The survey is accompanied by a public repository cataloging methods and metrics.

**Audience:**

Yes

**Audience Explanation:**

As LLM-generated synthetic data becomes ubiquitous across ML pipelines, the lack of systematic evaluation frameworks is a genuine gap. This survey correctly identifies that the community has focused heavily on generation methods while under-investing in intrinsic data quality assessment. The shift from extrinsic to intrinsic evaluation is a valuable conceptual reorientation.

**Broader Impact Concerns:**

I do not have any concerns on the ethical implications.

**Claims And Evidence:**

Yes

**Claims Explanation:**

- The Quality/Trustworthiness taxonomy and the formal metric definitions are the strongest part of the paper. Each metric is presented with a mathematical formula grounded in cited prior work. These are reproducible and clearly specified. However, the paper does not provide evidence that this particular organization of metrics is superior to alternatives -- the taxonomy is asserted rather than derived or validated.

- The gap claims are qualitative, not quantitative. Statements like "fairness evaluation in tabular data generation remains notably underdeveloped" are supported only by showing × entries in Table 4. There is no systematic count of how many papers across the field evaluate fairness, no meta-analytic methodology, and no statistical characterization of the gap.

- The paper claims the LLM Data Auditor framework provides "a comprehensive guide for generating and evaluating high-quality, multi-modal data." However, no empirical evidence demonstrates that using this framework improves data quality or reveals issues that would otherwise be missed. The framework is never applied end-to-end to audit any synthetic dataset. There are no case studies showing, for example, that a dataset passing extrinsic evaluation fails intrinsic evaluation under the proposed metrics, which would be the most convincing evidence for the paper's central thesis.

- The paper claims to provide "unified cross-modal coverage," but the evidence for cross-modal insights is thin. The parallel structure mechanically applies the same section template to each modality, but there is limited synthesis showing how evaluation challenges or solutions transfer across modalities. The conclusion mentions cross-modal patterns (e.g., trustworthiness is broadly neglected) but these observations are surface-level and not systematically derived.

**Requested Changes:**

- The parallel modality-by-modality structure provides breadth but the paper currently lacks a section that synthesizes findings across modalities. I would suggest the authors to include discussions on which metrics transfer across modalities and which are modality-specific.
- Discuss benchmark contamination as a trustworthiness dimension. LLM-generated synthetic data risks contaminating evaluation benchmarks. This is a growing concern and fits naturally within the trustworthiness pillar but is not discussed.
- The gap analysis audits only 5-6 methods per modality, but the paper does not explain the selection criteria. State explicitly whether methods were chosen by citation count, recency, diversity of approach, or another criterion. Ideally, expand the tables to cover a broader sample or provide evidence that the selected methods are representative of the field. A biased sample undermines the generalizability of the gap claims.

---

> ### Author Response · Authors · 2026-04-07
> **Thank you! (1/4)**
>
> **[W1]:** However, the paper does not provide evidence that this particular organization of metrics is superior to alternatives -- the taxonomy is asserted rather than derived or validated.
>
>
> **[Answer 1]:** We appreciate your feedback on the strength of our formal metric definitions and the Quality/Trustworthiness taxonomy. To clarify, our framework is designed as a motivated and operationalized lens, rather than being empirically proven as universally superior to all alternative organizations. Our primary objective is not to claim a uniquely optimal taxonomy, but rather to establish an urgently needed *audit-oriented* framework to systematize the currently fragmented landscape of intrinsic evaluation for LLM-generated data.
>
> As discussed in the paper, while existing surveys typically organize the literature around workflows, lifecycles, or single modalities, we adopt the Quality/Trustworthiness axis because *"the literature still lacks a unified framework to audit synthetic data before it enters the training loop."* We have carefully revised the manuscript:
>
> * **Clarified Motivation in the Introduction:** We have explicitly stated that *"we adopt a data-based perspective and establish metrics as our primary organizing principle."* We further describe the *LLM Data Auditor* as a unified structure covering generation methods, quality metrics, trustworthiness metrics, evaluation gaps, and data usage.
> * **Added Section 1.1 (“Representative-Work Audit Protocol”):** This new section clarifies that our proposed framework and the accompanying tables are designed to serve as representative audits for interpretable gap analysis, rather than as evidence for a universally definitive or optimal taxonomy.
>
> **[W2]:** Statements like "fairness evaluation in tabular data generation remains notably underdeveloped" are supported only by showing × entries in Table 4. There is no systematic count of how many papers across the field evaluate fairness, no meta-analytic methodology, and no statistical characterization of the gap.
>
> **[Answer 2]:** We thank the reviewer for this comment. As you pointed out, our survey is designed as a systematic, metric-oriented framework rather than a formal meta-analysis or an exhaustive statistical census of the entire literature.
>
> To explicitly establish this methodological boundary and address your concern, we have made the following specific revisions to the manuscript:
>
> * **Clarified the Scope in Section 1.1 (“Representative-Work Audit Protocol”):** We now explicitly state that Tables 2–7 are structured as *representative audits* rather than an exhaustive census. We clarify that the resulting gap analysis should be interpreted as a comparative snapshot of a representative sample, rather than a field-wide prevalence estimate derived from a meta-analysis.
>
>
>
>
>
> * **Tightened the Gap-Analysis Claims:** We have refined our wording throughout to strictly reflect what is supported by the audited sample. For instance, in the tabular data section, instead of relying on subjective terminology like “underdeveloped,” we now report the within-audit observation directly: *fairness-oriented evaluation appears in only 2 of the 8 audited representative tabular works.* Our intent is not to claim an exact prevalence rate for the entire field, but to highlight critical evaluation dimensions that are consistently under-reported within a representative cross-family audit.
>
> At the same time, we would like to emphasize that the core value of our work does not rest on exhaustive counting. The primary contribution of this survey is the introduction of a **unified, metric-oriented audit framework** for synthetic data evaluation across six modalities, alongside a structured representative-work audit protocol that makes under-reported evaluation dimensions visible in a transparent and actionable way. In other words, our goal is to offer the community a cross-modal evaluation lens, a consistent taxonomy, and an interpretable audit template that can guide more rigorous reporting and future benchmark design.
>
> We appreciate this comment, as it has helped us significantly sharpen both the methodological framing and the precision of our conclusions. The revised manuscript now clearly bounds the scope of our gap analysis while preserving the paper's central contribution as a systematic framework for auditing quality and trustworthiness in LLM-generated synthetic data.

---

> ### Author Response · Authors · 2026-04-07
> **Thank you! (2/4)**
>
> **[W3]:** The paper claims the LLM Data Auditor framework provides "a comprehensive guide for generating and evaluating high-quality, multi-modal data." However, no empirical evidence demonstrates that using this framework improves data quality or reveals issues that would otherwise be missed. The framework is never applied end-to-end to audit any synthetic dataset. There are no case studies showing, for example, that a dataset passing extrinsic evaluation fails intrinsic evaluation under the proposed metrics, which would be the most convincing evidence for the paper's central thesis.
>
>
> **[Answer 3]:** Thank you for your constructive suggestion regarding the inclusion of an end-to-end case study. We would like to respectfully clarify that conducting an empirical, end-to-end case study falls outside the intended scope of this paper.
>
> As a survey, the primary intent of our work is to provide a comprehensive synthesis of fragmented evaluation practices and to systematize them into a unified, taxonomy-based auditing lens across six distinct modalities. Our objective is to establish the theoretical and structural groundwork for comprehensive evaluation, rather than to empirically validate specific data generation pipelines. For empirical validations, readers can refer to existing domain-specific studies that investigate subsets of the metrics we have surveyed. And all the metrics have been validated in literature. This survey paper aims to summarize the available metrics in a structured way and better inform researchers how to use these metrics.
>
>
>
> Nevertheless, we fully recognize the importance of your suggestion. To incorporate your feedback and ensure our boundaries are perfectly clear, we have made the following updates to the revised manuscript:
>
> * **Added an Explicit Limitations in Conclusion Section:** We have added a paragraph discussing the limitations of our current scope, explicitly clarifying that this work focuses on taxonomy and synthesis rather than empirical execution, and highlighting end-to-end empirical auditing as a critical direction for future research.
> * **Strengthened the Motivation for Multi-Dimensional Auditing:** To further justify the necessity of our proposed framework without conducting a new case study, we have expanded our motivation section. We now cite recent empirical works [1, 2] demonstrating that narrow or simplistic evaluation protocols often fail to detect critical utility, fairness, and privacy issues. This reinforces why a unified auditing lens is urgently needed before models enter the training loop.
>
>
> **References added to the manuscript:**
>
> [1] Pereira M, Kshirsagar M, Mukherjee S, et al. Assessment of differentially private synthetic data for utility and fairness in end-to-end machine learning pipelines for tabular data. *PLOS ONE*, 2024, 19(2): e0297271.
>
> [2] Ganev G, De Cristofaro E. The inadequacy of similarity-based privacy metrics: Privacy attacks against “truly anonymous” synthetic datasets. *2025 IEEE Symposium on Security and Privacy (SP)*. IEEE, 2025: 4007-4025.

---

> ### Author Response · Authors · 2026-04-07
> **Thank you! (3/4)**
>
> **[W4]:** The paper claims to provide "unified cross-modal coverage," but the evidence for cross-modal insights is thin. The parallel structure mechanically applies the same section template to each modality, but there is limited synthesis showing how evaluation challenges or solutions transfer across modalities. The conclusion mentions cross-modal patterns (e.g., trustworthiness is broadly neglected) but these observations are surface-level and not systematically derived.
>
>
> **[Answer 4]:** We thank the reviewer for this constructive feedback. To directly address this concern and elevate the manuscript beyond parallel presentation, we have made substantial additions to synthesize our findings across the six modalities covered in the survey:
>
> * **Introduced Section 8 and Table 11 (“Cross-Modal Synthesis of Evaluation Dimensions and Gaps”):** This brand-new section is specifically designed to provide a deep, interpretive synthesis. Rather than merely reiterating the modality-by-dimension inventory, **Table 11** actively reorganizes the entire evaluation space into three distinct cross-modal roles: *(i) foundational dimensions, (ii) transferred but modality-dependent dimensions, and (iii) modality-concentrated dimensions*.
> * **Deepened Interpretive Analysis (Sections 8.1 & 8.2 & 8.3):** In **Section 8.1**, we explicitly establish that Table 11 serves as a “cross-modal interpretive layer,” making comparative patterns much more explicit than the previous modality-by-modality presentation. In **Section 8.2**, we further synthesize our findings by identifying *Validity* and *Fidelity* as the most stable shared foundations. Conversely, we conceptualize *Diversity, Utility, Faithfulness, Safety,* and *Robustness* as crucial concerns that transfer conceptually across modalities but fundamentally require modality-specific operationalization. In **Section 8.3**, we analyze the Modality-Concentrated Dimensions, which include *Privacy, Fairness, Provenance,* and *Benchmark Contamination*.
> * **Refined the Conclusion:** We have thoroughly revised the conclusion to avoid overly broad cross-modal claims and align it perfectly with our new synthesis. The updated conclusion emphasizes the necessity of multi-dimensional post-generation evaluation and the critical trade-offs among dimensions. It also carefully reiterates that our work is a conceptual and analytical synthesis rather than an end-to-end empirical validation.
>
> Through these revisions, we wish to clarify that our intention is not to claim that identical evaluation metrics transfer seamlessly or remain unchanged across all modalities. Instead, we demonstrate that a **unified evaluation lens** reveals underlying structural patterns: highlighting which dimensions are consistently foundational, which require re-instantiation upon transfer, and which are distinctly salient to specific data types.
>
> **[W5]:** The parallel modality-by-modality structure provides breadth but the paper currently lacks a section that synthesizes findings across modalities. I would suggest the authors to include discussions on which metrics transfer across modalities and which are modality-specific.
>
> **[Answer 5]:** We thank the reviwer for this valuable suggestion. To directly address this point in the revised manuscript, we have incorporated **Section 8 (“Cross-Modal Synthesis of Evaluation Dimensions and Gaps”)** along with **Table 11**. Specifically, we have restructured our discussion to clearly articulate both transferability and modality-specificity:
>
> * **Introduced a Cross-Modal Interpretive Layer (Section 8.1):** We explicitly establish that Table 11 goes beyond the parallel modality-by-modality organization to provide a dedicated cross-modal interpretive layer.
> * **Categorized Transferability vs. Specificity (Section 8.2 & Table 11):** We systematically differentiate the evaluation dimensions into three distinct transferability profiles:
>   * *Stable Shared Foundations:* We identify **Validity** and **Fidelity** as the most consistent foundational metrics across all six data types.
>   * *Transferable but Modality-Dependent:* We demonstrate that dimensions such as **Diversity, Utility, Faithfulness, Safety,** and **Robustness** conceptually transfer across modalities but fundamentally require modality-specific operationalization.
>   * *Modality-Concentrated:* In Table 11, we highlight **Privacy, Fairness, Provenance,** and **Benchmark Integrity/Contamination** as dimensions that become particularly salient and explicit only in specific data settings.
>
> With these updates, the revised paper now provides an explicit, structured discussion of metric transferability and modality-specificity, moving away from leaving these critical insights merely implicit in the section structure.
>
> *(Note: This comment is closely aligned with the concerns raised in Comment 4. Our complete rationale regarding the cross-modal synthesis can also be found in our response to W4).*

---

> ### Author Response · Authors · 2026-04-07
> **Thank you! (4/4)**
>
> **[W6]:** Discuss benchmark contamination as a trustworthiness dimension. LLM-generated synthetic data risks contaminating evaluation benchmarks. This is a growing concern and fits naturally within the trustworthiness pillar but is not discussed.
>
> **[Answer 6]:** Thank you for this important and timely suggestion. We agree that addressing data contamination and benchmark integrity is crucial for a comprehensive trustworthiness analysis, especially given the rapid proliferation of LLM-generated data.
>
> To fully incorporate your suggestion, we have substantially revised the manuscript to explicitly discuss benchmark integrity and contamination within our trustworthiness framework. Specifically, we have made the following updates:
>
> * **Expanded Trustworthiness Analysis for Text Data (Section 2.3):** We now formally define benchmark contamination as the risk of overlap between generated/training data and evaluation benchmarks, which artificially inflates downstream scores and compromises evaluation integrity. Furthermore, we have summarized several practical audit protocols to detect this, including *Testset Slot Guessing (TS-Guessing), retrieval-based overlap search, perplexity-based memorization analysis,* and *output-distribution-based detection*.
>
>
>
> * **Addressed Contamination in Symbolic/Reasoning Data (Section 3.3):** For reasoning tasks, we discuss contamination through the lens of overlap between benchmark problems (or reference solutions) and training corpora. We also summarize representative quantitative formulations used in the literature, such as the *overlap-based contamination rate* and the *time-segmented cutoff gap*.
> * **Updated Representative Audits (Tables 3 & 4):** To operationalize this addition, we have introduced a dedicated **"Benchmark Contam."** column to Tables 3 and 4. Works that proactively implement contamination-aware evaluations are now explicitly marked in our audit.
>
> * **Added Benchmark Contamination to Evaluation Practice Gap analysis (Section 2.4 & 3.4):** We have explicitly integrated a discussion of benchmark contamination into the analysis of the Evaluation Practice Gap for both text data and symbolic/reasoning data.
>
>
> * **Integrated into Cross-Modal Synthesis (Table 11):** Finally, in our newly added cross-modal synthesis section, we treat benchmark integrity and contamination as a critically important *modality-concentrated concern*, noting its particular salience for text and reasoning data ecosystems.
>
>
>
>
>
> **[W7]:** The gap analysis audits only 5-6 methods per modality, but the paper does not explain the selection criteria. State explicitly whether methods were chosen by citation count, recency, diversity of approach, or another criterion. Ideally, expand the tables to cover a broader sample or provide evidence that the selected methods are representative of the field. A biased sample undermines the generalizability of the gap claims.
>
> **[Answer 7]:** Thank you for this constructive comment. We agree that the selection rationale for our gap-analysis tables required explicit articulation to preclude any concerns regarding sampling bias.
>
> To address this, we have made our methodology fully transparent in the revised manuscript:
>
> * **Formalized the Audit Protocol (Section 1.1):** We have added a dedicated *Representative-Work Audit Protocol* to clarify our approach. Specifically, Tables 2–7 are designed as structured *representative audits* rather than an exhaustive literature census. Our selection strictly follows a **family-coverage criterion**. For each modality, we systematically sample papers to capture a diverse range of generation paradigms, construction strategies, and evaluation styles. We deliberately prioritize broad coverage across distinct methodological families over merely relying on citation counts, recency, or including near-duplicate methods from the same family.
> * **Enhanced Transparency via Annotations and Coding Schemes:** To make this structural design immediately interpretable, we now explicitly annotate each audited paper with its specific **Family** and **Role**. Furthermore, we have clearly defined the **✓ / △ / × coding scheme** utilized throughout the tables. These additions unequivocally demonstrate that the audited set is a rigorously structured cross-family sample, rather than an arbitrary convenience sample.
> * **Calibrated the Scope of Claims:** Accordingly, we have refined the wording in the manuscript to accurately reflect the intended level of generalization. We explicitly state that our gap analysis provides a *comparative snapshot* of representative works analyzed through our unified framework, rather than a field-wide statistical prevalence estimate derived from a comprehensive meta-analysis.

---

### Author Response · Authors · 2026-04-07
**Thanks for the Constructive Feedback**

We would like to express our sincere gratitude to all the reviewers for their valuable and meticulous comments. We have carefully addressed each point raised and revised the manuscript accordingly.

Notably, in the updated version, revised expressions are highlighted in **blue**, while newly added content is highlighted in **red**. Please feel free to let us know if you have any further questions or suggestions. Thank you.

---

### Decision · Action_Editor_W8Q8 · 2026-05-06

**Recommendation:** Accept as is

**Audience:**

Yes

**Audience Explanation:**

This survey provides a timely "audit-oriented" lens that is highly relevant to researchers working on LLM alignment, RAG optimization, and multi-modal systems.

**Claims And Evidence:**

Yes

**Claims Explanation:**

The authors have effectively categorized fragmented evaluation practices into a two-pillar architecture of quality and trustworthiness.
During the rebuttal, the authors have successfully addressed initial concerns regarding the "taxonomy drift" and the selection criteria for the audited papers. By formalizing the Representative-Work Audit Protocol (Section 1.1) and adding the Cross-Modal Synthesis (Section 8), they have clarified the organization and structure. The formal mathematical definitions of metrics are well-grounded in existing literature, and the "gap analysis" is now clearly framed as a comparative snapshot of representative methodological families rather than a global census.